# SELF-CONTRASTIVE LEARNING

## ABSTRACT

This paper proposes a novel supervised contrastive learning framework, called **Self-Contrastive (SelfCon) Learning**, that *self-contrasts* within multiple outputs from the different levels of a multi-exit network. SelfCon learning does not require additional augmented samples, which resolves the concerns of multi-viewed batch (e.g., high computational cost and generalization error). Furthermore, we prove that SelfCon loss guarantees the lower bound of label-conditional mutual information between the intermediate and the last feature. In our experiments including ImageNet-100, SelfCon surpasses cross-entropy and Supervised Contrastive (SupCon) learning without the need for a multi-viewed batch. We demonstrate that the success of SelfCon learning is related to the regularization effect associated with the single-view and sub-network.

## 1 INTRODUCTION

Recent studies have studied the success of deep neural networks by investigating how neural networks can encode representations with rich information (Tishby & Zaslavsky, 2015; Shwartz-Ziv & Tishby, 2017; Hjelm et al., 2018; Saxe et al., 2019a). Among the various approaches suggested, contrastive loss functions, designed to maximize the lower bound of mutual information (MI) between the target and context, have achieved considerable success in self-supervised representation learning first (Gutmann & Hyvärinen, 2010; Oord et al., 2018) and supervised learning recently (Khosla et al., 2020; Gunel et al., 2020; Wang et al., 2021). The main objective of the contrastive loss functions in supervision is to make representations from the same class closer and representations from different classes farther. To this end, they define positive samples, i.e., augmented samples from the same image or (augmented) images sharing the same class label, and negative samples, i.e., all other samples, for every data batch.

Contrasting two random augmented samples, often referred to as a multi-viewed batch, has shown impressive results in representation learning (Chen et al., 2020; He et al., 2020; Grill et al., 2020; Caron et al., 2020; Chen & He, 2020; Khosla et al., 2020), yet a multi-viewed batch causes the following issues. (1) A multi-viewed batch doubles the batch size, which is a huge burden on memory and computation. (2) Oracle needs to carefully choose the augmentation policies (Chen et al., 2020; Tian et al., 2020; Caron et al., 2020; Kim et al., 2020). (3) A multi-viewed contrastive task is domain-specific, because data-level augmentation requires specific domain knowledge such as image cropping and flipping (Verma et al., 2021). In spite of these concerns, a supervised contrastive learning framework (Khosla et al., 2020) still relies on multi-views, although they are not necessarily needed with the help of usable label information.

In this work, we propose a novel contrastive learning framework in supervision, called **Self-Contrastive (SelfCon) learning**, that does not require additional augmented samples. Instead, SelfCon uses a multi-exit framework (Teerapittayanon et al., 2016; Zhang et al., 2019a;b) where sub-networks produce multiple outputs for the same input and *self-contrasts* **within multiple outputs from the different levels of a single network.** For training, SelfCon learning can use the multi-viewed batch as well. However, the multi-exit framework already generates positive pairs from a single image, which can replace the augmentation based multi-view. In Figure 1, we compare SelfCon learning and Supervised Contrastive (SupCon, Khosla et al. (2020)) learning.

SelfCon learning improves the classification performance of the encoder network, owing to **(1) the increase of the lower bound of label-conditional MI between the intermediate and the last features, and (2) the regularization effect from the single-view and sub-network.** We theoretically demonstrate that SelfCon loss is the lower bound of label-conditional MI. Furthermore, unlike

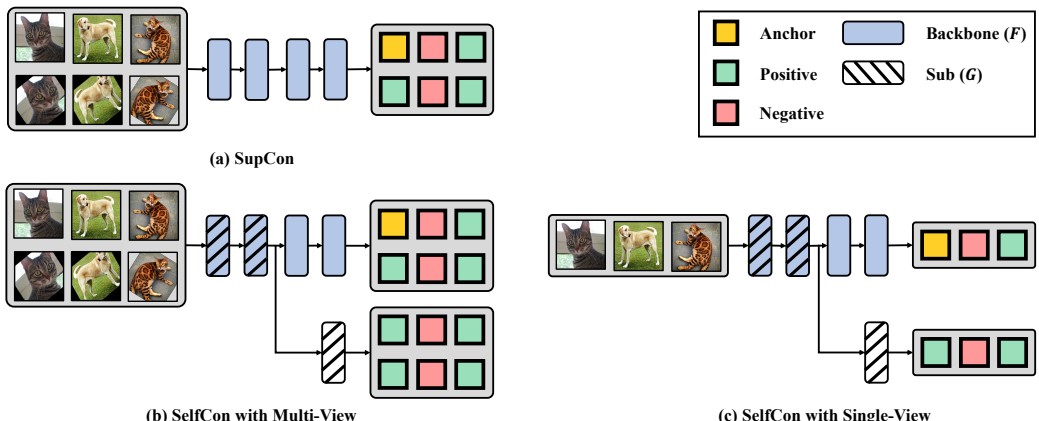

(a) SupCon

(b) SelfCon with Multi-View       (c) SelfCon with Single-View

Figure 1: **Comparison of learning framework in terms of augmentation and architecture.** Both SupCon (Khosla et al., 2020) and SelfCon use all the samples of the same ground-truth label as the positive pairs. In all three methods, every output can be an anchor feature. Specifically, in (b) and (c), an anchor from the backbone network contrasts other features from the backbone, as well as the features from the sub-network. Best seen in color.

SupCon loss, it encourages the encoder to learn the label information that only intermediate features can explain. Second, we empirically show that SelfCon learning reduces the generalization error with the single-view and sub-network. The former prevents the encoder from overfitting to each instance, and the latter regularizes the intermediate feature to be similar to the last feature.

The contributions of our paper can be summarized as follows:

[S3] We propose Self-Contrastive learning, a novel supervised contrastive framework between the multiple features from the different levels of a single network.

[S4] SelfCon loss guarantees the lower bound of label-conditional MI between the intermediate and the last features, and leads to increasing interaction information between the features and the label.

[S5.1, S5.2] SelfCon learning achieved higher classification accuracy for various benchmarks and architectures than cross-entropy and SupCon loss. Our empirical study of MI estimation provides evidence for superior performance.

[S5.3, S5.4] We investigate the advantages of single-viewed batch in terms of the generalization error and computation resources. We also identify that SelfCon learning benefits from the sub-network owing to the regularization effect, vanishing gradient, and ensemble prediction.

## 2 RELATED WORKS

### 2.1 CONTRASTIVE LEARNING

As the cost of data labeling increases exponentially, there is an increased need for acquiring representation under unsupervised scenarios. After Oord et al. (2018) proposed an InfoNCE loss (also called a contrastive loss), contrastive learning-based algorithms showed a remarkable improvement in performance (Chen et al., 2020; He et al., 2020; Grill et al., 2020; Caron et al., 2020; Chen & He, 2020). Most previous works use the structure of a Siamese network, which is a weight-sharing network applied on two or more inputs, with negative pairs (SimCLR (Chen et al., 2020), MoCo (He et al., 2020)), momentum encoders (MoCo (He et al., 2020), BYOL (Grill et al., 2020)), online clustering (SwAV (Caron et al., 2020)), or a stop-gradient operation (SimSiam (Chen & He, 2020)). While the softmax form is frequently used for the contrastive loss (Chen et al., 2020; He et al., 2020), recent state-of-the-art algorithms utilize an MSE loss (Grill et al., 2020; Chen & He, 2020), or a cross-entropy loss (Caron et al., 2020; 2021). Khosla et al. (2020), inspired by the success in self-supervised learning, propose a label-based contrastive loss in supervision, named SupCon loss. Supervised contrastive learning has also been extended to semantic segmentation (Wang et al.,

2021) and language tasks (Gunel et al., 2020). While the SupCon loss utilizes the output features from two randomly augmented images, we also contrast the features from different network paths by introducing the multi-exit framework (Teerapittayanon et al., 2016; Zhang et al., 2019a;b). We further propose a novel loss function that we can apply on the single-viewed batch.

## 2.2 MUTUAL INFORMATION

Mutual information is a measure to quantify the information held in a random variable about the other variable, and it has been used as a powerful tool to open the black box of deep neural networks (Tishby & Zaslavsky, 2015; Shwartz-Ziv & Tishby, 2017; Saxe et al., 2019a). As it is difficult to compute the MI exactly (Paninski, 2003), several works have proposed variational MI estimators based on neural networks, e.g., InfoNCE (Oord et al., 2018), MINE (Belghazi et al., 2018), NWJ (Nguyen et al., 2010), ML-CPC (Song & Ermon, 2020), and SMILE (Song & Ermon, 2019). Recently, MI estimator-based objectives have been proposed to improve the performance of contrastive learning (Hjelm et al., 2018; Bachman et al., 2019; Song & Ermon, 2020; Wu et al., 2020b) and knowledge distillation (Tian et al., 2019b; Ahn et al., 2019). Among previous approaches, we are highly motivated by those that aim to increase the MI between the intermediate and the last features. DIM (Hjelm et al., 2018) and AMDIM (Bachman et al., 2019) propose novel contrastive losses between the global features and local features (i.e., all pixels or patches of the intermediate features), while we contrast the refined local features via sub-networks. VID (Ahn et al., 2019) makes the student learn the distribution of the activations in the auxiliary teacher's intermediate features. Our method differs from VID in that we self-distill within the network itself. Zhang et al. (2019a;b) share a similar idea with the aforementioned works, but they implicitly increase MI using Kullback–Leibler (KL) divergence loss.

## 3 SELF-CONTRASTIVE LEARNING

**We propose a new supervised contrastive loss that maximizes the similarity of the outputs from different network paths by introducing the multi-exit framework.** We define the encoder structure with $\boldsymbol{F}$ as a backbone network and $\boldsymbol{G}$ as a sub-network that shares the backbone's parameters up to some intermediate layer. $\boldsymbol{T}$ denotes those sharing layers that produce the intermediate feature. The sub-network has additional non-sharing parameters attached after $\boldsymbol{T}$. Note that $\boldsymbol{F}$ and $\boldsymbol{G}$ include the projection head on the encoder. We highlight the **positive** and **negative** pairs with respect to an **anchor** sample, following Figure 1.

**SupCon loss** To mitigate the weaknesses of cross-entropy, such as the reduced generalization performance and the possibility of poor margins, Khosla et al. (2020) proposed a supervised version of the contrastive loss that defines the positive pairs as every sample with the same ground-truth label. We reformulate the SupCon loss function as follows:

$$\mathcal{L}_{sup} = -\sum_{i,p_1} \log \frac{\exp(\boldsymbol{F}(\boldsymbol{x}_i) \bullet \boldsymbol{F}(\boldsymbol{x}_{p_1})/\tau)}{\sum_{p_2} \exp(\boldsymbol{F}(\boldsymbol{x}_i) \bullet \boldsymbol{F}(\boldsymbol{x}_{p_2})/\tau) + \sum_{n} \exp(\boldsymbol{F}(\boldsymbol{x}_i) \bullet \boldsymbol{F}(\boldsymbol{x}_n)/\tau)} \tag{1}$$

$$\boxed{i \in I \equiv \{1, \ldots, 2B\}} \quad \boxed{p_* \in P_{i*} \equiv \{p \in I \setminus \{i\} | y_p = y_i\}} \quad \boxed{n \in N_i \equiv \{n \in I | y_n \neq y_i\}}$$

where $\bullet$ denotes the inner product, $B$ is the batch size, and $\tau$ is the temperature to soften or harden the softmax value. $I$ denotes a set of indices of the multi-viewed batch that concatenates the original $B$ images and the augmented ones, i.e., $\boldsymbol{x}_{B+i}$ is an augmented pair of $\boldsymbol{x}_i$. $P_{i*}$ and $N_i$ are a set of positive and negative pair indices with respect to an anchor $i$, respectively. Note that Eq. 1 is the same as NT-Xent loss (Chen et al., 2020) when $P_{i*} \equiv \{(i+B) \bmod 2B\}$. We dropped the dividing term of sums for brevity.

**SelfCon loss** Besides minimizing the above loss, we aim to maximize the similarity between the outputs from the backbone and the sub-network. To this end, we define SelfCon loss, which forms a self-contrastive task for every output including the features from the sub-network. In Section 4, we clarify the connection of this loss with the label-conditional MI between the intermediate and the last features.

$$\mathcal{L}_{self} = -\sum_{\boldsymbol{\omega},\boldsymbol{\omega}_1} \sum_{i,p_1} \log \frac{\exp(\boldsymbol{\omega}(\boldsymbol{x}_i) \bullet \boldsymbol{\omega}_1(\boldsymbol{x}_{p_1})/\tau)}{\sum_{\boldsymbol{\omega}_2} \left( \sum_{p_2} \exp(\boldsymbol{\omega}(\boldsymbol{x}_i) \bullet \boldsymbol{\omega}_2(\boldsymbol{x}_{p_2})/\tau) + \sum_{n} \exp(\boldsymbol{\omega}(\boldsymbol{x}_i) \bullet \boldsymbol{\omega}_2(\boldsymbol{x}_n)/\tau) \right)} \quad (2)$$

$$i \in I \equiv \left\{ \begin{array}{l} \{1,\ldots,B\} \ \textbf{(SelfCon-S)} \\ \{1,\ldots,2B\} \ \textbf{(SelfCon-M)} \end{array} \right. \qquad \begin{array}{l} p_* \in P_{i*} \equiv \{p \in I | y_p = y_i\} \\ n \ \in N_i \ \equiv \{n \in I | y_n \neq y_i\} \end{array}$$

$\boldsymbol{\omega}, \boldsymbol{\omega}_1, \boldsymbol{\omega}_2 \in \boldsymbol{\Omega} = \{\boldsymbol{F}, \boldsymbol{G}\}$, a function set of the backbone network and the sub-network. We exclude an anchor sample from the positive set to avoid contrasting the same feature, i.e., $P_{i*} \leftarrow P_{i*} \setminus \{i\}$ when $\boldsymbol{\omega} = \boldsymbol{\omega}_*$. While prevalent contrastive approaches (Khosla et al., 2020; Chen et al., 2020; He et al., 2020; Grill et al., 2020; Caron et al., 2020) force a multi-viewed batch generated by data augmentation, in SelfCon learning, the sub-network plays a role as the augmentation and provides an alternative view on the feature space. Therefore, we formulate our **SelfCon** loss function with **s**ingle-viewed batch (**SelfCon-S**); $I \equiv \{1,\ldots,B\}$ without the additional augmented samples. Besides, we also define **SelfCon** with **m**ulti-viewed batch (**SelfCon-M**) loss, i.e., $\mathcal{L}_{self}$ with $I \equiv \{1,\ldots,2B\}$.

In the development of our SelfCon loss formulation, we can further use multiple sub-networks, i.e., $\boldsymbol{\Omega} = \{\boldsymbol{F}, \boldsymbol{G}_1, \boldsymbol{G}_2, \ldots\}$. Appendix C provides the classification performance of the expanded network, which was comparable to that of the single sub-network in our experiments. Thus, we simply use a single sub-network throughout our paper.

## 4 DISCUSSIONS

**Question 1: How does SelfCon loss maximize the lower bound of label-conditional MI between the intermediate and the last features?**

We explain in the following propositions to answer Question 1. All the proofs can be found in Appendix A.

**Proposition 4.1.** *Let $\boldsymbol{x}$ and $\boldsymbol{z}$ be different samples that share the same class label c. Then, with some discriminator function modeled by a neural network $\boldsymbol{F}$ and $2(K-1)$ negative sample size, SupCon loss maximizes the lower bound of conditional MI between the output features of a positive pair,*

$$\log(2K-1) - \mathcal{L}_{sup}(\boldsymbol{x}, \boldsymbol{z}; \boldsymbol{F}, K) \leq \mathcal{I}(\boldsymbol{F}(\boldsymbol{x}); \boldsymbol{F}(\boldsymbol{z})|c). \quad (3)$$

**Proposition 4.2.** *Denote $\mathcal{L}_{self\text{-}s}$ as SelfCon loss with single-viewed batch. SelfCon-S loss maximizes the lower bound of MI between the output features from the backbone and the sub-network,*

$$\log(2K-1) - \mathcal{L}_{self\text{-}s}(\boldsymbol{x}; \{\boldsymbol{F}, \boldsymbol{G}\}, K) \leq \mathcal{I}(\boldsymbol{F}(\boldsymbol{x}); \boldsymbol{G}(\boldsymbol{x})|c). \quad (4)$$

**Proposition 4.3.** *Let $\boldsymbol{T}(\boldsymbol{x})$ be the intermediate feature of the backbone network, which is also an input to the auxiliary network path. Then the r.h.s. of Eq. 4 is upper-bounded by*

$$\mathcal{I}(\boldsymbol{F}(\boldsymbol{x}); \boldsymbol{T}(\boldsymbol{x})|c) = \underbrace{\mathcal{I}(\boldsymbol{F}(\boldsymbol{x}); c|\boldsymbol{T}(\boldsymbol{x})) - \mathcal{I}(\boldsymbol{F}(\boldsymbol{x}); c)}_{(\blacksquare)} + \underbrace{\mathcal{I}(\boldsymbol{F}(\boldsymbol{x}); \boldsymbol{T}(\boldsymbol{x}))}_{(\square)}. \quad (5)$$

Proposition 4.1 and 4.2 can be derived from the exact bound of InfoNCE (Poole et al., 2019). SupCon and SelfCon-S loss have $2(K-1)$ negative sample size because of the augmented negative pairs for SupCon and the sub-network features for SelfCon-S. Note that SelfCon-M loss has a similar upper bound as Eq. 3 and Eq. 4 (refer to Appendix A.2).

In Proposition 4.3, ($\blacksquare$) is an interaction information (Yeung, 1991) that measures the influence of $\boldsymbol{T}(\boldsymbol{x})$ on the amount of shared information between $\boldsymbol{F}(\boldsymbol{x})$ and $c$. SelfCon-S loss is related to increasing this interaction information, so that the intermediate feature enhances the correlation between the last feature and the label, which may result in improving the downstream classification tasks. In addition, ($\square$) implies that SelfCon loss increases the MI between the intermediate and the last features in the backbone. $\boldsymbol{T}(\boldsymbol{x})$ is distilled from $\boldsymbol{F}(\boldsymbol{x})$ that has richer class-related information and consequently, earlier layers are trained to produce better representation (Zhang et al., 2019a; Ahn et al., 2019). We believe this is theoretical evidence for the improved performance of SelfCon. Also, refer to Appendix A.4 for the connection to the classification performance of the SelfCon loss.

Table 1: **The results of linear evaluation on ResNet-18 and ResNet-50 for various datasets.** In the supervised setting, we compare our SelfCon-M and SelfCon-S with cross-entropy (CE) loss, supervised contrastive loss with multi-view (SupCon), and without multi-view (SupCon-S). **Bold** type is for all the values of which the standard deviation range overlaps with that of the best accuracy.

| Method | Single-View | ResNet-18 | | | ResNet-50 | | |
|---|---|---|---|---|---|---|---|
| | | CIFAR-10 | CIFAR-100 | Tiny-ImageNet | CIFAR-10 | CIFAR-100 | Tiny-ImageNet |
| CE | ✓ | $94.7_{\pm 0.1}$ | $72.9_{\pm 0.1}$ | $57.5_{\pm 0.3}$ | $94.9_{\pm 0.2}$ | $74.8_{\pm 0.1}$ | $62.3_{\pm 0.4}$ |
| SupCon | | $94.7_{\pm 0.2}$ | $73.0_{\pm 0.0}$ | $56.9_{\pm 0.4}$ | $95.6^{a}_{\pm 0.1}$ | $75.5^{a}_{\pm 0.2}$ | $61.6_{\pm 0.2}$ |
| SupCon-S[b] | ✓ | $94.9_{\pm 0.0}$ | $73.9_{\pm 0.1}$ | $58.4_{\pm 0.3}$ | $\mathbf{95.8}_{\pm 0.1}$ | $76.7_{\pm 0.1}$ | $62.0_{\pm 0.2}$ |
| SelfCon-M (ours) | | $95.0_{\pm 0.1}$ | $74.9_{\pm 0.1}$ | $59.2_{\pm 0.0}$ | $95.5_{\pm 0.1}$ | $76.9_{\pm 0.1}$ | $63.0_{\pm 0.2}$ |
| SelfCon-S (ours) | ✓ | $\mathbf{95.3}_{\pm 0.2}$ | $\mathbf{75.4}_{\pm 0.1}$ | $\mathbf{59.8}_{\pm 0.4}$ | $\mathbf{95.7}_{\pm 0.2}$ | $\mathbf{78.5}_{\pm 0.3}$ | $\mathbf{63.7}_{\pm 0.2}$ |

[a]We have re-implemented SupCon method (Khosla et al., 2020) and also run their official code for credibility, but the accuracy was slightly lower than their reported numbers.
[b]SupCon-S sets $I$ as $\{1, ..., B\}$ in Eq. 1. Although Khosla et al. (2020) did not propose the version of the single-view, we implemented SupCon-S since it is worth investigating the effect of multi-viewed batch.

**Question 2: Is it applicable to unsupervised representation learning?**

Good representations should get rid of redundant input information and extract meaningful features (Tishby & Zaslavsky, 2015). However, in Eq. 2, the backbone network can be an anchored function where the contrastive features are those from the backbone as well as the sub-network. Unfortunately, this might allow the last feature to follow the intermediate feature, learning redundant information about the input. This could be why SelfCon learning might *not* work in an unsupervised environment where there is no label information (refer to Appendix D.1). However, in Appendix D.2, we suggest that blocking the backbone from following the sub-network, i.e., removing the term in Eq. 2 where $\boldsymbol{\omega} = \boldsymbol{F}, \boldsymbol{\omega}_* = \boldsymbol{G}$, can mitigate the problem.

Supervised contrastive framework, which is the main interest of our paper, is away from the above problem owing to the label information. Note that the unsupervised version of SelfCon-S loss guarantees only the lower bound of ($\square$) in Proposition 4.3. Meanwhile, ($\blacksquare$), induced by the condition on label, encourages the backbone network to be in accordance with label information. Thus, we believe that jointly maximizing the lower bound of ($\square + \blacksquare$) offers an evidence for the success of supervised SelfCon learning.

## 5 EXPERIMENT

We presented the image classification accuracy for standard benchmarks, such as CIFAR-10, CIFAR-100 (Krizhevsky et al., 2009), Tiny-ImageNet (Le & Yang, 2015), ImageNet-100 (Tian et al., 2019a), and ImageNet (Deng et al., 2009), and extensively analyzed the results. We summarized the mean and standard deviation of top-1 accuracy over three random seeds for the reliability of the experimental results.

We used the optimal structure and position of the sub-network for all the network architectures (see Appendix C). We compared three types of the sub-network: the same structure with non-sharing layers of the backbone network (*same*), a structure with the number of non-sharing layers reduced in half (*small*), and a simple fully-connected layer (*fc*). The overall performance was comparable or better than the baselines. We observed a trend: in shallow networks the *same* structure was better, while *fc* was better in deeper networks. Besides, the performance was consistently good when the exit path is attached after the midpoint of the encoder (e.g., $2^{\text{nd}}$ block in ResNet or $3^{\text{rd}}$ block in VGG architecture). Complete implementation details are in Appendix B.

### 5.1 REPRESENTATION LEARNING

We measured the classification accuracy on the representation learning protocol (Chen et al., 2020), which consists of **2-stage training**, (1) pretraining an encoder network and (2) fine-tuning on a linear classifier with the frozen encoder (called a linear evaluation).

Table 2: **The classification accuracy on ResNet-18 and ResNet-50 for ImageNet-100.** Parentheses indicate the performance of ensemble prediction in the multi-exit framework (refer to Section 5.4). We reported the results with one random seed. Note that Acc@5 shows the same trend of Acc@1.

| Method | ResNet-18 | ResNet-50 |
|---|---|---|
| CE | 81.1 | 82.3 |
| SupCon | 83.0 | 86.3 |
| SupCon-S | 83.5 | 86.8 |
| SelfCon-M | $83.9_{(85.9)}$ | $86.8_{(88.1)}$ |
| SelfCon-S | $\mathbf{84.5}_{(\mathbf{86.0})}$ | $\mathbf{87.8}_{(\mathbf{88.2})}$ |

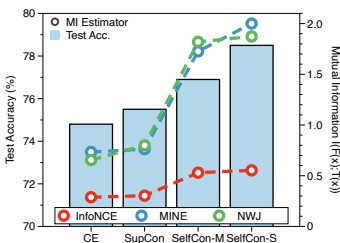

Figure 2: **Test accuracy and the estimated mutual information of different methods.** Mutual information estimators are measured between the intermediate and the last features.

The classification accuracy is summarized in Table 1. Interestingly, the loss functions in the single-viewed batch, such as SupCon-S and SelfCon-S, outperform their multi-view counterparts in all settings. Furthermore, our SelfCon learning, which trains with the sub-network, shows higher classification accuracy than CE and SupCon. The effects of the sub-network are analyzed in Section 5.4. We also experimented on the ImageNet-100 benchmark, of which 100 classes were randomly sampled (Tian et al., 2019a), to verify that SelfCon learning has the same effect on large scale datasets. Table 2 presents the consistent performance improvement of SelfCon learning. Refer to Appendix G for the ImageNet results.

## 5.2 MUTUAL INFORMATION ESTIMATION

Minimizing the SelfCon loss is highly related to maximizing the MI between the intermediate and the last features (Proposition 4.3). To empirically confirm this claim, we estimated MI using various estimators: InfoNCE (Oord et al., 2018), MINE (Belghazi et al., 2018), and NWJ (Nguyen et al., 2010). Specifically, we extracted the features of the CIFAR-100 dataset from the pretrained ResNet-50 encoders and computed the estimators. Then we optimized a simple 3-layer Conv-ReLU network with the MI estimator objectives. We consider the encoder output without the projection head, which differs from Section 3.

In Figure 2, SelfCon-M and SelfCon-S both show high $\mathcal{I}(\boldsymbol{F}(\boldsymbol{x}); \boldsymbol{T}(\boldsymbol{x}))$, which supports our claim, while the values of CE and SupCon are lower. Recall that $\boldsymbol{T}(\boldsymbol{x})$ is the intermediate feature of the backbone network, which is an input to the auxiliary network path. Although $\mathcal{I}(\boldsymbol{F}(\boldsymbol{x}); \boldsymbol{T}(\boldsymbol{x}))$ does not explicitly guarantee an increase in accuracy, our results imply that it has a positive correlation with the test accuracy of the backbone network. We believe that the richer information in earlier features makes the encoder output better representation because the intermediate feature is also the input for the subsequent layers.

We also found that SelfCon learning increases the information between $\boldsymbol{T}(\boldsymbol{x})$ and the label, implying that the intermediate feature is imbued with class-related knowledge, while the information between $\boldsymbol{T}(\boldsymbol{x})$ and the input $\boldsymbol{x}$ is decreased (refer to Table 12 in Appendix E). This suggests that SelfCon learning is in agreement with IB principle of training a deep neural network, i.e., fitting-compression phase (Saxe et al., 2019a). Furthermore, refer to Appendix K for the additional empirical evidence of correlation between MI and improved performance.

## 5.3 MULTI-VIEW VS. SINGLE-VIEW

We compare the multi-view and single-view in terms of generalization error, efficiency, and batch size.

**Single-view reduces generalization error.** In Figure 3, SupCon shows higher train accuracy but lower test accuracy than SupCon-S, and the same trend is observed with SelfCon-M and SelfCon-S. Compared to single-view, multi-view from the augmented image makes the encoder amplify the memorization of data and result in overfitting to each instance. Figure 4 shows that SelfCon-S gradually enhanced the generalization ability, while SelfCon-M or SupCon achieved little gain in test accuracy despite the fast convergence.

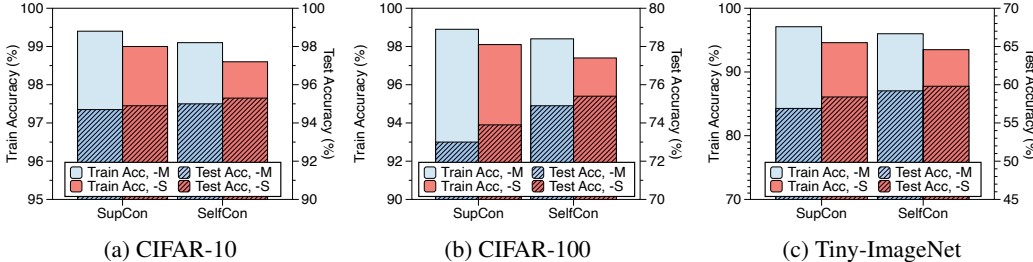

(a) CIFAR-10        (b) CIFAR-100        (c) Tiny-ImageNet

Figure 3: **Train accuracy and test accuracy of ResNet-18 for different views and loss functions.** The train and test accuracy are measured with a linear classifier during the linear evaluation. The axis on the left and right denotes the train accuracy and test accuracy, respectively.

Table 3: **Memory (GiB / GPU) and computation time (sec / step) comparison.** All numbers are measured with ResNet-18 training on 8 RTX 2080 Ti GPUs and Intel i9-10940X CPU. Note that FLOPS is for one sample. $B$ stands for batch size. For the results of ResNet-50, see Appendix H.

| Dataset (Image size) | Method | Params | FLOPS | $B = 256$ | | $B = 512$ | | $B = 1024$ | |
|---|---|---|---|---|---|---|---|---|---|
| | | | | Memory | Time | Memory | Time | Memory | Time |
| CIFAR-100 (32x32) | SupCon | **11.50 M** | 1.11 G | 2.14 | **0.13** | 2.35 | 0.16 | 3.18 | 0.27 |
| | SelfCon-S | 11.89 M | **0.56 G** | **1.83** | **0.13** | **2.03** | **0.14** | **2.54** | **0.18** |
| Tiny-ImageNet (64x64) | SupCon | **11.50 M** | 1.13 G | 2.01 | 0.14 | 2.69 | 0.17 | 3.97 | 0.31 |
| | SelfCon-S | 11.89 M | **0.56 G** | **1.75** | **0.13** | **2.05** | **0.13** | **2.68** | **0.18** |
| ImageNet-100 (224x224) | SupCon | **11.50 M** | 3.64 G | 3.34 | 0.51 | 5.34 | 1.04 | 9.54 | 2.11 |
| | SelfCon-S | 11.90 M | **1.82 G** | **2.54** | **0.35** | **3.38** | **0.70** | **5.67** | **1.38** |

**Single-view is efficient in terms of memory usage and computational cost.** In Table 3, we compared SupCon and SelfCon-S to observe the efficiency of the single-viewed batch. SelfCon-S requires a larger number of parameters owing to the extra sub-network but is more efficient in memory and computation. In both SupCon and SelfCon-S, the same batch size implies the same number of anchor features; however, they differ in memory consumption due to the data augmentation of the multi-viewed batch.

**Multi-view is advantageous for small batch size.** In supervised learning, large batch size reduces the generalization ability, which results in decreasing the performance (You et al., 2017; Luo et al., 2018; Wu et al., 2020a). We examined whether the performance in a supervised contrastive framework is also dependent on the batch size. Table 4 summarizes the results. The multi-viewed method, e.g., SupCon, which doubles the effective number of training data, outperformed the single-viewed counterpart in 64-batch experiments; the opposite was observed in all other batch sizes. For small batch size such as 64, doubling the effective batch size can make the learning stable, but as batch size gets larger the stabilizing effect from multi-views decreases and the necessity of regularization appears to be critical. Hence, the optimal batch size of single-viewed methods was higher (SupCon-S vs. SupCon and SelfCon-S vs. SelfCon-M) because single-view itself helps regularization (i.e., small batch with single-view might lead to under-fitting). Also, note that SelfCon learning with sub-network still surpasses the SupCon counterpart. Refer to Appendix I for the sensitivity study of the learning rates.

Table 4: **CIFAR-100 results on ResNet-18 with various batch sizes.** We omitted the standard deviation due to the lack of margin.

| | Batch Size | | | | |
|---|---|---|---|---|---|
| Method | 64 | 128 | 256 | 512 | 1024 |
| CE | 74.9 | 74.9 | 74.1 | 73.3 | 72.9 |
| SupCon | 74.8 | 73.8 | 72.9 | 72.5 | 73.0 |
| SupCon-S | 73.6 | 75.3 | 75.0 | 74.0 | 73.9 |
| SelfCon-M | **75.8** | 76.5 | 75.9 | 75.0 | 74.9 |
| SelfCon-S | 74.0 | **76.6** | **77.0** | **75.8** | **75.4** |

### 5.4 WHAT DOES THE SUB-NETWORK ACHIEVE?

**Regularization effect** SelfCon loss regularizes the sub-network to output the similar features to the backbone network. It prevents the encoder from overfitting to data, and it is effective in multi-viewed as well as single-viewed batch. In Figure 3, we confirmed the regularization effect (i.e., lower train accuracy, but higher test accuracy) by comparing each bar of the same color. The strong regularization of the sub-network helped SelfCon (-M, -S) outperform the SupCon counterparts. This trend is also observed in Table 4 and Figure 4.

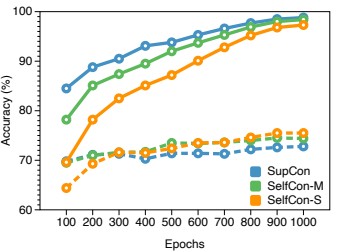 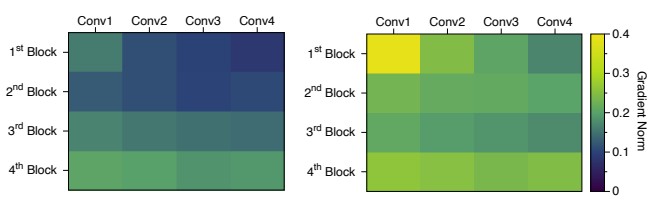

Figure 4: **CIFAR-100 accuracy at different training epochs.** The solid line is for train accuracy and the dashed line is for test accuracy. ResNet-18 is used.

Figure 5: **Gradient norm of each ResNet-18 block and convolutional layer.** We computed gradients from the SupCon loss (**Left**) and SelfCon-M loss (**Right**), both from the same initialized model. All convolution layers in the block are named by order.

Table 5: **Classification accuracy with the classifiers after backbone, sub-network, and the ensemble of them.** The encoder is pretrained by SelfCon-S loss function.

| Method | ResNet-18 | | | ResNet-50 | | |
|---|---|---|---|---|---|---|
| | CIFAR-10 | CIFAR-100 | Tiny-ImageNet | CIFAR-10 | CIFAR-100 | Tiny-ImageNet |
| Backbone | $\mathbf{95.3}_{\pm 0.2}$ | $75.4_{\pm 0.1}$ | $59.8_{\pm 0.4}$ | $\mathbf{95.7}_{\pm 0.2}$ | $78.5_{\pm 0.2}$ | $63.7_{\pm 0.2}$ |
| Sub-network | $92.6_{\pm 0.1}$ | $69.1_{\pm 0.3}$ | $53.5_{\pm 0.1}$ | $93.6_{\pm 0.2}$ | $73.3_{\pm 0.3}$ | $58.9_{\pm 0.6}$ |
| Ensemble | $95.2_{\pm 0.1}$ | $\mathbf{77.4}_{\pm 0.0}$ | $\mathbf{62.2}_{\pm 0.3}$ | $95.5_{\pm 0.1}$ | $\mathbf{80.0}_{\pm 0.2}$ | $\mathbf{65.7}_{\pm 0.5}$ |

**Vanishing gradient** SelfCon learning can send more abundant information to the earlier layers through the gradients flowing from the sub-networks. Previous works (Lee et al., 2015; Teerapittayanon et al., 2016; Zhang et al., 2019a) point out that the success of the multi-exit framework is owing to solving the vanishing gradient. We showed that the same argument applies to our SelfCon learning. Note that the sub-network is positioned after the 2nd block of the backbone. In Figure 5, a larger gradient flows up to the earlier layer in the SelfCon-M, while a large amount of the SupCon loss gradient vanishes. In particular, there is a significant difference in the gradient norm in the 2nd block.

**Ensemble with sub-network** The sub-network in SelfCon learning can also be used on downstream tasks such as image classification. In our previous experiments, we followed the linear evaluation protocol, fine-tuning a classifier with the frozen backbone network. The network pretrained on SelfCon learning has an exit path, which is similarly allowed to be frozen and used as linear evaluation. We thus demonstrated two additional types of linear evaluation scenarios: (1) fine-tuning a classifier after the sub-network output and (2) fine-tuning two classifiers after the backbone and the sub-network and ensembling two classifiers' predictions. Table 5 indicates that the sub-network could make appropriate, although not the best, predictions as the backbone did, while ensembling their predictions is found to be the most powerful trick we have come up with.

**Feature-level multi-view** One of the advantages of SelfCon learning is to relax the dependency on multi-viewed batch. This is accomplished by the multi-views on the representation space made by the parameters of the sub-network. In Figure 6, we visualized via Grad-CAM (Selvaraju et al., 2017) the gradient of SelfCon-S loss with respect to the intermediate layer of the backbone network, right before the exit path. Both networks focus on similar, but clearly different pixels of the same input image, implying that the sub-network learns another view in the feature space. As multi-view in contrastive learning requires domain-specific augmentation, recent studies have explored the domain-agnostic way of augmentation (Lee et al., 2020; Verma et al., 2021). SelfCon might be an intriguing future work in that auxiliary networks could be an efficient substitute for data augmentation.

### 5.5 ABLATION STUDY

**Different encoder architectures** We experimented with other architectures: VGG-16 (Simonyan & Zisserman, 2014) with Batch Normalization (BN) (Ioffe & Szegedy, 2015) and WRN-16-8 (Zagoruyko & Komodakis, 2016), and the results are presented in Table 6. The classification accuracy

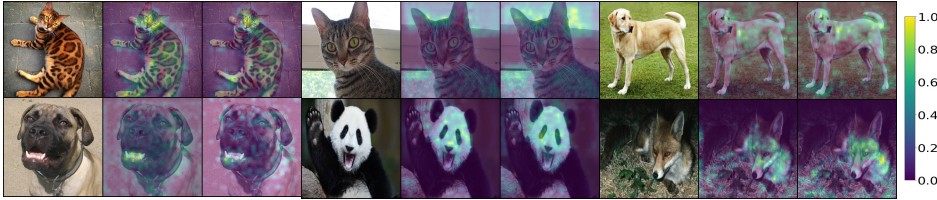

Figure 6: **Visualizations for the feature-level multi-view generated by the sub-network.** Along with the original image, each map visualizes the gradients from the sub-network (**Left**) and the backbone network (**Right**), respectively. We measured the gradient of the pretrained ResNet-18 with SelfCon-S loss.

Table 6: **The results of linear evaluation on WRN-16-8 and VGG-16 with BN for various datasets.** We tuned the best structure and position of the sub-network for each architecture. Appendix C summarizes the implementation details.

| Method | Single-View | WRN-16-8 | | | VGG-16 wih BN | | |
|---|---|---|---|---|---|---|---|
| | | CIFAR-10 | CIFAR-100 | Tiny-ImageNet | CIFAR-10 | CIFAR-100 | Tiny-ImageNet |
| CE | ✓ | $94.6_{\pm 0.1}$ | $73.6_{\pm 0.6}$ | $56.5_{\pm 0.5}$ | $\mathbf{93.8}_{\pm \mathbf{0.3}}$ | $71.2_{\pm 0.2}$ | $\mathbf{60.7}_{\pm \mathbf{0.1}}$ |
| SupCon | | $95.3_{\pm 0.0}$ | $75.1_{\pm 0.3}$ | $57.4_{\pm 0.3}$ | $93.6_{\pm 0.1}$ | $69.6_{\pm 0.1}$ | $57.3_{\pm 0.4}$ |
| SupCon-S | ✓ | $95.2_{\pm 0.1}$ | $76.0_{\pm 0.1}$ | $57.3_{\pm 0.5}$ | $\mathbf{93.8}_{\pm \mathbf{0.3}}$ | $71.1_{\pm 0.0}$ | $58.4_{\pm 0.2}$ |
| SelfCon-M (ours) | | $\mathbf{95.4}_{\pm \mathbf{0.2}}$ | $75.6_{\pm 0.1}$ | $58.7_{\pm 0.1}$ | $93.4_{\pm 0.1}$ | $71.7_{\pm 0.3}$ | $59.4_{\pm 0.1}$ |
| SelfCon-S (ours) | ✓ | $\mathbf{95.5}_{\pm \mathbf{0.0}}$ | $\mathbf{76.6}_{\pm \mathbf{0.1}}$ | $\mathbf{59.3}_{\pm \mathbf{0.2}}$ | $93.5_{\pm 0.1}$ | $\mathbf{72.0}_{\pm \mathbf{0.0}}$ | $\mathbf{60.7}_{\pm \mathbf{0.1}}$ |

for WRN-16-8 showed a similar trend as that of ResNet architectures. However, for VGG-16 with BN architecture, SupCon had lower performance than CE on every dataset. Although the contrastive learning approach does not seem to result in significant changes for the VGG-16 with BN encoders, SelfCon-S was better than or comparable to CE.

**1-stage training** 1-stage training framework, i.e., not decoupling the encoder pretraining and linear evaluation, on the single-viewed batch, is standard for supervised learning. With a multi-exit framework, Zhang et al. (2019a) propose Self-Distillation (SD), which distills logit information within the network itself, i.e., $\mathcal{L}_{SD} = \alpha\mathcal{L}_{CE} + (1 - \alpha)\mathcal{L}_{KL}$. We replaced the KL divergence term with $\mathcal{L}_{self\text{-}s}$ in Eq. 2, and distilled the features from the projection head, instead of the logits. From Table 7, we observed that adding cross-entropy loss to the sub-network (CE w/ Sub) improved the backbone network's classification performance. However, the results of SD suggested that there is a saturation of

Table 7: **CIFAR-100 1-stage training results on ResNet architectures.** † describes a modification to 1-stage training with multi-exit framework. We omitted the standard deviation. Parentheses indicate the sub-network's accuracy.

| Method | ResNet-18 | ResNet-50 |
|---|---|---|
| CE | 72.9 | 74.8 |
| CE w/ Sub† | $73.5_{(69.2)}$ | $76.2_{(72.3)}$ |
| SD† | $73.5_{(\mathbf{71.5})}$ | $76.1_{(\mathbf{73.3})}$ |
| SelfCon-S† | $74.5_{(70.6)}$ | $76.8_{(72.6)}$ |
| **SelfCon-S (2-stage)** | $\mathbf{75.4}_{(69.1)}$ | $\mathbf{78.5}_{(\mathbf{73.3})}$ |

increase even when the classifier of sub-network better converged. Meanwhile, we hypothesize that feature distillation of SelfCon-S makes the encoder learn better representation than logit distillation of classifiers. We would like to highlight that SelfCon loss in 2-stage training still demonstrated the best classification accuracy, as Khosla et al. (2020) argued that representation learning mitigates the poor generalization performance of CE-based 1-stage training.

# 6 CONCLUSION

We proposed SelfCon learning, which self-contrasts the features from the multiple levels of a network. SelfCon learning is free from the issues that multi-viewed batch triggers. We found that SelfCon loss maximizes the lower bound of label-conditional MI between the intermediate and the last features, and it is related to improving the classification performance. In addition, we analyzed why SelfCon is better by exploring the effect of single-view and sub-network. We verify by extensive experiments, including ImageNet-100 and ImageNet, that SelfCon-S loss outperforms CE and SupCon.

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

# Appendix

## A  PROOFS

### A.1  PROOF OF PROPOSITION 4.1

*Proof.*  We simply extend the exact bound of InfoNCE (Poole et al., 2019; Sordoni et al., 2021). Here, we consider the supervised setting where there are $C$ training classes. Without loss of generality, choose a class $c$ out of $C$ classes, and let $\boldsymbol{x}$ and $\boldsymbol{z}$ be different samples that share the same class label $c$. The derivation for the multi-view ($\boldsymbol{z}$ being an augmented sample of $\boldsymbol{x}$) is similar. For conciseness of the proof, we consider that no other image in a batch shares the same class. We prove that minimizing the SupCon loss (Khosla et al., 2020) maximizes the lower bound of conditional MI between two samples $\boldsymbol{x}$ and $\boldsymbol{z}$ given the label $c$:

$$\mathcal{I}(\boldsymbol{x};\boldsymbol{z}|c) \geq \log(2K-1) - \mathcal{L}_{sup}(\boldsymbol{x},\boldsymbol{z};\boldsymbol{F},K) \tag{6}$$

for some function $\boldsymbol{F}$ and hyperparameter $K$.

We start from Barber and Agakov's variational lower bound on MI (Barber & Agakov, 2004).

$$\mathcal{I}(\boldsymbol{x};\boldsymbol{z}|c) = \mathbb{E}_{p(\boldsymbol{x},\boldsymbol{z}|c)} \log \frac{p(\boldsymbol{z}|\boldsymbol{x},c)}{p(\boldsymbol{z}|c)} \geq \mathbb{E}_{p(\boldsymbol{x},\boldsymbol{z}|c)} \log \frac{q(\boldsymbol{z}|\boldsymbol{x},c)}{p(\boldsymbol{z}|c)} \tag{7}$$

where $q$ is a variational distribution. Since $q$ is arbitrary, we can set the sampling strategy as follows. First, sample $\boldsymbol{z}_1$ from the proposal distribution $\pi(\boldsymbol{z}|c)$ where $c$ is a class label of $\boldsymbol{x}$. Then, sample $(K-1)$ negative samples $\{\boldsymbol{z}_2,\cdots,\boldsymbol{z}_K\}$ from the distribution $\sum_{c'\neq c}\pi(\boldsymbol{z},c')$, so that these negative samples do not share the class label with $\boldsymbol{x}$. We augment each negative sample by random augmentation and concatenate with the original samples, i.e., $\{\boldsymbol{z}_2,\cdots,\boldsymbol{z}_K,\boldsymbol{z}_{K+1},\cdots,\boldsymbol{z}_{2K-1}\}$, where $\boldsymbol{z}_{K+i-1}$ is the augmented sample from $\boldsymbol{z}_i$ for $2 \leq i \leq K$. We define the unnormalized density of $\boldsymbol{z}_1$ given a specific set $\{\boldsymbol{z}_2,\cdots,\boldsymbol{z}_{2K-1}\}$ and $\boldsymbol{x}$ of label $c$ is

$$q(\boldsymbol{z}_1|\boldsymbol{x},\boldsymbol{z}_{2:(2K-1)},c) = \pi(\boldsymbol{z}_1|c) \cdot \frac{(2K-1) \cdot e^{\psi_{\boldsymbol{F}}(\boldsymbol{x},\boldsymbol{z}_1)}}{e^{\psi_{\boldsymbol{F}}(\boldsymbol{x},\boldsymbol{z}_1)} + \sum_{k=2}^{2K-1} e^{\psi_{\boldsymbol{F}}(\boldsymbol{x},\boldsymbol{z}_k)}} \tag{8}$$

where $\psi$ is often called a discriminator function (Hjelm et al., 2018), defined as $\psi_{\boldsymbol{F}}(\boldsymbol{u},\boldsymbol{v}) = \boldsymbol{F}(\boldsymbol{u}) \cdot \boldsymbol{F}(\boldsymbol{v})$ for some vectors $\boldsymbol{u},\boldsymbol{v}$. By setting the proposal distribution as $\pi(\boldsymbol{z}|c) = p(\boldsymbol{z}|c)$, we obtain the MI bound:

$$\mathcal{I}(\boldsymbol{x};\boldsymbol{z}|c) \geq \mathbb{E}_{p(\boldsymbol{x},\boldsymbol{z}_1|c)} \log \frac{q(\boldsymbol{z}_1|\boldsymbol{x},c)}{p(\boldsymbol{z}_1|c)} \tag{9}$$

$$= \mathbb{E}_{p(\boldsymbol{x},\boldsymbol{z}_1|c)} \log \frac{\mathbb{E}_{p(\boldsymbol{z}_{2:(2K-1)}|c)} q(\boldsymbol{z}_1|\boldsymbol{x},\boldsymbol{z}_{2:(2K-1)},c)}{p(\boldsymbol{z}_1|c)} \tag{10}$$

$$\geq \mathbb{E}_{p(\boldsymbol{x},\boldsymbol{z}_1|c)} \left[ \mathbb{E}_{p(\boldsymbol{z}_{2:(2K-1)}|c)} \log \frac{p(\boldsymbol{z}_1|c) \cdot \frac{(2K-1) \cdot e^{\psi_{\boldsymbol{F}}(\boldsymbol{x},\boldsymbol{z}_1)}}{e^{\psi_{\boldsymbol{F}}(\boldsymbol{x},\boldsymbol{z}_1)} + \sum_{k=2}^{2K-1} e^{\psi_{\boldsymbol{F}}(\boldsymbol{x},\boldsymbol{z}_k)}}}{p(\boldsymbol{z}_1|c)} \right] \tag{11}$$

$$= \mathbb{E}_{p(\boldsymbol{x},\boldsymbol{z}_1|c)p(\boldsymbol{z}_{2:(2K-1)}|c)} \log \frac{e^{\psi_{\boldsymbol{F}}(\boldsymbol{x},\boldsymbol{z}_1)}}{\frac{1}{2K-1} \sum_{k=1}^{2K-1} e^{\psi_{\boldsymbol{F}}(\boldsymbol{x},\boldsymbol{z}_k)}} \tag{12}$$

$$= \log(2K-1) - \mathcal{L}_{sup}(\boldsymbol{x},\boldsymbol{z};\boldsymbol{F},K). \tag{13}$$

where the second inequality is derived from Jensen's inequality. Because Eq. 12 is an expectation with respect to the sampled $\boldsymbol{x}$ and $\boldsymbol{z}_1$, the case where the anchor is swapped to $\boldsymbol{z}_1$ is also being considered.

A neural network $\boldsymbol{F}$ (backbone in our framework) with $L$ layers are formulated as $\boldsymbol{F} = f_L \circ f_{L-1} \circ \cdots \circ f_1$. Then, $\psi_{\boldsymbol{F}}(\boldsymbol{u}, \boldsymbol{v}) = \boldsymbol{F}(\boldsymbol{u}) \cdot \boldsymbol{F}(\boldsymbol{v}) = f_{1:L}(\boldsymbol{u}) \cdot f_{1:L}(\boldsymbol{v})$. We define another discriminator function as $\psi_{\boldsymbol{F}}^{\dagger}(\boldsymbol{u}, \boldsymbol{v}) = f_{(\ell+1):L}(\boldsymbol{u}) \cdot f_{(\ell+1):L}(\boldsymbol{v})$. Obviously, the following equivalence holds:

$$\psi_{\boldsymbol{F}}^{\dagger}(f_{1:\ell}(\boldsymbol{u}), f_{1:\ell}(\boldsymbol{v})) = \psi_{\boldsymbol{F}}(\boldsymbol{u}, \boldsymbol{v}). \tag{14}$$

Note that $f_{1:\ell}(\boldsymbol{u})$ is the $\ell$-th intermediate feature of input $\boldsymbol{u}$. Following the same procedure as in Eq. 9-13,

$$\mathcal{I}(f_{1:\ell}(\boldsymbol{x}); f_{1:\ell}(\boldsymbol{z})|c) \geq \mathbb{E}_{p(\boldsymbol{x}, \boldsymbol{z}_1|c) p(\boldsymbol{z}_{2:(2K-1)}|c)} \log \frac{e^{\psi_{\boldsymbol{F}}^{\dagger}(f_{1:\ell}(\boldsymbol{x}), f_{1:\ell}(\boldsymbol{z}_1))}}{\frac{1}{2K-1} \sum_{k=1}^{2K-1} e^{\psi_{\boldsymbol{F}}^{\dagger}(f_{1:\ell}(\boldsymbol{x}), f_{1:\ell}(\boldsymbol{z}_k))}} \tag{15}$$

$$= \mathbb{E}_{p(\boldsymbol{x}, \boldsymbol{z}_1|c) p(\boldsymbol{z}_{2:(2K-1)}|c)} \log \frac{e^{\psi_{\boldsymbol{F}}(\boldsymbol{x}, \boldsymbol{z}_1)}}{\frac{1}{2K-1} \sum_{k=1}^{2K-1} e^{\psi_{\boldsymbol{F}}(\boldsymbol{x}, \boldsymbol{z}_k)}} \tag{16}$$

$$= \log(2K - 1) - \mathcal{L}_{sup}(\boldsymbol{x}, \boldsymbol{z}; \boldsymbol{F}, K) \tag{17}$$

From above, as the intermediate feature is arbitrary to the position, we can obtain a similar inequality:

$$\mathcal{I}(f_{(\ell+1):L}(f_{1:\ell}(\boldsymbol{x})); f_{(\ell+1):L}(f_{1:\ell}(\boldsymbol{z}))|c) = \mathcal{I}(\boldsymbol{F}(\boldsymbol{x}); \boldsymbol{F}(\boldsymbol{z})|c) \tag{18}$$

$$= \mathcal{I}(f_{1:L}(\boldsymbol{x}); f_{1:L}(\boldsymbol{z})|c) \tag{19}$$

$$\geq \log(2K - 1) - \mathcal{L}_{sup}(\boldsymbol{x}, \boldsymbol{z}; \boldsymbol{F}, K). \tag{20}$$

$$\square$$

## A.2 PROOF OF PROPOSITION 4.2

*Proof.* In Section 4.1, we proved that SupCon loss maximizes the lower bound of conditional MI between the output features of a positive pair. We can think of another scenario where the network $\boldsymbol{F}$ now has a sub-network $\boldsymbol{G}$. Assume that the sub-network has $M > \ell$ layers: $\boldsymbol{G} = g_M \circ g_{M-1} \circ \cdots \circ g_1$. As we discussed in the paper, the exit path is placed after the $\ell$-th layer, so regarding our definition of the sub-network, $\boldsymbol{G}$ shares the same parameters with $\boldsymbol{F}$ up to $\ell$-th layer, i.e., $g_1 = f_1, g_2 = f_2, \cdots, g_\ell = f_\ell$. Define $\psi_{\boldsymbol{G}}(\boldsymbol{u}, \boldsymbol{v}) = \boldsymbol{G}(\boldsymbol{u}) \cdot \boldsymbol{G}(\boldsymbol{v})$

We introduce a discriminator function that measures the similarity between the outputs from the backbone and the sub-network, $\psi_{\boldsymbol{FG}}(\boldsymbol{u}, \boldsymbol{v}) = \boldsymbol{F}(\boldsymbol{u}) \cdot \boldsymbol{G}(\boldsymbol{v})$. Similarly, $\psi_{\boldsymbol{GF}}(\boldsymbol{u}, \boldsymbol{v}) = \boldsymbol{G}(\boldsymbol{u}) \cdot \boldsymbol{F}(\boldsymbol{v})$. Considering that the SelfCon-S loss has the anchored function of $\boldsymbol{F}$ and $\boldsymbol{G}$, we obtain an upper bound of two symmetric mutual information. Here, $\boldsymbol{z}_1 = \boldsymbol{x}$ because SelfCon-S loss is defined on the single-viewed batch and we assume that other images in a batch (i.e., $\boldsymbol{z}_2, \cdots, \boldsymbol{z}_K$) are sampled from the different class label with $\boldsymbol{x}$.

$$\mathcal{I}(\boldsymbol{F}(\boldsymbol{x}); \boldsymbol{G}(\boldsymbol{x})|c) + \mathcal{I}(\boldsymbol{G}(\boldsymbol{x}); \boldsymbol{F}(\boldsymbol{x})|c) \tag{21}$$

$$\geq \mathbb{E} \log \frac{e^{\psi_{\boldsymbol{FG}}(\boldsymbol{x}, \boldsymbol{x})}}{\frac{1}{2K-1} \left( e^{\psi_{\boldsymbol{FG}}(\boldsymbol{x}, \boldsymbol{x})} + \sum_{k=2}^{K} e^{\psi_{\boldsymbol{FG}}(\boldsymbol{x}, \boldsymbol{z}_k)} + \sum_{k=2}^{K} e^{\psi_{\boldsymbol{F}}(\boldsymbol{x}, \boldsymbol{z}_k)} \right)}$$

$$+ \mathbb{E} \log \frac{e^{\psi_{\boldsymbol{GF}}(\boldsymbol{x}, \boldsymbol{x})}}{\frac{1}{2K-1} \left( e^{\psi_{\boldsymbol{GF}}(\boldsymbol{x}, \boldsymbol{x})} + \sum_{k=2}^{K} e^{\psi_{\boldsymbol{GF}}(\boldsymbol{x}, \boldsymbol{z}_k)} + \sum_{k=2}^{K} e^{\psi_{\boldsymbol{G}}(\boldsymbol{x}, \boldsymbol{z}_k)} \right)} \tag{22}$$

$$= 2 \log(2K - 1) - 2\mathcal{L}_{self\text{-}s}(\boldsymbol{x}; \{\boldsymbol{F}, \boldsymbol{G}\}, K) \tag{23}$$

Due to the symmetry of mutual information,

$$\mathcal{I}(\boldsymbol{F}(\boldsymbol{x}); \boldsymbol{G}(\boldsymbol{x})|c) \geq \log(2K - 1) - \mathcal{L}_{self\text{-}s}(\boldsymbol{x}; \{\boldsymbol{F}, \boldsymbol{G}\}, K) \tag{24}$$

$$\square$$

In addition, we can similarly bound the SelfCon-M loss. As the derivation of SupCon loss bound, only consider the anchor $\boldsymbol{x}$ and its positive pair $\boldsymbol{z}_1$. When the anchored feature is $\boldsymbol{F}(\boldsymbol{x})$, the contrastive features are: $\boldsymbol{G}(\boldsymbol{x})$, $\boldsymbol{G}(\boldsymbol{z})$, and $\boldsymbol{F}(\boldsymbol{z})$. By symmetry, when the anchored feature is $\boldsymbol{G}(\boldsymbol{x})$, the contrastive features are: $\boldsymbol{F}(\boldsymbol{x})$, $\boldsymbol{F}(\boldsymbol{z})$, and $\boldsymbol{G}(\boldsymbol{z})$. As the derivation of the SupCon loss bound, we assume the augmented negative samples, i.e., $\{\boldsymbol{z}_2, \cdots, \boldsymbol{z}_K, \boldsymbol{z}_{K+1}, \cdots, \boldsymbol{z}_{2K-1}\}$.

$$
\begin{aligned}
&\mathcal{I}(\boldsymbol{F}(\boldsymbol{x}); \boldsymbol{G}(\boldsymbol{x})|c) + \mathcal{I}(\boldsymbol{F}(\boldsymbol{x}); \boldsymbol{G}(\boldsymbol{z})|c) + \mathcal{I}(\boldsymbol{F}(\boldsymbol{x}); \boldsymbol{F}(\boldsymbol{z})|c) \\
&+ \mathcal{I}(\boldsymbol{G}(\boldsymbol{x}); \boldsymbol{F}(\boldsymbol{x})|c) + \mathcal{I}(\boldsymbol{G}(\boldsymbol{x}); \boldsymbol{F}(\boldsymbol{z})|c) + \mathcal{I}(\boldsymbol{G}(\boldsymbol{x}); \boldsymbol{G}(\boldsymbol{z})|c)
\end{aligned} \tag{25}
$$

$$
\geq \frac{1}{3} \, \mathbb{E} \log \frac{e^{\psi_{FG}(\boldsymbol{x},\boldsymbol{x})} \cdot e^{\psi_{FG}(\boldsymbol{x},\boldsymbol{z}_1)} \cdot e^{\psi_F(\boldsymbol{x},\boldsymbol{z}_1)}}{\frac{1}{4K-1}\left(e^{\psi_{FG}(\boldsymbol{x},\boldsymbol{x})} + \sum_{k=1}^{2K-1} e^{\psi_{FG}(\boldsymbol{x},\boldsymbol{z}_k)} + \sum_{k=1}^{2K-1} e^{\psi_F(\boldsymbol{x},\boldsymbol{z}_k)}\right)}
$$

$$
+ \frac{1}{3} \, \mathbb{E} \log \frac{e^{\psi_{GF}(\boldsymbol{x},\boldsymbol{x})} \cdot e^{\psi_{GF}(\boldsymbol{x},\boldsymbol{z}_1)} \cdot e^{\psi_G(\boldsymbol{x},\boldsymbol{z}_1)}}{\frac{1}{4K-1}\left(e^{\psi_{GF}(\boldsymbol{x},\boldsymbol{x})} + \sum_{k=1}^{2K-1} e^{\psi_{GF}(\boldsymbol{x},\boldsymbol{z}_k)} + \sum_{k=1}^{2K-1} e^{\psi_G(\boldsymbol{x},\boldsymbol{z}_k)}\right)} \tag{26}
$$

$$
= \frac{2}{3} \, \log(4K-1) - 2\mathcal{L}_{self\text{-}m}(\boldsymbol{x}, \boldsymbol{z}; \{\boldsymbol{F}, \boldsymbol{G}\}, K) \tag{27}
$$

## A.3   PROOF OF PROPOSITION 4.3

$\boldsymbol{F}(\boldsymbol{x})$ and $\boldsymbol{G}(\boldsymbol{x})$ are the output features from the backbone network and the sub-network, respectively. Recall that $\boldsymbol{T}$ denotes the sharing layers between $\boldsymbol{F}$ and $\boldsymbol{G}$. $\boldsymbol{T}(\boldsymbol{x})$ is the intermediate feature of the backbone which is also an input to the auxiliary network path.

Before proving the following Lemma, we would like to note that the usefulness of mutual information should be carefully discussed on the stochastic mapping of a neural network. If a mapping $\boldsymbol{T}(\boldsymbol{x}) \mapsto \boldsymbol{F}(\boldsymbol{x})$ is a deterministic mapping, then the MI between $\boldsymbol{T}(\boldsymbol{x})$ and $\boldsymbol{F}(\boldsymbol{x})$ is degnerate because $\mathcal{I}(\boldsymbol{T}(\boldsymbol{x}); \boldsymbol{F}(\boldsymbol{x}))$ is either infinite for continuous $\boldsymbol{T}(\boldsymbol{x})$ (conditional differential entropy is $-\infty$) or a constant for discrete $\boldsymbol{T}(\boldsymbol{x})$ which is independent on the network's parameters (equal to $\mathcal{H}(\boldsymbol{T}(\boldsymbol{x}))$). However, for studying the usefulness of mutual information in a deep neural network, the map $\boldsymbol{T}(\boldsymbol{x}) \mapsto \boldsymbol{F}(\boldsymbol{x})$ is considered as a stochastic parameterized channel. In many recent works about information theory with DNN, they view the training via SGD is a stochastic process, and the stochasticity in the training procedure lets us define the MI with stochastically trained representations (Shwartz-Ziv & Tishby, 2017; Goldfeld et al., 2019; Saxe et al., 2019b; Goldfeld et al., 2019). Our theoretical claim focuses on the SelfCon loss as a *training* loss optimized by SGD algorithm. Therefore, analyzing the MI between the hidden representations while training with the SelfCon loss is based on the information theory to understand DNN (Tishby & Zaslavsky, 2015).

Also, information theory in deep learning, especially in contrastive learning, is based on the InfoMax principle (Linsker, 1989) which is about learning a neural network that maps a set of input to a set of output to maximize the average mutual information between the input and output of a neural network, subject to stochastic processes. This InfoMax principle is nowadays widely used for analyzing and optimizing DNNs. Most works for contrastive learning based on maximizing mutual information grounds on the InfoMax principle, and they are grounding on the stochastic mapping of an encoder. Moreover, Poole et al. (2019) rigorously discussed the mutual information with respect to a stochastic encoder. This is common practice in representation learning context where $\boldsymbol{x}$ is data and $\boldsymbol{z}$ is a learned stochastic representation.

**Lemma A.1.**

$$
\mathcal{I}(\boldsymbol{F}(\boldsymbol{x}); \boldsymbol{G}(\boldsymbol{x})|c) \leq \mathcal{I}(\boldsymbol{F}(\boldsymbol{x}); \boldsymbol{T}(\boldsymbol{x})|c) \tag{28}
$$

*Proof.* As $\boldsymbol{F}(\boldsymbol{x})$ and $\boldsymbol{G}(\boldsymbol{x})$ are conditionally independent given the intermediate representation $\boldsymbol{T}(\boldsymbol{x})$, they formulate a Markov chain as follows: $\boldsymbol{G} \leftrightarrow \boldsymbol{T} \leftrightarrow \boldsymbol{F}$. Under this relation, the following is

satisfied:

$$\mathcal{I}(\boldsymbol{F}(\boldsymbol{x}); \boldsymbol{G}(\boldsymbol{x})|c) = \mathcal{H}(\boldsymbol{F}(\boldsymbol{x})|c) - \mathcal{H}(\boldsymbol{F}(\boldsymbol{x})|\boldsymbol{G}(\boldsymbol{x}), c) \tag{29}$$
$$\leq \mathcal{H}(\boldsymbol{F}(\boldsymbol{x})|c) - \mathcal{H}(\boldsymbol{F}(\boldsymbol{x})|\boldsymbol{T}(\boldsymbol{x}), \boldsymbol{G}(\boldsymbol{x}), c) \tag{30}$$
$$= \mathcal{H}(\boldsymbol{F}(\boldsymbol{x})|c) - \int_{\mathbf{t},\mathbf{f},\mathbf{g}} p(\mathbf{t}, \mathbf{f}, \mathbf{g}|c) \log p(\mathbf{f}|\mathbf{t}, \mathbf{g}, c) d\mathbf{t}d\mathbf{f}d\mathbf{g} \tag{31}$$
$$= \mathcal{H}(\boldsymbol{F}(\boldsymbol{x})|c) - \int_{\mathbf{t},\mathbf{f}} p(\mathbf{t}, \mathbf{f}|c) \log p(\mathbf{f}|\mathbf{t}, c) d\mathbf{t}d\mathbf{f} \tag{32}$$
$$= \mathcal{H}(\boldsymbol{F}(\boldsymbol{x})|c) - \mathcal{H}(\boldsymbol{F}(\boldsymbol{x})|\boldsymbol{T}(\boldsymbol{x}), c) \tag{33}$$
$$= \mathcal{I}(\boldsymbol{F}(\boldsymbol{x}); \boldsymbol{T}(\boldsymbol{x})|c) \tag{34}$$

Eq. 30 is from the property of conditional entropy, and Eq. 32 is due to the conditional independence and marginalization of $\mathbf{g}$. $\qquad\square$

From Lemma A.1 and Eq. 24, we can prove the Proposition 4.3.

*Proof.*

$$\log(2K-1) - \mathcal{L}_{self\text{-}s}(\boldsymbol{x}; \{\boldsymbol{F}, \boldsymbol{G}\}, K) \tag{35}$$
$$\leq \mathcal{I}(\boldsymbol{F}(\boldsymbol{x}); \boldsymbol{G}(\boldsymbol{x})|c) \tag{36}$$
$$\leq \mathcal{I}(\boldsymbol{F}(\boldsymbol{x}); \boldsymbol{T}(\boldsymbol{x})|c) \tag{37}$$
$$= \mathcal{I}(\boldsymbol{F}(\boldsymbol{x}); \boldsymbol{T}(\boldsymbol{x}), c) - \mathcal{I}(\boldsymbol{F}(\boldsymbol{x}); c) \tag{38}$$
$$= \underbrace{\mathcal{I}(\boldsymbol{F}(\boldsymbol{x}); c|\boldsymbol{T}(\boldsymbol{x})) - \mathcal{I}(\boldsymbol{F}(\boldsymbol{x}); c)}_{(\blacksquare)} + \underbrace{\mathcal{I}(\boldsymbol{F}(\boldsymbol{x}); \boldsymbol{T}(\boldsymbol{x}))}_{(\square)} \tag{39}$$

$\qquad\square$

Strictly speaking, SelfCon-S loss does not guarantee the lower bound of either ($\blacksquare$) or ($\square$) in Eq. 39. However, SelfCon-S loss guarantees the label-conditional MI between the intermediate and the last feature, which is ($\blacksquare + \square$).

### A.4 CONNECTION BETWEEN SELFCON LOSS AND CLASSIFICATION PERFORMANCE

We used the variational inference to prove that SelfCon loss is the lower bound of label-conditional MI between the intermediate and the last features; thus, we assumed a probabilistic model as Eq. 8. When the anchor feature is similar to the negative pairs (i.e., different class representations, $\boldsymbol{z}_{2:(2K-1)}$), this model becomes a distribution with random mapping, and SelfCon loss cannot be optimized. Therefore, representations of other classes should be farther to decrease the gap between SelfCon loss and MI. **After all, SelfCon loss has improved performance because it aims to increase the lower bound of the label-conditional MI between $\boldsymbol{F}$ and $\boldsymbol{G}$ while increasing the distinction between different class representations.**

## B IMPLEMENTATION DETAILS

**Network architectures**   We modified the architecture of networks according to the benchmarks. For smaller scale of benchmarks (e.g., CIFAR-10, CIFAR-100, and Tiny-ImageNet) and the residual networks (e.g., ResNet-18, ResNet-50, and WRN-16-8), we changed the kernel size and stride of a convolution head to 3 and 1, respectively. We also excluded Max-Pooling on the top of the ResNet architecture for the CIFAR datasets. Moreover, for VGG-16 with BN, the dimension of the fully-connected layer was changed from 4096 to 512 for CIFAR and Tiny-ImageNet. MLP projection head for contrastive learning consisted of two convolution layers with 128 dimensions and one ReLU activation. For the architectures of sub-networks, refer to Appendix C.

**Representation learning**   We refer to the technical improvements used in SupCon, i.e., a cosine learning rate scheduler (Loshchilov & Hutter, 2016), an MLP projection head (Chen et al., 2020), and the augmentation strategies (Cubuk et al., 2019): {ResizedCrop, HorizontalFlip, ColorJitter, GrayScale}. ColorJitter and GrayScale are only used in the pretraining stage. We used 8 RTX 2080 Ti GPUs and set the batch size to 1024 for the pretraining and 512 for the linear evaluation. We used the batch size of 512 for ResNet-18 and 256 for ResNet-50 when pretraining on the ImageNet-100 dataset. We trained the encoder and the linear classifier for 400 epochs and 40 epochs, respectively, in ImageNet-100 benchmark. Meanwhile, we trained 1000 epochs and 100 epochs, respectively, in all other benchmarks.

Every experiment used SGD with 0.9 momentum and weight decay of 1e-4 without Nesterov momentum. All contrastive loss functions used temperature $\tau$ of 0.1. For a fair comparison to (Khosla et al., 2020), we set the same learning rate of the encoder network as 0.5. Then, we linearly scaled the learning rate according to the batch size (Goyal et al., 2017). For ImageNet-100 dataset, we used the small learning rate of 0.0625. We used 5.0 as a learning rate of the linear classifier for the residual architecture, but it was robust to any value and converged in nearly 20 epochs. Meanwhile, for VGG architecture, only a small learning rate of 0.1 converged.

**1-stage training**   In the 1-stage training protocol, we trained the encoder network jointly with a linear classifier on the single-viewed batch. Most of the experimental settings were same as those of representation learning, but we trained the encoder for 500 epochs on CIFAR and Tiny-ImageNet dataset. We used the batch size of 512 and the learning rate of 0.8. For the cross-entropy result of ImageNet-100 dataset, we trained for 90 epochs with learning rate decay of 0.1 after 30 and 60 epochs. Here, we used the batch size 512 and the learning rate 0.2.

In the multi-exit framework, we used linear combination of loss functions for the backbone and sub-network. We used only cross-entropy loss for the backbone network, and weighted linear combinations of various loss functions (e.g., KL divergence and SelfCon-S) for the sub-network. Self-Distillation (Zhang et al., 2019a) used the interpolation coefficient $\alpha$ of 0.5. For the 1-stage version of SelfCon loss, we follow the coefficient form in (Tian et al., 2019b): $\mathcal{L} = \mathcal{L}_{CE} + \beta\mathcal{L}_{self}$. We set the coefficient $\beta = 1.0$ for ResNet-18 and $\beta = 0.8$ for ResNet-50. Note that we used the outputs from the projection head instead of the logits. We used temperature $\tau = 3.0$ for SD and $\tau = 0.1$ for SelfCon loss.

## C   ABLATION STUDY ON SUB-NETWORK

The structure, position, and the number of sub-networks are important to the performance of SelfCon learning. First, in order to find a suitable structure of the sub-network, the following three structures were attached after the $2^{nd}$ block of an encoder: (1) a simple *fc*, fully-connected, layer, (2) *small* structure which reduced by half the number of layers in the non-sharing blocks, (3) *same* structure which is same as the backbone's non-sharing block structure. After we found the optimal structure, we fixed the structure of sub-network and found which position is the best. For ResNet architectures, there are three positions to attach; after the $1^{st}$, $2^{nd}$, and $3^{rd}$ block. For VGG-16 with BN, there are four positions and for WRN-16-8, there are two positions possible. Note that blocks are divided based on the Max-Pooling layer in VGG-16 with BN.

Table 8 presents the ablation study results for ResNet-18 and ResNet-50, and Table 9 presents the results for WRN-16-8 and VGG-16 with BN. We highlighted the selected structure and position in Table 8 and Table 9. For the ImageNet-100 dataset, we used *same* structure positioned after the $2^{nd}$ block of ResNet-18.

Obviously, there are many combinations of placing sub-networks, and Table 8 and Table 9 presented an interesting result that some performance was the best when sub-networks are attached to all blocks. It seems that increasing the number of positive and negative pairs by the various views improves the performance. It is consistent with the argument of CMC (Tian et al., 2019a) that the more views, the better the representation, but our SelfCon learning is much more efficient in terms of the computational cost and GPU memory usage. However, for the efficiency of the experiments and better understanding of the framework, we stuck to a single sub-network in all experimental settings.

Table 8: **The results of SelfCon-S loss according to the structure and position of sub-network.** The classification accuracy is for ResNet-18 **(Left)** and ResNet-50 **(Right)** on the CIFAR-100 benchmark.

| Structure | Position | | | Accuracy |
|---|---|---|---|---|
| | 1st Block | 2nd Block | 3rd Block | |
| FC | | ✓ | | $\mathbf{75.3}_{\pm\mathbf{0.1}}$ |
| Small | | ✓ | | $74.7_{\pm 0.2}$ |
| Same | | ✓ | | $74.5_{\pm 0.0}$ |
| FC | ✓ | | | $73.2_{\pm 0.2}$ |
| FC | | ✓ | | $75.4_{\pm 0.1}$ |
| FC | | | ✓ | $75.5_{\pm 0.1}$ |
| FC | ✓ | ✓ | ✓ | $74.5_{\pm 0.1}$ |

| Structure | Position | | | Accuracy |
|---|---|---|---|---|
| | 1st Block | 2nd Block | 3rd Block | |
| FC | | ✓ | | $\mathbf{78.5}_{\pm\mathbf{0.3}}$ |
| Small | | ✓ | | $\mathbf{78.1}_{\pm\mathbf{0.2}}$ |
| Same | | ✓ | | $77.4_{\pm 0.2}$ |
| FC | ✓ | | | $77.0_{\pm 0.2}$ |
| FC | | ✓ | | $\mathbf{78.5}_{\pm\mathbf{0.3}}$ |
| FC | | | ✓ | $77.4_{\pm 0.1}$ |
| FC | ✓ | ✓ | ✓ | $\mathbf{78.7}_{\pm\mathbf{0.5}}$ |

Table 9: **The results of SelfCon-S loss according to the structure and position of sub-network.** The classification accuracy is for WRN-16-8 **(Left)** and VGG-16 with BN **(Right)** on the CIFAR-100 benchmark.

| Structure | Position | | Accuracy |
|---|---|---|---|
| | 1st Block | 2nd Block | |
| FC | ✓ | | $74.4_{\pm 1.2}$ |
| Small | ✓ | | $76.2_{\pm 0.0}$ |
| Same | ✓ | | $\mathbf{76.6}_{\pm\mathbf{0.1}}$ |
| Same | ✓ | | $\mathbf{76.6}_{\pm\mathbf{0.1}}$ |
| Same | | ✓ | $76.5_{\pm 0.2}$ |
| Same | ✓ | ✓ | $76.5_{\pm 0.0}$ |

| Structure | Position | | | | Accuracy |
|---|---|---|---|---|---|
| | 1st Block | 2nd Block | 3rd Block | 4th Block | |
| FC | | ✓ | | | $71.4_{\pm 0.0}$ |
| Small | | ✓ | | | $\mathbf{71.5}_{\pm\mathbf{0.4}}$ |
| Same | | ✓ | | | $\mathbf{71.5}_{\pm\mathbf{0.3}}$ |
| FC | ✓ | | | | $70.9_{\pm 0.1}$ |
| FC | | ✓ | | | $71.4_{\pm 0.0}$ |
| FC | | | ✓ | | $72.0_{\pm 0.0}$ |
| FC | | | | ✓ | $71.5_{\pm 0.1}$ |
| FC | ✓ | ✓ | ✓ | ✓ | $\mathbf{72.5}_{\pm\mathbf{0.1}}$ |

# D EXTENSIONS OF SELFCON LEARNING

## D.1 SELFCON IN UNSUPERVISED LEARNING

Although we have experimented only in the supervision, our motivation of contrastive learning with a multi-exit framework can also be extended to unsupervised learning. We propose a SelfCon loss function for the unsupervised scenario and present the linear evaluation performance of ResNet-18 architecture on CIFAR-100 dataset.

**Loss function** Under the unsupervised setting, Chen et al. (2020) proposed a simple framework for contrastive learning of visual representations (SimCLR) with NT-Xent loss. SimCLR suggests a contrastive task that contrasts the augmented pair among other images. Therefore, the objective of SimCLR is exactly same as Eq. 1, while each sample has only one positive pair of its own augmented image, i.e., $P \equiv \{(i + B) \bmod 2B\}$. We denote this loss as $\mathcal{L}_{sim}$.

We fomulate SelfCon loss in unsupervised setting as SimCLR, using a positive set without label information. We formulate **SelfCon** loss with the **m**ulti-viewed and **u**nlabeled batch (**SelfCon-MU**) as follows:

$$\mathcal{L}_{self\text{-}mu} = -\sum_{\boldsymbol{\omega},\boldsymbol{\omega}_1}\sum_{i,p_1}\log\frac{\exp(\boldsymbol{\omega}(\boldsymbol{x}_i)\cdot\boldsymbol{\omega}_1(\boldsymbol{x}_{p_1})/\tau)}{\sum\limits_{\boldsymbol{\omega}_2}\left(\sum\limits_{p_2}\exp(\boldsymbol{\omega}(\boldsymbol{x}_i)\cdot\boldsymbol{\omega}_2(\boldsymbol{x}_{p_2})/\tau)+\sum\limits_{n}\exp(\boldsymbol{\omega}(\boldsymbol{x}_i)\cdot\boldsymbol{\omega}_2(\boldsymbol{x}_{n})/\tau)\right)}$$

(40)

$$\boxed{i \in I \equiv \{1,\dots,2B\}} \quad \boxed{p_* \in P_{i*} \equiv \{i,(i+B) \bmod 2B\}} \quad \boxed{n \in N_i \equiv I \setminus P_{i2}}$$

As SelfCon loss in supervised setting, we exclude anchor sample from $P_{i*}$ and $N_i$ to avoid contrasting the same feature, i.e., $P_{i*} \leftarrow P_{i*} \setminus \{i\}$ when $\boldsymbol{\omega} = \boldsymbol{\omega}_*$ and $N_i \leftarrow N_i \setminus \{i\}$ when $\boldsymbol{\omega} = \boldsymbol{\omega}_2$. Here, we used $\tau = 0.5$ for the unsupervised SelfCon loss and $\mathcal{L}_{sim}$.

Table 10: **The results under the unsupervised scenario.** We compared our SelfCon-MU and SelfCon-SU loss with SimCLR in the unsupervised setting. For the comparison with supervised learning, we also added the classification accuracy of CE loss. We used ResNet-18 encoder and CIFAR-100 dataset. **Accuracy\*** denotes the accuracy of SelfCon learning with the anchors only from the sub-network (see details in Appendix D.2).

| Method | CE | SimCLR | SelfCon-MU | SelfCon-SU |
|---|---|---|---|---|
| Multi-view | - | ✓ | ✓ | ✗ |
| Accuracy | $72.9_{\pm 0.1}$ | $63.3_{\pm 0.3}$ | $5.0_{\pm 0.1}$ | $6.4_{\pm 0.2}$ |
| **Accuracy\*** | - | - | $\mathbf{64.6_{\pm 0.1}}$ | $12.8_{\pm 0.1}$ |

We also formulate **SelfCon** loss with the **s**ingle-viewed and **u**nlabeled batch (**SelfCon-SU**) as follows:

$$\mathcal{L}_{self\text{-}su} = -\sum_{\boldsymbol{\omega},\boldsymbol{\omega}_1}\sum_{i,p_1} \log \frac{\exp(\boldsymbol{\omega}(\boldsymbol{x}_i)\bullet\boldsymbol{\omega}_1(\boldsymbol{x}_{p_1})/\tau)}{\sum_{\boldsymbol{\omega}_2}\left(\sum_{p_2}\exp(\boldsymbol{\omega}(\boldsymbol{x}_i)\bullet\boldsymbol{\omega}_2(\boldsymbol{x}_{p_2})/\tau) + \sum_n \exp(\boldsymbol{\omega}(\boldsymbol{x}_i)\bullet\boldsymbol{\omega}_2(\boldsymbol{x}_n)/\tau)\right)}$$

(41)

$$\boxed{i \in I \equiv \{1,\dots,B\}} \quad \boxed{p_* \in P_{i*} \equiv \{i\}} \quad \boxed{n \in N_i \equiv I \setminus P_{i2}}$$

Similarly, $N_i \leftarrow N_i \setminus \{i\}$ when $\boldsymbol{\omega} = \boldsymbol{\omega}_2$. For the positive set, since this loss is based on the single-viewed batch, we have an empty positive set when $\boldsymbol{\omega} = \boldsymbol{\omega}_*$.

**Experimental results** All implementation details for unsupervised representation learning are identical with those of supervised representation learning in Appendix B, except for temperature $\tau$ of 0.5 and linear evaluation learning rate of 1.0. We used a *small* sub-network attached after the 2nd block. Table 10 shows the linear evaluation performance of unsupervised learning on ResNet-18 in CIFAR-100 dataset. However, we empirically found that the encoder failed to converge with SelfCon-MU and SelfCon-SU loss.

D.2    SELFCON WITH ANCHORS ONLY FROM THE SUB-NETWORK

We suspect that Eq. 40 and 41 allow the backbone network to follow the sub-network, which makes the last feature learn more redundant information about the input variable, without any label information. Thus, unsupervised loss function under the SelfCon framework needs to modified.

When the anchor feature is from the backbone network, we remove the loss term which contrasts the features of sub-network. Strictly speaking, it does not perfectly prevent the backbone from following the sub-network, since there is no stop-gradient operation on the outputs of backbone network when the outputs of sub-network are the anchors. However, we hypothesize that it helps prevent the encoder from collapsing to the trivial solution by the contradiction of IB principle. We confirmed the performance of revised loss functions in both unsupervised and supervised scenarios.

**Loss function**

$$\mathcal{L}_{self\text{-}mu*} = \mathcal{L}_{sim} - \alpha\left(\sum_{\boldsymbol{\omega}_1}\sum_{i,p_1}\log\frac{\exp(\boldsymbol{G}(\boldsymbol{x}_i)\bullet\boldsymbol{\omega}_1(\boldsymbol{x}_{p_1})/\tau)}{\sum_{\boldsymbol{\omega}_2}\left(\sum_{p_2}\exp(\boldsymbol{G}(\boldsymbol{x}_i)\bullet\boldsymbol{\omega}_2(\boldsymbol{x}_{p_2})/\tau)+\sum_n\exp(\boldsymbol{G}(\boldsymbol{x}_i)\bullet\boldsymbol{\omega}_2(\boldsymbol{x}_n)/\tau)\right)}\right)$$

(42)

$$\boxed{i \in I \equiv \{1,\dots,2B\}} \quad \boxed{p_* \in P_{i*} \equiv \{i,(i+B)\bmod 2B\}} \quad \boxed{n \in N_i \equiv I \setminus P_{i2}}$$

We exclude anchor sample from $P_{i*}$ and $N_i$, and all notations are same as Eq. 40, except for the coefficient $\alpha$ where we used 1.0. For the supervised setting, simply change $P_{i*}$ to $\{p \in I|y_p = y_i\}$ and $\mathcal{L}_{sim}$ to $\mathcal{L}_{sup}$. Note that $P_{i*} \leftarrow P_{i*} \setminus \{i\}$ when $\boldsymbol{\omega}_* = \boldsymbol{G}$ and $N_i \leftarrow N_i \setminus \{i\}$ when $\boldsymbol{\omega}_2 = \boldsymbol{G}$. We get rid of the situation that the anchor $\boldsymbol{F}(\boldsymbol{x})$ contrasts the positive pair in sub-network $\boldsymbol{G}(\boldsymbol{x}_{p_*})$. Still,

Table 11: **CIFAR-100 results with SelfCon extensions. Accuracy\*** denotes the accuracy of SelfCon learning with the anchors only from the sub-network.

| Method | Architecture | Accuracy | **Accuracy\*** |
|---|---|---|---|
| SelfCon-M | ResNet-18 | $\mathbf{74.9}_{\pm \mathbf{0.1}}$ | $\mathbf{74.9}_{\pm \mathbf{0.2}}$ |
| SelfCon-S | | $75.4_{\pm 0.1}$ | $\mathbf{75.6}_{\pm \mathbf{0.1}}$ |
| SelfCon-M | ResNet-50 | $76.9_{\pm 0.1}$ | $\mathbf{77.7}_{\pm \mathbf{0.4}}$ |
| SelfCon-S | | $\mathbf{78.5}_{\pm \mathbf{0.3}}$ | $\mathbf{78.8}_{\pm \mathbf{0.1}}$ |

Table 12: **The detailed results of mutual information estimation.** $x$, $y$, $T(x)$, and $F(x)$ respectively denotes the input variable, label variable, intermediate feature, and the last feature. Recall that $T(x)$ is the intermediate feature of the backbone network, which is an input to the auxiliary network path. We summarized the average of estimated MI through multiple random seeds. We highlighted the MI between the intermediate and the last features, which is the main concern of the SelfCon loss. **Bold** type indicates the smallest values for $\mathcal{I}(x; T(x))$ and the largest values for $\mathcal{I}(y; T(x))$ and $\mathcal{I}(F(x); T(x))$, according to the IB principle.

| Estimator | MI | CE | SupCon | SelfCon-M | SelfCon-S |
|---|---|---|---|---|---|
| InfoNCE (Oord et al., 2018) | $\mathcal{I}(x; T(x))$ | 0.509 | 0.344 | **0.231** | 0.236 |
| | $\mathcal{I}(y; T(x))$ | 0.214 | 0.235 | 0.469 | **0.479** |
| | $\mathcal{I}(F(x); T(x))$ | 0.288 | 0.303 | 0.530 | **0.553** |
| MINE (Belghazi et al., 2018) | $\mathcal{I}(x; T(x))$ | 1.639 | 0.898 | 0.558 | **0.554** |
| | $\mathcal{I}(y; T(x))$ | 0.398 | 0.627 | 1.486 | **1.680** |
| | $\mathcal{I}(F(x); T(x))$ | 0.734 | 0.762 | 1.726 | **2.123** |
| NWJ (Nguyen et al., 2010) | $\mathcal{I}(x; T(x))$ | 1.485 | 0.930 | 0.601 | **0.589** |
| | $\mathcal{I}(y; T(x))$ | 0.390 | 0.546 | **1.450** | 1.389 |
| | $\mathcal{I}(F(x); T(x))$ | 0.655 | 0.799 | 1.820 | **1.873** |

the proposed loss function includes SimCLR loss ($\mathcal{L}_{sim}$) or SupCon loss ($\mathcal{L}_{sup}$) in the unsupervised or supervised setting, respectively.

$$\mathcal{L}_{self\text{-}su^*} = -\sum_{i,p_1} \log \frac{\exp(G(x_i) \cdot F(x_{p_1})/\tau)}{\sum_{p_2} \exp(G(x_i) \cdot F(x_{p_2})/\tau) + \sum_{\omega_2} \sum_n \exp(G(x_i) \cdot \omega_2(x_n)/\tau)} \quad (43)$$

$$\boxed{i \in I \equiv \{1, \ldots, B\}} \quad \boxed{p_* \in P_{i*} \equiv \{i\}} \quad \boxed{n \in N_i \equiv I \setminus P_{i2}}$$

For the supervised setting, we change the above equation as the equally-weighted linear combination of SupCon-S loss and Eq. 43 with $P_{i*} \equiv \{p \in I | y_p = y_i\}$. Note that we also exclude contrasting the anchor itself in SupCon-S loss term.

**Experimental results** In Table 10, we also reported the accuracy of SelfCon-MU and SelfCon-SU loss according to Eq. 42 and 43. Surprisingly, in this case, SelfCon-MU outperformed SimCLR loss (Chen et al., 2020), improving 1.3%p. Unfortunately, SelfCon-SU had not converged again, although it improved the result in a small amount compared to Eq. 41. While SelfCon-MU has SimCLR loss term which makes the backbone encoder still learn meaningful features, SelfCon-SU loss does not have the anchor features from the backbone, which makes the backbone hard to be trained. Table 11 summarizes SelfCon-M and SelfCon-S loss, removing the anchors from the backbone in the supervised setting, i.e., supervised version of Eq. 42 and 43. As we expected, these variants of SelfCon-M and SelfCon-S further improved the classification performance.

## E DETAILS OF MI ESTIMATION

Table 12 summarizes the full details of mutual information estimation, measured on CIFAR-100 with the pretrained ResNet-50 encoders. Note that we considered the encoder output without the projection head. We used three types of estimators: InfoNCE (Oord et al., 2018), MINE (Belghazi et al., 2018), and NWJ (Nguyen et al., 2010). As expected, SelfCon-M and SelfCon-S loss exhibited

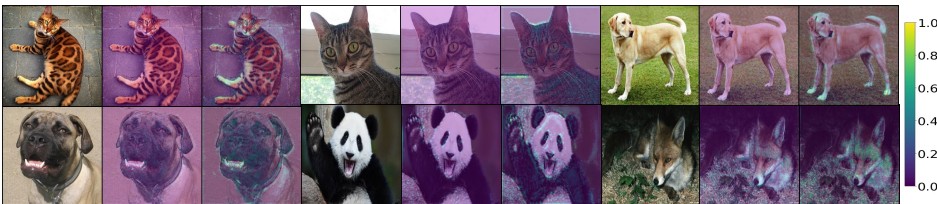

Figure 7: **Qualitative examples for mitigating vanishing gradient.** Along with the original image, we visualized the gradient when training with SupCon (**Left**) and SelfCon-M loss (**Right**). Note that all the gradients are from the same model checkpoint of ResNet-18.

larger MI between the intermediate and the last feature of the backbone network than CE and SupCon loss.

## F    QUALITATIVE EXAMPLES FOR VANISHING GRADIENT

In Figure 5, we have already showed that the sub-network solves the vanishing gradient problem through the visualization for gradient norms of each layer. In Figure 7, we also visualized qualitative examples using Grad-CAM (Selvaraju et al., 2017). We used the gradient measured on the last layer in the 2$^{nd}$ block when the sub-network is attached after the 2$^{nd}$ block. In order to compare the absolute magnitude of the gradient, it is normalized by the maximum and minimum values of the two methods, SelfCon-M and SupCon. As in Figure 7, SelfCon learning led to a larger gradient via sub-networks, and more clearly Grad-CAM highlighted the pixels containing important information in the images.

## G    IMAGENET RESULTS

We experimented with the full-scale ImageNet benchmark to enhance the reliability of Self-Con learning on the large-scale dataset. Due to the limited computational resources, we used ResNet-18 architecture for both the pretraining and linear evaluation. For the pretraining stage, we experimented with various batch sizes and epochs. Besides, we tuned the learning rate as 0.125 for 1024 batch size and 0.25 for 2048 batch size. In the linear evaluation stage, we trained the linear classifier with a batch size of 256, 40 epochs, and a learning rate of 6.0. Other hyperparameters are the same as Appendix B. For the cross-entropy result, we used the batch size of 256 and the learning rate of 0.1. The settings of epochs and the learning rate rule are the same as the ImageNet-100 experiments.

Table 13: **The classification accuracy on ResNet-18 for ImageNet.**

| Method | $B$ | Epoch | Acc@1 / Acc@5 |
|---|---|---|---|
| CE | 256 | 90 | 69.9 / 89.4 |
| SupCon | 1024 | 400 | 70.9 / 89.6 |
| SupCon-S | 1024 | 400 | 69.2 / 89.2 |
| SelfCon-M | 1024 | 400 | **71.2 / 90.1** |
| SelfCon-S | 1024 | 400 | 70.3 / 89.6 |
| SupCon | 1024 | 800 | 71.7 / 90.3 |
| SupCon-S | 1024 | 800 | 69.7 / 89.6 |
| SelfCon-M | 1024 | 800 | **71.9 / 90.4** |
| SelfCon-S | 1024 | 800 | 70.8 / 89.9 |
| SupCon | 2048 | 800 | 71.9 / 90.3 |
| SelfCon-S | 2048 | 800 | **72.0 / 90.4** |

We summarized the experimental results in Table 13. For the experiment with 1024 batch size and 400 epochs, it is still consistent that SelfCon-M showed better performance than SupCon, but the multi-viewed methods outperformed the single-viewed counterparts. We consider the possible reasons as follows. (1) Under-fitting issue due to the tremendous number of samples and relatively small size of architecture. (2) Insufficient number of training epochs (also refer to Figure 4). (3) Relatively small batch size with respect to the number of classes (1000) in ImageNet (also refer to Table 4). Therefore, we did two more ablation studies: (1) pretraining with 800 epochs and (2) pretraining with a larger batch size ($B = 2048$).

In the experiments with 800 epochs, the trend that multi-viewed methods outperformed the single-viewed counterparts did not change. However, the classification accuracy of each algorithm significantly increased as the pretraining epochs became longer. On the other hand, for the larger batch

Table 14: **Memory (GiB / GPU) and computation time (sec / step) comparison.** All numbers are measured with ResNet-50 training on 8 RTX 2080 Ti GPUs and Intel i9-10940X CPU. Note that FLOPS is for one sample. $B$ stands for batch size.

| Dataset (Image size) | Method | Params | FLOPS | $B = 256$ | | $B = 512$ | | $B = 1024$ | |
|---|---|---|---|---|---|---|---|---|---|
| | | | | Memory | Time | Memory | Time | Memory | Time |
| CIFAR-100 (32x32) | SupCon | **27.96 M** | 2.62 G | 4.00 | **0.11** | 6.40 | 0.14 | 11.29 | 0.20 |
| | SelfCon-S | 33.47 M | **1.31 G** | 2.73 | **0.11** | **3.92** | **0.12** | **6.28** | **0.16** |
| Tiny-ImageNet (64x64) | SupCon | **27.96 M** | 2.63 G | 4.41 | **0.21** | 6.71 | 0.26 | 11.84 | 0.36 |
| | SelfCon-S | 33.47 M | **1.32 G** | 2.98 | **0.21** | **4.21** | **0.23** | **6.82** | **0.27** |

size (i.e., $B = 2048$), we observed that SelfCon-S is better than SupCon. The difference seems to be marginal, but it is noteworthy that the training time for SelfCon-S is more than twice shorter than SupCon, and memory consumption is much lower. It is an important result in terms of the recent research flow for the compression in contrastive learning (Koohpayegani et al., 2020; Fang et al., 2021). From the result of SelfCon-S in the 2048 batch size, we are highly convinced that the trend in ImageNet is the same as the other benchmarks. The experiments for the larger architecture (e.g., ResNet-50, ResNet-101) or the larger batch size (e.g., 4096, 6144) are left for the future work.

## H MEMORY USAGE AND COMPUTATIONAL COST

In Table 14, we reported the computational cost for pretraining with ResNet-50. ImageNet-100 dataset result is not reported because ResNet-50 with batch size over 256 exceed the GPU limit. The overall trend is similar to that in Table 3.

## I SENSITIVITY STUDY FOR LEARNING RATE

In Table 4, we experimented with the supervised contrastive algorithms with various batch sizes and confirmed that the classification accuracy decreases in the large batch size. We supposed that this trend is induced by the regularization effect from the batch size. However, there could be a concern for using a sub-optimal learning rate on the large batch size. We further studied the sensitivity for the learning rate in a batch size of 1024 and summarized the results in Table 15.

Table 15: **CIFAR-100 results on ResNet-18 with various learning rates. Bold** type is for the best accuracy within each method.

| Method | Learning Rate | | | |
|---|---|---|---|---|
| | 0.1 | 0.5 | 1.0 | 1.5 |
| SupCon | 71.4 | 73.0 | 73.4 | **73.7** |
| SupCon-S | 72.1 | 73.9 | 74.6 | **75.0** |
| SelfCon-M | 72.9 | 74.9 | 75.4 | **75.8** |
| SelfCon-S | 73.7 | 75.4 | 75.7 | **76.2** |

We concluded that the performance comparison in Table 4 is consistent with hyperparameter tuning. The experimental results supported that a larger learning rate than 0.5 may be a better choice but the trend between all methods maintained in parallel with the learning rate of 0.5. Therefore, we stick to the initial learning rate of 0.5 that Khosla et al. (2020) had used.

## J ABLATION STUDY FOR DIFFERENT AUGMENTATION POLICIES

We could think of the optimal scenario where we have prior domain knowledge about a good augmentation policy. Also, there could be the opposite scenario where we do not know an optimal policy and just use the basic augmentations. Hence, we investigated three different augmentation policies as follows and compared the performances of SupCon and SelfCon-S under these scenarios.

Table 16: **CIFAR-100 results on ResNet-18 with various augmentation policies.**

| Method | Augmentation Policy | | |
|---|---|---|---|
| | Optimal | Standard | Simple |
| SupCon | **74.3**$_{\pm 0.1}$ | 73.0$_{\pm 0.0}$ | 72.0$_{\pm 0.3}$ |
| SelfCon-S | 72.5$_{\pm 0.0}$ | **75.4**$_{\pm 0.1}$ | **74.2**$_{\pm 0.2}$ |

- **Optimal** : We used RandAugment (Cubuk et al., 2020) for an optimal augmentation policy. RandAugment randomly samples $N$ out of 14 transformation choices (e.g., shear, translate, auto-Contrast, and posterize) with $M$ magnitude parameter. We used the optimized value of $N = 2$ and $M = 9$ in (Cubuk et al., 2020). It is also known that SupCon performs better with RandAugment policy (Khosla et al., 2020).

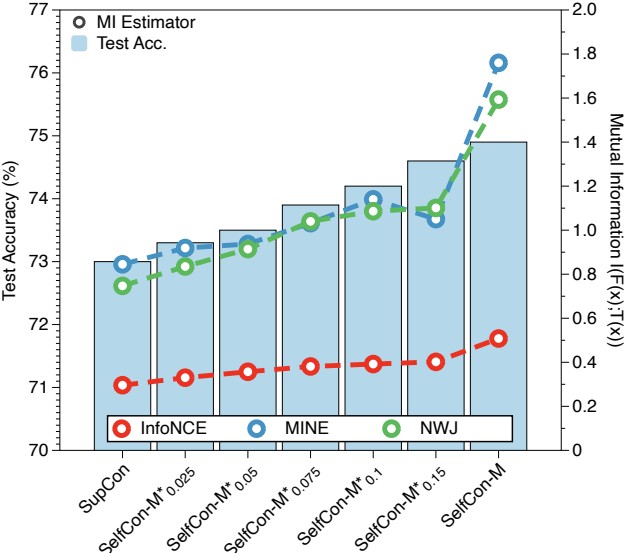

Figure 8: **Test accuracy and the estimated mutual information of different methods.** SelfCon-M*$_\alpha$ denotes SelfCon-M* loss with hyperparameter $\alpha$. We used ResNet-18 on CIFAR-100 dataset for the measurements.

- **Standard**: For standard augmentation, we used {RandomResizedCrop, RandomHorizontalFlip, RandomColorJitter, RandomGrayscale}. This is the same basic policy we used in the paper.

- **Simple**: When we do not have domain knowledge, it might be difficult to choose the appropriate augmentation policies. We assumed a scenario where we might not know that the color would be important in this visual recognition task. Therefore, we removed the color-related augmentation policies from **Standard** policy, i.e., we only used {RandomResizedCrop, RandomHorizontalFlip} for a simple augmentation policy.

The results are presented in Table 16. Interestingly, when we apply **Optimal** augmentations, SupCon outperformed SelfCon-S, but for **Standard** and **Simple** augmentations, SelfCon-S outperformed SupCon. SupCon with the multi-viewed batch can benefit more from the strong and optimized augmentation policy because training each sample twice more encouraged the memorization. Meanwhile, SelfCon learning did not work well with RandAugment (Cubuk et al., 2020), as SupCon degraded with the Stacked RandAugment (Tian et al., 2020) in their experiments. There would be an optimal policy for SelfCon learning but, finding an optimal policy, such as with RandAugment or AutoAugment (Cubuk et al., 2019), is not a trivial process and needs a lot of computational cost. SelfCon learning can relieve this concern and is more robust to the simple augmentation (i.e., weak augmentation policy).

## K ADDITIONAL EXPERIMENT FOR CORRELATION BETWEEN SELFCON LOSS AND MUTUAL INFORMATION

To clearly show the correlation between the mutual information and test accuracy, we experimented with the interpolation between SupCon loss and SelfCon-M loss (SupCon loss is a special case of SelfCon-M loss). However, the current formulation of Eq. 1 and Eq. 2 cannot make the exact interpolation between SupCon and SelfCon-M because the SelfCon-M loss should have negative pairs from different levels of a network (i.e., backbone and sub-network), but the SupCon loss cannot produce those. Therefore, we should modify the SelfCon-M loss to a supervised version of Eq. 42. In the supervised version of Eq. 42, we can break down the SupCon loss term and SelfCon-like loss

Table 17: **The detailed results of mutual information estimation.** Every notation is same as Table 12. We used ResNet-18 on CIFAR-100 dataset for the measurements.

| | | SupCon | SelfCon-M* | SelfCon-M* | SelfCon-M* | SelfCon-M* | SelfCon-M* | SelfCon-M |
|---|---|---|---|---|---|---|---|---|
| Estimator | MI | - | $\alpha = 0.02$ | $\alpha = 0.05$ | $\alpha = 0.075$ | $\alpha = 0.1$ | $\alpha = 0.15$ | - |
| InfoNCE | $\mathcal{I}(\boldsymbol{x}; \boldsymbol{T}(\boldsymbol{x}))$ | 0.285 | 0.296 | 0.277 | 0.299 | 0.293 | 0.232 | **0.203** |
| | $\mathcal{I}(\boldsymbol{y}; \boldsymbol{T}(\boldsymbol{x}))$ | 0.221 | 0.299 | 0.290 | 0.309 | 0.329 | 0.341 | **0.454** |
| | $\mathcal{I}(\boldsymbol{F}(\boldsymbol{x}); \boldsymbol{T}(\boldsymbol{x}))$ | 0.296 | 0.330 | 0.357 | 0.381 | 0.392 | 0.402 | **0.508** |
| MINE | $\mathcal{I}(\boldsymbol{x}; \boldsymbol{T}(\boldsymbol{x}))$ | 0.758 | 0.843 | 0.719 | 0.744 | 0.665 | 0.697 | **0.508** |
| | $\mathcal{I}(\boldsymbol{y}; \boldsymbol{T}(\boldsymbol{x}))$ | 0.616 | 0.719 | 0.700 | 0.834 | 0.928 | 0.961 | **1.261** |
| | $\mathcal{I}(\boldsymbol{F}(\boldsymbol{x}); \boldsymbol{T}(\boldsymbol{x}))$ | 0.845 | 0.919 | 0.937 | 1.032 | 1.140 | 1.050 | **1.760** |
| NWJ | $\mathcal{I}(\boldsymbol{x}; \boldsymbol{T}(\boldsymbol{x}))$ | 0.714 | 0.798 | 0.694 | 0.764 | 0.692 | 0.592 | **0.496** |
| | $\mathcal{I}(\boldsymbol{y}; \boldsymbol{T}(\boldsymbol{x}))$ | 0.467 | 0.594 | 0.641 | 0.808 | 0.799 | 0.843 | **1.287** |
| | $\mathcal{I}(\boldsymbol{F}(\boldsymbol{x}); \boldsymbol{T}(\boldsymbol{x}))$ | 0.747 | 0.835 | 0.914 | 1.039 | 1.086 | 1.101 | **1.593** |

term, with an interpolating parameter $\alpha$:

$$\mathcal{L}_{self\text{-}m*} = \mathcal{L}_{sup} - \alpha\left(\sum_{\boldsymbol{\omega}_1}\sum_{i,p_1} \log \frac{\exp(\boldsymbol{G}(\boldsymbol{x}_i) \cdot \boldsymbol{\omega}_1(\boldsymbol{x}_{p_1})/\tau)}{\sum_{\boldsymbol{\omega}_2}\left(\sum_{p_2}\exp(\boldsymbol{G}(\boldsymbol{x}_i) \cdot \boldsymbol{\omega}_2(\boldsymbol{x}_{p_2})/\tau) + \sum_{n}\exp(\boldsymbol{G}(\boldsymbol{x}_i) \cdot \boldsymbol{\omega}_2(\boldsymbol{x}_n)/\tau)\right)}\right)$$

(44)

$$\boxed{i \in I \equiv \{1, \dots, 2B\}} \quad \boxed{p_* \in P_{i*} \equiv \{p \in I | y_p = y_i\}} \quad \boxed{n \in N_i \equiv I \setminus P_{i2}}$$

where $P_{i*} \leftarrow P_{i*} \setminus \{i\}$ when $\omega_* = \boldsymbol{G}$ and $N_i \leftarrow N_i \setminus \{i\}$ when $\omega_2 = \boldsymbol{G}$. Therefore, if $\alpha = 0$, $\mathcal{L}_{self\text{-}m*}$ is equivalent to the SupCon loss and if $\alpha = 1$, $\mathcal{L}_{self\text{-}m*}$ is almost equivalent to SelfCon-M loss. Figure 8 describes the estimated mutual information and its relationship with classification performance via controlling the hyperparameter $\alpha$, and Table 17 summarizes the detailed estimation values of the intermediate feature with respect to the input, label, and the last feature. We observed a clear increasing trend of both MI and test accuracy as the contribution of SelfCon gets larger. Also, the detailed MI estimation values in Table 17 imply the same interpretation as the IB principle and Appendix E.

