# OpenReview forum: "Self-Contrastive Learning"
_ICLR.cc/2022/Conference — ICLR 2022 Submitted_

### Official Review · Reviewer_hbNU · 2021-10-29

**Correctness:** 3
**Technical Novelty And Significance:** 3
**Empirical Novelty And Significance:** 4
**Recommendation:** 6
**Confidence:** 4

**Main Review:**

-	To the best of my knowledge, this is the first work to propose self-contrastive learning via sub-networks. The paper holds significant methodological contributions and provides both theoretical and empirical grounds to support the effectiveness of the proposed model. While I believe the paper is clearly above the acceptance threshold, I have a few comments to clarify or supplement some points.
-	While the authors state that SelfCon has also potentials for an unsupervised setting (Appendix D.2.), the current version is still mostly focused on the supervised setting. I think it will be better if it is more clearly stated both in the abstract and introduction.
-	In the 3rd paragraph of the introduction, the authors stated that “the multi-exit framework already generates positive pairs from a single image, making the data augmentation redundant.” However, I think this is not entirely correct considering that SupCon-S outperforms SupCon even without the multi-exit framework (Table 1). Section 5.3 seems to provide more logical explanations that multi-view might cause the overfitting to each instance.
-	Can you provide additional explanations for why increasing the lower bound of label-conditional MI between the intermediate and the last features is desirable and improved the classification performance of the encoder network? It might be unclear considering that “good representations should get rid of redundant input information.” (Section 4). In addition, can you also explain why increasing the “influence of T(x)” on the amount of shared information between the last feature and the label? (cf. increasing the MI btw the last feature and the label).
-	Can you clarify the meaning of the indicator functions I(F(x_i) != F(x_p)) from the equations? I could not quite understand (1) why the indictor functions contain F or G rather than the labels and (2) why those in the numerators have inequalities rather than equalities. To define the positive pairs as samples with the same ground-truth labels, shouldn’t the indicator functions in the numerators be something like I(y_i == y_p)?


**Summary Of The Paper:**

Recent contrastive learning-based algorithms proposed various ways to make representations from semantically similar (positive) samples closer and those from semantically different (negative) samples farther. While they have shown remarkable performance improvement, they usually require a multi-viewed batch for defining the positive samples. In this paper, the authors proposed Self-Contrastive (SelfCon) Learning which self-contrasts within multiple outputs from the different levels of a multi-exit model. The authors theoretically demonstrated that SelfCon loss guarantees the lower bound of label-conditional mutual information (MI) between the intermediate and the last features. Then, they empirically showed the proposed method significantly outperforms the cross-entropy and Supervised Contrastive Learning (SupCon) baselines.

**Summary Of The Review:**

While I believe the paper is clearly above the acceptance threshold, I have a few comments to clarify or supplement some points in terms of the supervised settings, the rationale for increasing the label-conditional MI, and the proposed loss equations.

---

> ### Author Response · Authors · 2021-11-12
> **Response to Reviewer hbNU (1-2)**
>
> **Q4. Can you also explain why increasing the “influence of $T(x)$” on the amount of shared information between the last feature and the label?**
>
> **Increasing the influence of $T(x)$ on the amount of shared information between the last feature and the label makes the encoder's intermediate layers more informative about the label.** Note that SupCon depends on the extra augmented sample, so it leads to increasing the influence of $F(z)$ on the amount of shared information between $F(x)$ and the label, where $z$ is an augmented sample of $x$ (refer to Appendix A.2).  On the contrary, SelfCon does not depend on the other sample and rather uses its intermediate feature $T(x)$, thus maximizing the representation ability of the last features (related to interaction information $\mathcal{I}(F(x);c | T(x)) - \mathcal{I}(F(x);c)$). At the same time, since our SelfCon loss is highly correlated to $\mathcal{I}(F(x);T(x))$, the intermediate layers can be more informative about the label.
>
> **Q5. Why do the indicator functions contain $F$ or $G$ rather than the labels?**
>
> **We used the indicator functions in Eq. 1 and Eq. 2 to exclude computing similarity between the same feature itself.** Specifically, in Eq. 1, $\mathbb{1}[F(x_i) \neq F(x_p)]$ excludes the case when the anchor feature is contrasted with itself, since the positive set $P$ includes $i$. Similarly, in Eq. 2, $\mathbb{1}[\omega(x_i) \neq \omega'(x_p)]$ excludes the case when the anchor feature is contrasted with itself, but includes the case when the anchor features are from different levels of a network, i.e., $F(x_i)$ and $G(x_p)$.
>
> **However, we guess the definition for positive set $P$ made you misleading.** We originally defined the positive set $P$ as the samples with the same class label, including the anchor itself. We modified the equations in the revised paper (refer to (R1) in the overall response) as described below.
>
> **Eq. 1:** We modified the definition of a positive set $P$ and defined the negative set $N$ instead of $J$ for the sake of intuition. In detail, we defined the positive set $P_{i*}=$ {$p \in I \setminus$ {$i$} $| y_p=y_i$} that excludes the anchor index $i$. Besides, we removed $\mathbb{1}[F(x_i) \neq F(x_p)]$ because placing the indicator in log term is wrong, and we do not need this function.
>
> **Eq. 2:** First, we changed the term $\omega', \omega''$ to $\omega_1, \omega_2$, respectively. When $\omega \neq \omega_*$, positive set $P_{i}$ should include the anchor sample $x_i$. Therefore, different from Eq. 1, we included the sample $i$ into $P_{i*}$, and added the detailed explanation for the positive set (i.e., $P_{i*} \leftarrow P_{i*} \setminus$ {$i$}) to the paragraph below Eq. 2.
>
> **Q6. Why do the indicator functions have 'not equal'?**
>
> **It is also from our misleading definition of $P$.** As you mentioned, the numerator should include the positive samples, but due to our misleading definition of $P$, we used 'not equal' for excluding the anchor sample from the positive set.
>
> **References**
> [1] Chen, Ting, et al. "A simple framework for contrastive learning of visual representations." International conference on machine learning. PMLR, 2020.
> [2] Hjelm, R. Devon, et al. "Learning deep representations by mutual information estimation and maximization." International Conference on Learning Representations. 2018.
> [3] Oord, Aaron van den, Yazhe Li, and Oriol Vinyals. "Representation learning with contrastive predictive coding." arXiv preprint arXiv:1807.03748 (2018).
> [4] Shwartz-Ziv, Ravid, and Naftali Tishby. "Opening the black box of deep neural networks via information." arXiv preprint arXiv:1703.00810 (2017).

---

> ### Author Response · Authors · 2021-11-12
> **Response to Reviewer hbNU (1-1)**
>
> Thank you for your insightful comments. We are encouraged that you found our method novel among the contrastive learning works. We are also grateful that you went through our theoretical claims and examined in detail the empirical grounds that support SelfCon learning’s effectiveness.
>
> **Q1. Can you clearly state the potentials for an unsupervised setting?**
>
> **Although SelfCon learning with a sub-network can have the potentials for an unsupervised setting, it is far from our main contribution and overall flow of this paper.** The main focus of this paper is on the supervised learning framework. As we mentioned in the Introduction, “a supervised contrastive learning framework still relies on multi-views, although they are not necessarily needed with the help of usable label information.” Since there is label information, a supervised contrastive learning framework can work without augmentation-based multi-views, and we suggested SelfCon with multiple outputs from a multi-exit network.
>
> **Moreover, the extension of SelfCon learning to an unsupervised setting needs further technical modification.** We had to remove the anchor feature from the backbone network in the loss function (i.e., when the output from the backbone network becomes an anchor, we removed the loss term that contrasts the features of the sub-network to prevent the backbone from following sub-networks. (See Appendix D.2 for detailed loss function) With this loss variant, SelfCon-MU* outperformed SimCLR loss [1], but unfortunately, SelfCon-SU* did not converge. Therefore, this is the reason why we discussed it only in Discussion 2 in Section 4.
>
> **Q2. Is the sentence in the 3rd paragraph of the Introduction correct?**
>
> **We chose the word “redundant” because we wanted to say that using a multi-exit framework “can replace” the multi-view from data augmentation since the multi-exit framework already provides the feature-level multi-view.** We agree with your comment that the over-fitting issue from the multi-view makes the data augmentation useless. Besides, what makes the data augmentation redundant (i.e., not necessary) is actually the multiple positive pairs from the supervised contrastive framework. To clearly deliver the message, we changed the pointed-out sentence as “the multi-exit framework already generates positive pairs from a single image, replacing the augmentation based multi-view.” Thank you for your detailed comment.
>
> **Q3. Why increasing the lower bound of label-conditional MI is desirable and improves the classification performance?**
>
> **With proposition 4.3., we gave two clues for the correlation between MI and the improved performance: (1) Interaction information (i.e., $\mathcal{I}(F(x);c | T(x)) - \mathcal{I}(F(x);c)$) and (2) MI estimation between the intermediate and the last features (i.e., $\mathcal{I}(F(x);T(x))$).**
>
> **(1) Interaction information: We showed that minimizing SelfCon loss is highly correlated to maximizing interaction information, making the intermediate feature enhance the correlation between the last feature and the label.** We believe that this is theoretical evidence for the higher classification performance of SelfCon than SupCon. However, it is hard to estimate the interaction information. Besides, as we mentioned in Appendix A, we cannot strictly guarantee the bound of interaction information.
>
> **(2) MI estimation between the intermediate and the last features: Our intuition is that our SelfCon loss makes the intermediate feature have more meaningful information about the label, improving the classification performance.** $T(x)$ is distilled from $F(x)$ that has richer class-related information, and consequently, earlier layers are trained to produce better representation. Many prior works in contrastive learning literature argued that contrastive learning increases the lower bound of MI between the features or inputs [2][3], but few of them empirically showed the ground for their claims. In this work, we want to see ifSelfCon loss leads to the increase of MI between the intermediate and the last representations and thus measure MI with three estimators (refer to Figure 2).
>
> **Good representations should compress redundant input information *while increasing the correlation with label information*.** This is a key concept of information theory in deep learning [4]. Because of the above clues, we believe that minimizing SelfCon loss can make both the intermediate and last feature be rich with the label information.
>
> **We also added a detailed explanation for the connection between SelfCon and performance improvement.** Because we used variational inference and introduced the probabilistic model as Eq. 8, representations of other classes should be farther to decrease the gap between SelfCon loss and MI. After all, SelfCon loss has improved performance because it aims to increase the above information while increasing the distinction between different class representations (refer to Appendix A.4 ).

---

> ### Author Response · Authors · 2021-11-26
> **Further Discussion**
>
> Dear Reviewer hbNU,
>
> We sincerely appreciate your valuable comments for improving our work. We have answered your concerns (e.g., on the unsupervised setting of SelfCon) and put in efforts by modifying the words and equations to clarify our motivation.
>
> We would like to send you a gentle reminder that the rolling discussion is now at the end of its period. We hope we have addressed your concerns, and we look forward to knowing if any of our response is still in question.
>
> Thanks,
> Authors

---

> ### Comment · Reviewer_hbNU · 2021-12-05
> **Post-Rebuttal**
>
> Thank you for the detailed feedback on my comments. Although I think that the quality of the manuscript would be further increased if revised according to the authors' feedbacks, unfortunately, I also agree with some of the concerns from other reviewers. Therefore, I lowered my rating to 6.

---

### Official Review · Reviewer_MBzi · 2021-11-01

**Correctness:** 2
**Technical Novelty And Significance:** 2
**Empirical Novelty And Significance:** 3
**Recommendation:** 5
**Confidence:** 4

**Main Review:**

Strength:
1. The idea of not using multi-views in contrastive learning is important. Good quality of data augmentation is an issue in contrastive learning.
2. The authors did a lot of analysis and qualitative experiments to explain their method and motivations.

Weakness:
1. The main theoretical contribution which is self-con can maximize the mutual information between intermediate and last feature is incorrect. To prove this claim (prop 4.3), the authors use the inequality that $I(F(x), G(x)) \leq I(F(x), T(x))$. Then the authors claim that the negative self-con loss is a lower bound of $I(F(x), T(x))$. Therefore, minimizing the self-con loss can increase the mutual information between $F(x)$ and $T(x)$. However, the equity may not hold. Especially, in the authors' prop, maximize negative self-con loss can only increase $I(F(x), G(x))$ but not $I(F(x), T(x))$. There is a gap between $I(F(x), G(x))$ and $I(F(x), T(x))$.  Maximizing $I(F(x), G(x))$ does not imply maximizing $I(F(x), T(x))$.
2. On the other hand, since $F(x)$ is a function of $T(x)$, the mutual information of $I(F(x), T(x))$ is either zero or infinity. (https://stats.stackexchange.com/questions/465056/mutual-information-between-x-and-fx) I am not sure why the authors want to maximize it.
The authors somehow shows that the mutual information between the intermediate layer and the last layer is finite in figure. I think this is the problem of the estimation.
3. In Table 2, why does the authors only report one seed? Multiple seeds are needed. Confidence interval needs to be reported.
4. In Table 4, why does single view benefits from larger batch size? What about batch size 512?
5. It seems that you use labels when building self-contrastive learning. If you already use the labels, why do not you compare with fully-supervised approaches? What's the motivation of not using fully-supervised approaches?

Minor:
1. The usage of "Markov chain" in the proof of prop 4.3 may be inaccurate. The arrows should be $F \leftarrow T \rightarrow G$.

-----------------------
Updates:
I thank the authors for detailed rebuttal and answer further questions. I increased my overall score from 3 to 5 and empirical novelty from 2 to 3. The authors add results for more batch sizes and explain that they post a new empirical method and not a theoretical understanding. However, in the original paper, the authors explain their method from the prop 3. Prop 3 does not tell the readers more than minimizing the mutual information between $G(x)$ and $F(x)$. I am against motivating the method from an invalid theoretical statement. So I am still against accepting the paper. The followup motivations like distilling good information from last layer to the intermediate layer also sounds vague. The authors may rewrite the paper in a fully empirical way without any vague theoretical motivation.


**Summary Of The Paper:**

This paper provides a new way of supervised contrastive learning which does not require multi-views. The main idea is to use different functions for contrastive pairs which allows the usage of single views in contrastive learning.

**Summary Of The Review:**

The authors propose a new method which does not require multiple views in contrastive learning. It has the potential to motivate follow-up works. However, the theoretical motivation seems to be incorrect. The author does not consider the gap in the lower bound. So I vote for rejection.

---

> ### Author Response · Authors · 2021-11-12
> **Response to Reviewer MBzi (1-3)**
>
> **Q5. Why do you not compare with fully-supervised methods?**
>
> **We have already compared our novel supervised contrastive learning framework, SelfCon, with the fully-supervised method, such as cross-entropy (CE), CE with sub-network, and Self-Distillation [22] (refer to Table 7).** CE with sub-network means that jointly training the sub-networks with the cross-entropy loss, which is equally weighted as the backbone network. Besides, Self-Distillation is the fully-supervised method with the multi-exit framework and knowledge distillation. We thought that those two methods are highly correlated to our SelfCon learning; thus, we actually compared and summarized the experimental results in the paper.
>
> If you would like to check other "fully-supervised methods", please let us know the list of algorithms. We will run the experiments as many as we can in this rebuttal period.
>
> **Q6. The arrows of the Markov chain should be changed.**
>
> **Thank you for your detailed feedback. We modified the direction of the Markov chain in the revision.**
>
> **References**
> [1] Oord, Aaron van den, Yazhe Li, and Oriol Vinyals. "Representation learning with contrastive predictive coding." arXiv preprint arXiv:1807.03748 (2018).
> [2] Hjelm, R. Devon, et al. "Learning deep representations by mutual information estimation and maximization." International Conference on Learning Representations. 2018.
> [3] Sordoni, Alessandro, et al. "Decomposed mutual information estimation for contrastive representation learning." International Conference on Machine Learning. PMLR, 2021.
> [4] Song, Jiaming, and Stefano Ermon. "Understanding the Limitations of Variational Mutual Information Estimators." International Conference on Learning Representations. 2019.
> [5] Song, Jiaming, and Stefano Ermon. "Multi-label Contrastive Predictive Coding." Advances in Neural Information Processing Systems 33 (2020).
> [6] Kingma, Diederik P., and Max Welling. "Auto-encoding variational bayes." arXiv preprint arXiv:1312.6114 (2013).
> [7] Rezende, Danilo, and Shakir Mohamed. "Variational inference with normalizing flows." International conference on machine learning. PMLR, 2015.
> [8] Mescheder, Lars, Sebastian Nowozin, and Andreas Geiger. "Adversarial variational bayes: Unifying variational autoencoders and generative adversarial networks." International Conference on Machine Learning. PMLR, 2017.
> [9] Shwartz-Ziv, Ravid, and Naftali Tishby. "Opening the black box of deep neural networks via information." arXiv preprint arXiv:1703.00810 (2017).
> [10] Goldfeld, Ziv, et al. "Estimating Information Flow in Deep Neural Networks." International Conference on Machine Learning. PMLR, 2019.
> [11] Saxe, Andrew Michael, et al. "On the Information Bottleneck Theory of Deep Learning." International Conference on Learning Representations. 2018.
> [12] Goldfeld, Ziv, et al. "Differential entropy estimation under Gaussian noise." Proc. IEEE Int. Conf. Sci. Elect. Eng.(ICSEE). 2018.
> [13] Goldfeld, Ziv, and Yury Polyanskiy. "The information bottleneck problem and its applications in machine learning." IEEE Journal on Selected Areas in Information Theory 1.1 (2020): 19-38.
> [14] Belghazi, Mohamed Ishmael, et al. "Mutual information neural estimation." International Conference on Machine Learning. PMLR, 2018.
> [15] Bell, Anthony J., and Terrence J. Sejnowski. "An information-maximization approach to blind separation and blind deconvolution." Neural computation 7.6 (1995): 1129-1159.
> [16] Linsker, Ralph. "An application of the principle of maximum information preservation to linear systems." Advances in neural information processing systems. 1989.
> [17] Tian, Yonglong, Dilip Krishnan, and Phillip Isola. "Contrastive multiview coding." Computer Vision–ECCV 2020: 16th European Conference, Glasgow, UK, August 23–28, 2020, Proceedings, Part XI 16. Springer International Publishing, 2020.
> [18] Poole, Ben, et al. "On variational bounds of mutual information." International Conference on Machine Learning. PMLR, 2019.
> [19] Khosla, Prannay, et al. "Supervised contrastive learning." arXiv preprint arXiv:2004.11362 (2020).
> [20] Tian, Yonglong, Dilip Krishnan, and Phillip Isola. "Contrastive multiview coding." Computer Vision–ECCV 2020: 16th European Conference, Glasgow, UK, August 23–28, 2020, Proceedings, Part XI 16. Springer International Publishing, 2020.
> [21] You, Yang, Igor Gitman, and Boris Ginsburg. "Large batch training of convolutional networks." arXiv preprint arXiv:1708.03888 (2017).
> [22] Zhang, Linfeng, et al. "Be your own teacher: Improve the performance of convolutional neural networks via self distillation." Proceedings of the IEEE/CVF International Conference on Computer Vision. 2019.

---

> ### Author Response · Authors · 2021-11-12
> **Response to Reviewer MBzi (1-2)**
>
> **Q2. Is mutual information $\mathcal{I}(F(x),T(x))$ either zero or infinity? (contd.)**
>
> **However, for studying the usefulness of mutual information in a deep neural network, the map $X \mapsto T$ is considered as a stochastic parameterized channel.** In many recent works about information theory with DNN, they estimate the mutual information via assuming a noisy system in the network. Refer to Sec. 2.4 in [9], Sec. 1 in [10], Sec. 2 in [11], and other related works [12][13][14][15]. By adopting the noisy DNN framework, it is possible to provide theoretical guarantees that are not degenerate.
>
> **Also, information theory in deep learning, especially in contrastive learning, is based on the InfoMax Principle [16], which is about learning a neural network that maps a set of input $X$ to a set of output $T$ to maximize the average mutual information between $X$ and $T$, subject to specified constraints and/or noise processes.** This InfoMax Principle is nowadays widely used for analyzing and optimizing DNNs. Most works for contrastive learning based on maximizing mutual information grounds on this InfoMax Principle [1][2][17], and they are grounded on the stochastic mapping of an encoder.
>
> **The paper from Poole et al. [18], which our proofs are based on, rigorously discussed the mutual information with respect to a stochastic encoder.** In Sec.1, they stated, “the goal is to learn a stochastic representation of the data $p_\theta(y|x)$ that has maximal MI with $X$ subject to constraints on the mapping." This is common in a representation learning context where $x$ is data, and $y$ is a learned stochastic representation. Nevertheless, few works in contrastive learning rigorously stated about the stochastic encoder, and we also used the mutual information in a vague context. However, as you pointed out, it would be better to understand if precisely stated, so we added an explanation in Appendix A.3 thanks to your thoughtful feedback.
>
> **We hope that it is then clearly accepted why we want to maximize the MI between the intermediate and the last features.** We showed that MI between the intermediate and the last features is correlated to the downstream task performance of a backbone network. We believe that the richer information in earlier features makes the encoder output better representation because the intermediate feature is also the input for the subsequent layers (Section 5.2). Also, we found that SelfCon learning increases the information between the intermediate features and the label, implying that the intermediate feature is imbued with class-related knowledge (Appendix E).
>
> **Q3. Can you provide ImageNet-100 results with three random seeds?**
>
> **Despite one seed experiment, we are very convinced that SelfCon-S outperforms other methods because of the significant increase (i.e., +1.5% for SelfCon-S vs. SupCon and +3.0% for SelfCon-S-Ensemble vs. SupCon).** Of course, as you pointed out, it would be more confident to run three seeds; it is not sufficient to run three seeds with our computational resources. Instead, for a thorough comparison, we verified SelfCon learning with the ResNet-50 backbone network. Referring to the revised Table 2, there were similar increases from ResNet-18 to ResNet-50. We are certain that SelfCon methods outperformed SupCon methods in the same control setting since we used the same hyperparameter rules between every method (see Appendix B).
>
> **Furthermore, ImageNet results usually have small CIs, and many works report the performance with only one seed (as [19] or [20] did), and even the performance improvement is marginal.** Note that the ImageNet-100 dataset is a subset of ImageNet, having 100 classes out of the original 1000 classes and the same full images for each class as ImageNet.
>
> **Q4. Why does single-view benefit from a larger batch size?**
>
> **It is because the large batch size can reduce the regularization effect and result in a decreased test performance [21].** However, single-view itself helps regularization, compensating the decreased test performance with a large batch size. Please refer to the revised version of our paper and also (R3) of the overall response. SupCon uses multi-view that doubles the effective number of batch sizes. In the 64-batch experiment, doubling the effective batch size can make the learning stable; thus, SupCon outperformed SupCon-S, and SelfCon-M outperformed SelfCon-S. However, as the batch size becomes large, the stabilizing effect from multi-views decreases, and the necessity of regularization appears to be critical. Therefore, the optimal batch size of SupCon-S and SelfCon-S was higher than SupCon and SelfCon-M, respectively, since single-view itself helps regularization.

---

> > ### Comment · Reviewer_MBzi · 2021-11-12
> > **Q2 is also not resolved.**
> >
> > I thank again for the authors' detailed response. You mentioned that stochastic encoder may avoid the infinity information problem. However, do you use a stochastic encoder in your work, for example, a variational encoder? From what I understand, you do not use a stochastic encoder. In Poole et al.'s paper, they actually use a stochastic encoder.

---

> > > ### Author Response · Authors · 2021-11-14
> > > **Response to Reviewer MBzi (2-2)**
> > >
> > > Thank you again for the comments.
> > >
> > > **A DNN being trained can be viewed as a probabilistic model, while DNN after the full training is interpreted as a deterministic function. There are a lot of stochastic factors in training a neural network, such as initialization, stochastic optimization, or batch normalization layers.** Therefore, in deep learning, especially in representation learning, we make a probabilistic model with an unknown input distribution (i.e., $p(x)$) and a feature (latent) distribution (i.e., $p_\theta(z|x)$). In the implementation, inputs are drawn from some datasets like ImageNet, but this is viewed in the probabilistic model as an input variable $x$ that can be sampled from $p(x)$. The randomness of training a DNN also lets us define the feature distribution represented with network parameters. This is why many deep learning works use the distribution $p_\theta(z|x)$, not just a deterministic representation of given input $x$.
> > >
> > > **Moreover, with our best knowledge, Poole et al. (2019) [18] did not use a stochastic encoder for their toy experiments;** they just sampled inputs from a Gaussian distribution and estimated MI between the features from a typical FC+ReLU network. However, in their proof of variational bound of MI, they assumed a "stochastic encoder" or "stochastic representation" (see Sec. 1 and Sec. 2) because they need to define a distribution for inputs and representations. Our theoretical derivation, as well as many other works in contrastive learning or information theory, is grounded on their "stochasticity" assumption [9][10][11][12][13][14][15].
> > >
> > > **Please let us know if we missed the experimental implementation for the stochastic encoder in Poole et al. (2019) [18].**

---

> > > > ### Comment · Reviewer_MBzi · 2021-11-14
> > > > **Is normal distribution not a stochastic representation?**
> > > >
> > > > From my understanding, normal distribution in Poole et al. is a stochastic encoder. They use a normal distribution for the encoders in "Bias-variance tradeoffs for representation learning." paragraph. Feel free to correct me if I am wrong.
> > > >
> > > > We need to be specially careful about the stochasticity here since you are applying the transformation $G(x)$ to be a direct function of  $T(x)$. Even though you mention initialization, stochastic optimization, batch normalization, when you calculate the mutual information at a given point, you do not sample anything. More citation or explanation needs to be included for this stochastic encoder point. Note that this stochastic encoder is not a problem for traditional contrastive learning since they use different views which already includes a sampling process and randomness.

---

> > > > > ### Author Response · Authors · 2021-11-15
> > > > > **Response to Reviewer MBzi (3-2)**
> > > > >
> > > > > Thank you for more comments. We are encouraged with your inspiring discussions.
> > > > >
> > > > > **For the normal distribution for the encoders in Poole et al. (2019) [1] (i.e., $p_\rho(y_i | x_i) = \mathcal{N}(\rho_i x, \sqrt{1-\rho_i^2})$), it does not mean that they used some stochastic encoder, but they sampled $(x_i, y_i)$ from a correlated Gaussian with mean 0 and correlation of $\rho$ for a toy example (for details refer to Appendix B in [1]).** Therefore, there is no implementation for stochastic encoder, as you can check in [github1](https://github.com/mboudiaf/Mutual-Information-Variational-Bounds) (re-implementation of Poole's toy experiments) and [github2](https://github.com/ermongroup/smile-mi-estimator) (SMILE [2] official code which exactly followed the Poole's toy experiments).
> > > > >
> > > > > **Stochasticity in the "training procedure" can let us define the MI with the stochastically trained representations.** Our theoretical claim focuses on the SelfCon loss as a "training" loss optimized by stochastic gradient descent (SGD) algorithm. For MI estimation, feature vectors can be sampled during the SGD training process. After all, the MI bound guarantees the effectiveness of optimizing SelfCon loss, and here defining MI is not problematic as you had a concern with. We emphasize that analyzing the mutual information between the hidden representations while training a network by SGD is a natural way to understand DNN in IB theory [3, 4].
> > > > >
> > > > > [1] Poole, Ben, et al. "On variational bounds of mutual information." International Conference on Machine Learning. PMLR, 2019.
> > > > > [2] Song, Jiaming, and Stefano Ermon. "Understanding the Limitations of Variational Mutual Information Estimators." International Conference on Learning Representations. 2019.
> > > > > [3] Shwartz-Ziv, Ravid, and Naftali Tishby. "Opening the black box of deep neural networks via information." arXiv preprint arXiv:1703.00810 (2017).
> > > > > [4] Tishby, Naftali, and Noga Zaslavsky. "Deep learning and the information bottleneck principle." 2015 IEEE Information Theory Workshop (ITW). IEEE, 2015.

---

> > > > > > ### Comment · Reviewer_MBzi · 2021-11-15
> > > > > > **Follow-up questions**
> > > > > >
> > > > > > Thanks for the instant reply. I have some further questions. You said "For MI estimation, feature vectors can be sampled during the SGD training process." What does this sentence mean? Are you saying that SGD is a way to sample the network and gives you different feature vectors at different training epochs? My question is that in a single batch, you are still calculating the mutual information between $Z$ and $F(Z)$ which should be either zero or infinity. This part I am not very sure. I really appreciate it if you can solve my concerns.

---

> > > > > > > ### Author Response · Authors · 2021-11-16
> > > > > > > **Response to Reviewer MBzi (4-2)**
> > > > > > >
> > > > > > > Thank you for the comments.
> > > > > > >
> > > > > > > **A network trained by SGD has randomness because the network is updated depending on the randomly sampled batches for SGD.** (SGD is expressed as "random diffusion" [1].) The network weights change randomly, and the mutual relationship between $F(x)$ and $T(x)$ also changes randomly for the same input $x$. Therefore, $F(x)$ and $T(x)$ are random variables.
> > > > > > >
> > > > > > > **What you mean by a "single batch" is a *realization of random variables*. A realization of $F(x)$ and $T(x)$ is obviously deterministic because it is an observed value.** Mutual information is not defined on the deterministic realizations but the random variables; this is why the sampling process is needed to calculate the MI estimators.
> > > > > > >
> > > > > > > [1] Saxe, Andrew Michael, et al. "On the Information Bottleneck Theory of Deep Learning." International Conference on Learning Representations. 2018.

---

> > > > > > > > ### Comment · Reviewer_MBzi · 2021-11-21
> > > > > > > > **Motivation for minimizing a network from SGD**
> > > > > > > >
> > > > > > > > A network trained from SGD can be considered as random.  If you consider the randomness comes from the sgd training, what's the motivation for maximizing mutual information between $F(x)$ and $T(x)$? In contrastive learning, I think the motivation for maximizing the mutual information between two representations of two views is to minimize the information from nuisance variables in two views.

---

> > > > > > > > > ### Author Response · Authors · 2021-11-22
> > > > > > > > > **Response to Reviewer MBzi (5-1)**
> > > > > > > > >
> > > > > > > > > Thank you for the further question.
> > > > > > > > >
> > > > > > > > > **Q. Why maximize the information between $F(x)$ and $T(x)$, when the randomness comes from SGD unlike multi-viewed batch?**
> > > > > > > > >
> > > > > > > > > **Our motivation of maximizing $\mathcal{I}(F(x);T(x))$ is not for learning information between two views, but for distilling good information from the last layers to the earlier layers [1][2][3][4].** Because $F(x)$ and $T(x)$ are random variables via stochastic training, they have mutual correlation, which the SelfCon loss function can maximize.
> > > > > > > > >
> > > > > > > > > **For the purpose of maximizing $\mathcal{I}(F(x); T(x))$, SelfCon allows the parameters of $T$ to be distilled with good information from $F$.** Note that $F(x)$ denotes the features from the deeper portion of a network, thus having more processed information than the features from earlier layers ($T(x)$) [1]. Our motivation is very similar to VID (see the first paragraph in Sec. 2.2 in [2]) and BYOT (see Abstract in [3]). Also, our loss formulation is similarly driven in [4], where their InfoMax loss has a term that maximizes the MI between the intermediate output and the last output of a single image.
> > > > > > > > >
> > > > > > > > > **Also, borrowing your words, the objective of SelfCon loss is to minimize the nuisance information between feature-level multi-views, i.e., $F(x)$ and $G(x)$ (randomness from SGD-trained parameters), which was that between data-level multi-views, i.e., $F(x)$ and $F(z)$ (randomness from data augmentations) in other multi-viewed contrastive learning.** In SelfCon learning, maximizing the MI between the feature-level multi-views minimizes the nuisance information between $F(x)$ and $G(x)$. Prior works used the data-level multi-views from augmentation randomness and Siamese network (parameter sharing network) to minimize the nuisance information between them. Likewise, we give randomness via SGD, and the trained network parameters (i.e., backbone and sub-network) create multiple views as stochastic representations.
> > > > > > > > >
> > > > > > > > > **References**
> > > > > > > > >
> > > > > > > > > [1] Shwartz-Ziv, Ravid, and Naftali Tishby. "Opening the black box of deep neural networks via information." arXiv preprint arXiv:1703.00810 (2017).
> > > > > > > > > [2] Ahn, Sungsoo, et al. "Variational information distillation for knowledge transfer." Proceedings of the IEEE/CVF Conference on Computer Vision and Pattern Recognition. 2019.
> > > > > > > > > [3] Zhang, Linfeng, et al. "Be your own teacher: Improve the performance of convolutional neural networks via self distillation." Proceedings of the IEEE/CVF International Conference on Computer Vision. 2019.
> > > > > > > > > [4] Hjelm, R. Devon, et al. "Learning deep representations by mutual information estimation and maximization." International Conference on Learning Representations. 2018.

---

> ### Author Response · Authors · 2021-11-12
> **Response to Reviewer MBzi (1-1)**
>
> Thank you for the feedback. We are encouraged that you found our motivation important in a supervised contrastive framework and checked the experimental results and theoretical correctness in detail. The weaknesses of our paper are discussed as follows.
>
> **Q1. The claim "SelfCon loss maximizes the MI between intermediate and last feature" is incorrect.**
>
> **We did not argue that SelfCon loss maximizes the "MI", rather always used with "lower bound", "imply", or "highly related" (please refer to original submission paper).** Minimizing SelfCon loss, although not tightly bounded, implies maximizing the *lower bound of MI* between the intermediate and the last features, and it is also empirically validated in Section 5.2. We claimed and showed that negative SelfCon loss is a lower bound of label conditional MI between the intermediate and the last features.
>
> **Recent advances in contrastive learning prove their success from the connection of contrastive loss to the lower bound of MI [1][2][3][4][5]. Besides, optimizing the loss as a lower bound of some objective function is widely used in the deep learning field [6][7][8].** We acknowledge the empirical success of these optimizations on the lower bound. Contrastive losses are also optimized because they are meaningful as the lower bound of MI between the features or inputs.
>
> **Moreover, we give empirical evidence by finding the clear increasing trend of MI between the intermediate and the last features.** We wanted to see if our SelfCon loss actually leads to the increase of MI between the intermediate and the last representations and thus made an attempt to measure MI with three estimators (MINE, NWJ, and InfoNCE).
>
> **Lastly, in your comment, "negative self-con loss can only increase $\mathcal{I}(F(x),G(x))$ but not $I(F(x), T(x))$" may be incorrect.** Note that $\mathcal{I}(F(x),G(x))$ and $\mathcal{I}(F(x),T(x))$ are both upper bound of negative SelfCon loss (refer to proposition 4.2 and 4.3).  We aimed to theoretically confirm the intuition that increasing negative SelfCon loss, which increases the similarity between $F(x)$ and $G(x)$, correlates to increasing the lower bound of $\mathcal{I}(F(x),G(x))$ and even $\mathcal{I}(F(x),T(x))$ (because $T(x)$ is the intermediate features for $G(x)$).
>
> **Q2. Is mutual information $\mathcal{I}(F(x),T(x))$ either zero or infinity?**
>
> **Yes, if an internal function is deterministic, then the mutual information between the input and output of the function is either infinity or constant.** For a deterministic DNN that maps $X \mapsto T$, the mutual information is degenerate because $\mathcal{I}(X;T)$ is infinite for continuous $X$ (conditional differential entropy is $-\infty$) or a constant for discrete $X$ which is independent on the network’s parameters (equal to $\mathcal{H}(X)$). **(contd. in the next comment)**

---

> > ### Comment · Reviewer_MBzi · 2021-11-12
> > **Q1 is not resolved.**
> >
> > I thank the authors for the thorough response. But Q1 is not resolved. Certainly, maximizing $I(F(x), G(x))$ could help maximizing $I(F(x), T(x))$. However, you cannot do more than maximizing $I(F(x), G(x))$ which is the original information to maximize in contrastive learning. So your point is vacuous on self-con loss as a lower bound of $I(F(x), T(x))$, i.e., your prop3 does not provide more theoretical intuition than the InfoNCE paper.

---

> > > ### Author Response · Authors · 2021-11-14
> > > **Response to Reviewer MBzi (2-1)**
> > >
> > > Thank you for the quick response.
> > >
> > > **Although $T(x)$ is not directly included in SelfCon loss, the SelfCon loss can maximize $\mathcal{I}(F(x);T(x))$ due to the data processing inequality with conditional independence.** $T$ is the parameters of the sharing layers (i.e., $F(x)$ and $G(x)$ are conditionally independent given $T(x)$). Thus, when we maximize $\mathcal{I}(F(x);G(x))$ via SelfCon loss, the intermediate feature $T(x)$ is also changed to capture the important information of both $F(x)$ and $G(x)$. We think it is natural that $T(x)$ with richer information of $F(x)$ and $G(x)$ could help increase $\mathcal{I}(F(x);T(x))$ even though there is a gap between $\mathcal{I}(F(x);G(x))$ and $\mathcal{I}(F(x);T(x))$.
> > >
> > > **Since deep learning is still a black box, many papers make a theory based on the assumptions and propose an algorithm with inspiration from the theory. They expect that optimizing the lower bound, which is easily estimated, increases the exact MI or likelihood, which is difficult to optimize [1][2][3][4][5].** We also wanted to increase $\mathcal{I}(F(x);G(x))$ and $\mathcal{I}(F(x);T(x))$ and made a structure that can be the lower bound due to the difficulty of directly estimating them.
> > >
> > > **In this paper, our main contribution is not the theoretical background due to the limit of our proposition (i.e., lower bound), but (1) proposing a novel supervised contrastive learning SelfCon and (2) confirming that SelfCon actually increases the MI and classification performance according to our intuition.** Figure 2 showed that SelfCon actually increases the information between the intermediate and the last representations ($\mathcal{I}(F(x);T(x))$), and there is a clear trend between MI and performance, despite the lower bound. It is consistent with the numerous works that experimentally prove the theory for optimizing the lower bound of a difficult (to estimate) function, e.g., [6][7][8]. Moreover, we are experimenting with the interpolation between SupCon loss and SelfCon-M loss. We will let you know as soon as the experiment is finished.

---

> > > > ### Comment · Reviewer_MBzi · 2021-11-14
> > > > **Suggestion of rewriting and resubmission**
> > > >
> > > > From my understanding, the paper theoretically motivates the self-con loss. And the theory is not well motivated due to the gap. I have to emphasize again here that I am not satisfied with the motivation NOT because of the gap, but the theory only tells us to maximize $I(F(x);G(x))$ not $I(F(x);T(x))$. For example, $I(F(x);G(x)) \leq \infty$ but you cannot say that minimizing the self-con loss is also maximizing $\infty$.
> > > >
> > > > If the authors believe that their main contribution is to propose self-con loss, I suggest the authors to rewrite the paper to reflect that. And this should be a resubmission. The current version is mainly based on this theoretical motivation. You mentioned it in the abstract and also proved some theories of it. I am not satisfied with the motivation from the theories that self-con loss can maximize $I(F(x);T(x))$.

---

> > > > > ### Author Response · Authors · 2021-11-15
> > > > > **Response to Reviewer MBzi (3-1)**
> > > > >
> > > > > Thank you for the comments.
> > > > >
> > > > > **We understood your question as following: It is agreed that SelfCon loss works as a lower bound of $\mathcal{I}(F(x);G(x))$ but not as a lower bound of $\mathcal{I}(F(x);T(x))$ beacuse there is no direct term about calculating similarity with $T(x)$ in SelfCon loss formulation.** If it is right, we can reformulate the part of SelfCon loss as follows. (Only the numerator is described for the simplicity) $\exp (F(x)\cdot G(x)/\tau) = \exp (f_{(\ell+1):L} (T(x)) \cdot g_{(\ell+1):L} (T(x)) / \tau)$, following the notations in Appendix A. In our experiments (refer to Appendix C), **we used a simple FC layer for $g_{(\ell+1):L}$ and used it as a linear transformation to match the size of $F(x)$ and $G(x)$.** Because it is a simple linear transformation, our loss form can be viewed as directly calculating the similarity between $F(x)$ and $T(x)$.
> > > > >
> > > > > **The above reformulation of SelfCon loss is also highly related to the Deep InfoMax principle, where we get the motivation. Please refer to Eq. 5 in [1].** They also proposed the InfoNCE estimator between the intermediate and the last features, similar to our SelfCon loss. In detail, they used the intermediate feature $C_\psi(x)$ and $E_\psi(x)=f_{\psi}(C_\psi(x))$ to measure the similarity by $D_\omega(C_\psi(x), E_\psi(x))$. Here, they flattened the features for matching the size, while we used the linear transformation. We also maximize the MI by the estimator-like loss, but the structure of a network and the format of the framework is different.
> > > > >
> > > > > **Once again, to explain our intuition about the theory, data processing inequality is the proof technique we used in theory.** We are saying that SelfCon loss CAN increase the lower bound of $\mathcal{I}(F(x);T(x))$ because it changes the intermediate representations as well as the last representations by minimizing SelfCon loss. Think again about the loss form. We maximize the similarity between $F(x)$ and $G(x)$. Here, $G(x)$ is also an output of $T(x)$, so this encourages an encoder to make good representations up until $T(x)$. This is our intuition under the theoretical background.
> > > > >
> > > > > [1] Hjelm, R. Devon, et al. "Learning deep representations by mutual information estimation and maximization." International Conference on Learning Representations. 2018.

---

> > > > > > ### Comment · Reviewer_MBzi · 2021-11-15
> > > > > > **I am against your theoretical motivation but not against the Self-Con loss**
> > > > > >
> > > > > > Thanks for your response. I am sorry if I did not state my claim clearly. My main concern is that you do NOT motivate the self-con loss correctly. It's FINE for you to propose self-con loss and empirically show that it maximizes the mutual information between the intermediate layer and last layer. But it's NOT good to have some proposition (prop 3) which does not correctly motivate the self-con loss.
> > > > > >
> > > > > > For you reformulation, it's unclear why it's the similarity between $F(x)$ and $T(x)$. Any intermediate function could be used here. For example, denote $F(x) = f_1(f_2(x))$ and $G(x) = g_1(g_2(x))$. Then the similarity can also be on $f_2(x)$ and $g_2(x)$. More explanation is needed.

---

> > > > > > > ### Author Response · Authors · 2021-11-16
> > > > > > > **Response to Reviewer MBzi (4-1)**
> > > > > > >
> > > > > > > Thank you for the clarity of the claim and your further questions.
> > > > > > >
> > > > > > > **Our motivation for increasing $\mathcal{I}(F(x);T(x))$ is as follows: $F(x)$ and $T(x)$ are the start and end of the overlapped part between the backbone network ($X \rightarrow T \rightarrow F$) and Markov chain $G \leftrightarrow T \leftrightarrow F$.** With conditional independence, a Markov chain can be defined as both $G \rightarrow T \rightarrow F$ and $F \rightarrow T \rightarrow G$. It is not only a path as an input actually flows (i.e., $G \leftarrow T \rightarrow F$ as in your Q6), but the information flow is defined in both directions (see pg. 34, Section 2.8 in [1]). If you increase the MI of $G$ and $F$ that are both ends of the Markov chain, MI between $T$ and $F$ will also increase (following the information theory). Our theoretical motivation of SelfCon loss is that SelfCon loss increases the correlation between $F(x)$ and $T(x)$ that are included in the backbone network, which is transferred to the downstream task. Moreover, intuition for SelfCon is training a better neural network by making the intermediate feature as informative as the last feature [2][3].
> > > > > > >
> > > > > > > **We focus on the similarity on $T(x)$ because $T$ denotes the sharing layers between $F$ and $G$, and thus, $T(x)$ is the last output for the sharing layers between the backbone and sub-network.** Let us break down the functions for clarity. Let the intermediate feature just before the exit be $T(x)=t_1(t_2(x))$, and assume each backbone and sub-network is break down into $f_1(f_2(T(x)))$ and $g_1(g_2(T(x)))$. As you mentioned, the similarity can also be on $f_2(T(x))$ and $g_2(T(x))$, rather than the very last output of each network. However, $g_2(T(x))$ is not what we really focus on, but the main backbone encoder is what we transplant to the downstream task. Therefore, it is important to see the information between the layers of a backbone network, which aligns with our theoretical motivation. Of course, you can say that the similarity can be on $F(x)$ and $t_2(x)$. However, it is more natural to consider the later (deeper) representations to be meaningful when we have a neural network which is a composite function of layers. Once we see the similarity between $F(x)$ and $T(x)$, it is trivial to see between $F(x)$ and $t_2(x)$.
> > > > > > >
> > > > > > > [1] Cover, Thomas M. Elements of information theory. John Wiley & Sons, 1999.
> > > > > > > [2] Hjelm, R. Devon, et al. "Learning deep representations by mutual information estimation and maximization." International Conference on Learning Representations. 2018.
> > > > > > > [3] Ahn, Sungsoo, et al. "Variational information distillation for knowledge transfer." Proceedings of the IEEE/CVF Conference on Computer Vision and Pattern Recognition. 2019.

---

> ### Author Response · Authors · 2021-11-26
> **Further Discussion**
>
> Dear Reviewer MBzi,
>
> We sincerely appreciate your valuable and active feedbacks for improving our work. We hope we have addressed every question. Please feel free to let us know if there still are concerns not resolved to re-evaluate/re-rate our paper.
>
> Thanks,
> Authors

---

### Official Review · Reviewer_qb4T · 2021-11-03

**Correctness:** 3
**Technical Novelty And Significance:** 3
**Empirical Novelty And Significance:** 3
**Recommendation:** 6
**Confidence:** 4

**Main Review:**

Strengths: The paper propose an interesting idea to employ a multi-exit network for conducting contrastive representation learning without using augmentation techniques to create positive pairs. The method has the potential of addressing the various issues of the multi-viewed contrastive learning such as requiring domain knowledge for augmentation and increased memore/computation. Besides, I also appreciate various ablation studies in the experiments and appendix investigating aspects such as gradient norm, the structure/position/number of subnetworks, different encoder structures, etc.

Below are some of my concerns/questions:
- There are two versions of the self-contrastive loss, the single-view and multi-view version. In the discussion below Eq (2), the paper defines SelfCon-M loss as exactly Eq (1)? I wonder if this is the best design of the loss since with the multi-exit network (different from previous method that only has one network output), when facing multi-view batch, there could be a similar multi-exit extension for the multi-batch setting and should not be just Eq (1)? Also I think even with multi-exit network, the multi-view setting does not just provide redundant information as inductive bias from data augmentation and inductive bias from encouraging similarity within multiple outputs should be different. Thus I am also confused why SelfCon-M, a "generalization" of SelfCon-S with additional information, is mostly worse than SelfCon-S. Is overfitting to each instance the only reason and is there any way to fix it for scenarios where we indeed have prior knowledge for doing augmentation?
- What is the conclusion for using more than one subnetworks (i.e. multi-exit network instead of just 2-exit network)? It seems the current observation is changing the structure will increase the computation a lot but does not really help?
- The theoretical analysis is mainly that the proposed loss provides a lower bound for the MI between intermediate features and last features. I think intuitively this is straightforward since if we maximize MI between the two outputs of the network, and the second output totally depends on the intermediate representation, then the intermediate representation has to be closely correlated with the last output such that the second head also contains similar information as the first head. That being said, I think it is still not clear that why such a theoretical result can explain an improved classification performance or better representation learning. A degrade case is, if we put an identity mapping between the intermediate layer and the last output of the backbone, the MI is also very large but it will not help for the downstream task.
- There are many recent advances in representation learning. For example, BYOL has received a lot of attention for a SOTA performance without using negative pairs. Any discussion and comparison to these methods? Also in section 2.1, BYOL is introduced following "with negative pairs", which is not accurate?
- At the beginning of sec 2.2, "Mutual information is a measure to quantify the amount of information held in a random variable" - I think what you are describing is entropy not MI.  MI is a measure of the mutual dependence between two random variables.
- Overall, I think the method derivation/description (section 3) is hard to follow and not clear enough. An algorithm box may also be helpful. I think the presentation can be significantly improved for better readability. For example, there is no appropriate explanation for the key equations like eq 2 and a formal definition of SelfCon-M. It takes me some time to parse it.

---------------After rebuttal-----------------------  Thanks for the detailed response to my feedback. I appreciate the efforts on updating the paper and I think some of my concerns have been addressed. I thus raised my score to weak accept.

**Summary Of The Paper:**

The paper proposed a contrastive learning framework to remove the requirement for additional data augmentation techniques for creating positive pairs, by leveraging the idea of multi-exit network and conduct "self-contrasts" among multiple outputs of the model. Theoretical analysis on the designed loss shows that the proposed SelfCon loss is a lower bound of MI between the intermediate and last feature. Extensive experiments and ablation studies demonstrate that the proposed method has better performance than than the supervised contrastive baseline.

**Summary Of The Review:**

In summary, I think this paper propose an interesting idea with some potential of improving contrastive learning by removing the requirement of data augmentation. Nevertheless, after reading the paper, I also have a number of concerns/questions on some statements, method derivation, theoretical analysis and empirical evaluation. Overall, I think the presentation can be improved to make it easier to follow.

---

> ### Author Response · Authors · 2021-11-11
> **Response to Reviewer qb4T (1-3)**
>
> **Q7. Wrong definition for the mutual information.**
>
> **Yes, we made the wrong definition for the mutual information.** The correct definition is that “mutual information is a measure to quantify the amount of information held in a random variable **about the other variable.“** We missed the last few words. Thank you for pointing it out, and we have revised the corresponding sentence. By the way, entropy can be interpreted as “a measure to quantify the amount of uncertainty held in a random variable”. Mutual information is the difference between entropy and conditional entropy, i.e., $I(X;Y)=H(X)-H(X|Y)$, measuring the reduced uncertainty of $X$ from knowing $Y$, which is the amount of information about $X$ held in $Y$.
>
> **Q8. The equations in Section 3 is not clear enough.**
>
> **We modified the equations in the revised paper (refer to (R1) in the overall response) as described below.** Thank you again for pointing out Section 3, which should clearly explain our main method and suggest a specific solution.
>
> **Eq. 1:** We changed the notation of a positive set $P$ as $P_{i*}=$ {$p \in I \setminus $ {$i$} $| y_p=y_i$} that excludes the index of anchor sample ($i$). Besides, we defined the negative set $N$ instead of $J$ for better readability. We also removed identity function $\mathbb{1}[F(x_i) \neq F(x_p)]$ because we do not need indicator function by modifying the notation $P$.
>
> **Eq. 2:** We modified the notations $\omega', \omega''$ to $\omega_1, \omega_2$, respectively. Different from Eq. 1, we included the sample $i$ into $P_{i*}$ because positive set $P_{i*}$ should include the anchor sample $x_i$ when $\omega \neq \omega_*$. It is complicated to express in the equation, therefore, we added the detailed explanation for the positive set (i.e., $P_{i*} \leftarrow P_{i*} \setminus$ {$i$}) to the paragraph below Eq. 2.
>
> **References**
> [1] Khosla, Prannay, et al. "Supervised contrastive learning." arXiv preprint arXiv:2004.11362 (2020).
> [2] Cubuk, Ekin D., et al. "Randaugment: Practical automated data augmentation with a reduced search space." Proceedings of the IEEE/CVF Conference on Computer Vision and Pattern Recognition Workshops. 2020.
> [3] Tian, Yonglong, et al. "What makes for good views for contrastive learning?." arXiv preprint arXiv:2005.10243 (2020).
> [4] Grill, Jean-Bastien, et al. "Bootstrap your own latent: A new approach to self-supervised learning." arXiv preprint arXiv:2006.07733 (2020).
> [5] He, Kaiming, et al. "Momentum contrast for unsupervised visual representation learning." Proceedings of the IEEE/CVF Conference on Computer Vision and Pattern Recognition. 2020.
> [6] Caron, Mathilde, et al. "Unsupervised learning of visual features by contrasting cluster assignments." arXiv preprint arXiv:2006.09882 (2020).
> [7] Chen, Xinlei, and Kaiming He. "Exploring simple siamese representation learning." Proceedings of the IEEE/CVF Conference on Computer Vision and Pattern Recognition. 2021.

---

> ### Author Response · Authors · 2021-11-11
> **Response to Reviewer qb4T (1-2)**
>
> **Q4.  How does the theoretical analysis explain an improved classification performance?**
>
> **Our main theoretical analysis is proposition 4.3, which implies high correlation between MI and improved classification performance in terms of the following information: (1) Interaction information (i.e., $\mathcal{I}(F(x);c | T(x)) - \mathcal{I}(F(x);c)$) and (2) MI estimation between the intermediate and the last features (i.e., $\mathcal{I}(F(x);T(x))$).**
>
> **(1) Interaction information: Minimizing SelfCon loss is highly correlated to maximizing the interaction information, which makes the intermediate feature enhance the correlation between the last feature and labels.** In other words, optimization for SelfCon loss maximizes the lower bound of information between the last feature and the ground truths that only intermediate features can explain. However, there are some concerns as follows: (a) we can not strictly guarantee the lower bound of interaction information term (refer to the last paragraph in Appendix A.3) and (b) It is hard to estimate the interaction information.
>
> **(2) MI estimation between the intermediate and the last features: Our intuition is that our SelfCon loss makes the intermediate feature have more meaningful information about the label, improving the classification performance.** $T(x)$ is distilled from $F(x)$ with richer class-related information, and consequently, earlier layers are trained to produce better representation. We observed the clear increasing trend of $\mathcal{I}(F(x);T(x))$ with three MI estimators in Figure 2. Moreover, we observed an increase of the MI between the intermediate feature and the label in Appendix E.
>
> **Q5. Then, why identity mapping between the intermediate and the last output of the backbone does not help the performance?**
>
> **Since identity function does not make different class representations farther, the identity function case is a too loose bound in terms of SelfCon loss.** We used the variational inference to prove SelfCon loss is the lower bound of MI; thus, we assumed a probabilistic model like Eq. 8. When the anchor feature is similar to the negative pairs (i.e., different class representations, $z_{2:(2K-1)}$), this model becomes a distribution with random mapping, and SelfCon loss can not be optimized. Therefore, representations of other classes should be farther to decrease the gap for MI, and it lets the encoder use every parameter for better representation ability. This is why making half of the parameters useless (i.e., identity function between $F$ and $T$) is not optimal. After all, SelfCon loss has improved performance because it aims to increase the MI between $F$ and $G$ while increasing the distinction between different class representations.
>
> **Q6. Is there any discussion to recent representation learning algorithms (e.g., BYOL [4])?**
>
> **Before the discussion of recent approaches, in Section 2.1, we did not state that BYOL is the algorithm "with negative pairs".** You probably have missed "momentum encoders" in front of "MoCo, BYOL". We stated the algorithms with negative pairs as "with negative pairs (SimCLR, MoCo)".
>
> **We do not consider the recent advances in representation learning (e.g., BYOL [4]) in this paper because they are "unsupervised" algorithm, and our SelfCon learning is a "supervised" contrastive algorithm.** As you already know, BYOL is "unsupervised" representation learning with "multi-view", "momentum encoder", and "distillation", without "negative pairs". To our best knowledge, there is no work on supervised contrastive learning without negative pairs.
>
> **Therefore, BYOL cannot be compared as a baseline for SelfCon learning, and too many considerations are needed. We believe it is beyond the scope of our paper.** For example, applying the BYOL technique to SupCon requires choosing how to distill multiple positive pairs, such as loss summation of all possible combinations or distillation with ensemble features. Besides, bringing BYOL to our SelfCon makes other questions as well as the issues mentioned above: (1) how to use a momentum encoder without multi-view, (2) if we use the same architecture between the backbone and sub-network, we can apply momentum encoder, but then what is the difference from SupCon-S with BYOL technique? (maybe the existence of sharing and non-sharing parameters). Although extending the supervised contrastive framework to using BYOL [4], MoCo [5], SwAV [6], or Simsiam [7] is worth investigating, and it is beyond the scope of this paper.

---

> ### Author Response · Authors · 2021-11-11
> **Response to Reviewer qb4T (1-1)**
>
> We are encouraged that you pointed out the novelty of SelfCon learning as the strength of our paper. Also, we thank you for listing the weaknesses of our paper in detail, especially misleading parts in the equations. We will explain how we fixed them in the paper.
>
> **Q1. Why do you define SelfCon-M loss as SupCon loss (Eq. 1)?**
>
> **No, SelfCon-M loss is not the same as Eq. 1.** The discussion below Eq. 2,  "We define SelfCon with multi-viewed batch (SelfCon-M) loss, if $I, J =$ {$1, \dots, 2B$} as SupCon loss (Eq. 1)", means that SelfCon-M uses the multi-viewed batch just like SupCon [1]. In Figure 1, SelfCon-M cannot be the same as SupCon because SelfCon-M has a multi-exit framework and can use positive and negative pairs with the features from the sub-network. We modified Eq. 2 (refer to Q7 and the revised paper) and removed the above sentence to avoid confusion.
>
> **Q2. Why SelfCon-M is mostly worse than SelfCon-S?**
>
> **We consider that overfitting to samples is the main reason for the lower test accuracy of SelfCon-M because the only difference between SelfCon-M and SelfCon-S is whether to use the multi-viewed batch or single-viewed batch.** Multi-view from the augmented image makes the encoder amplify the memorization of data and overfitting to each instance. In contrast, single-view does not produce a positive pair from the augmentation. This trend is not only observed with SelfCon-M and SelfCon-S but also with SupCon and SupCon-S. In most experiments in Table 1 and Table 2, SupCon-S outperformed SupCon. The experimental results of Figure 3, Figure 4, and Table 4 supported our claim.
>
> **However, SelfCon-M is not always worse than SelfCon-S.** For 64-batch experiments in Table 4, where the small batch size increases the regularization effect, SelfCon-M showed +1.8% increase than SelfCon-S. Moreover, SelfCon-M is better than all other methods for the ImageNet dataset (refer to Appendix G in the revision). Multi-views help memorize each instance; thus, multi-view is advantageous over single-view in small batch size or in the large-scale dataset where it is difficult to train with a large number of samples.
>
> **Also, we are running the experiment with a strong augmentation strategy (e.g., RandAugment [2]) and weak augmentation (e.g., use only cropping and flipping).** By confirming how the strength of data augmentation affects the performance of multi-view, it seems to support our claim of overfitting further. We will let you know as soon as the experiments are finished.
>
> **We would like to note that in SupCon paper [1], they tried diverse choices of augmentation policy and found that Stacked RandAugment [3] fitted the best with SupCon in ImageNet ResNet-101 experiments.** However, performance variance was very large depending on the numerous augmentation parameters, encoder architecture, and the combinations of augmentation policies (refer to Table 9 and Figure 7 in [1]). For example, Stacked RandAugment with ResNet-50 performed worse than other policies due to the lower capacity of the ResNet-50 model.
>
> **Therefore, we suppose that even if there is the best scenario for augmentation, it takes an enormous amount of time and cost to find the appropriate augmentation.** This is one of our main concerns of the multi-view introduced in Section 1.
>
> **Q3. What is the conclusion for using more than one sub-network?**
>
> **Performance improvement from SelfCon with multiple sub-networks is not as significant as its higher computational cost (refer to Table 4 in Appendix C).** Therefore, we simply used a single sub-network as we described in the last paragraph in Section 3.

---

> > ### Author Response · Authors · 2021-11-16
> > **Response to Reviewer qb4T (2-1)**
> >
> > **Q2 (updated response). Why SelfCon-M is mostly worse than SelfCon-S?**
> >
> > As you have mentioned, we could think of scenarios where we have prior knowledge for augmentation. Also, there could be the opposite scenario where we do not know an optimal policy and just use simple augmentations. **Thus, we experimented with an Optimal(Strong) augmentation strategy and a Simple(Weak) augmentation strategy to compare the performances of SupCon and SelfCon-S under these scenarios.**
> >
> > **As seen in Table 16 in Appendix J, when we applied strong augmentations (i.e., RandAugment [1]), SupCon outperformed SelfCon-S, but for standard and simple augmentations, SelfCon-S outperformed SupCon.** There would be an optimal policy also for SelfCon learning but, finding an optimal policy, such as with RandAugment or AutoAugment, is not a trivial process and needs a lot of computational cost. SelfCon learning can relieve this concern and is more robust to the simple augmentation (i.e., weak augmentation policy).
> >
> > [1] Cubuk, Ekin D., et al. "Randaugment: Practical automated data augmentation with a reduced search space." Proceedings of the IEEE/CVF Conference on Computer Vision and Pattern Recognition Workshops. 2020.

---

> ### Author Response · Authors · 2021-11-26
> **Further Discussion**
>
> Dear Reviewer qb4T,
>
> We sincerely appreciate your valuable comments for improving our work. We have put in efforts in the revision by modifying the words and equations to clarify our motivation, especially on the equation of SelfCon loss and the connection between SelfCon and the improved classification performance. We also experimented with an optimal/simple augmentation strategy to compare the performance under the scenarios with prior knowledge for augmentation.
>
> We would like to send you a gentle reminder that the rolling discussion is now at the end of its period. We hope we have addressed your concerns, and we look forward to knowing if any of our response is still in question.
>
> Thanks,
> Authors

---

### Official Review · Reviewer_ZiPE · 2021-11-04

**Correctness:** 3
**Technical Novelty And Significance:** 2
**Empirical Novelty And Significance:** 2
**Recommendation:** 5
**Confidence:** 4

**Main Review:**

The paper is mostly well-written and nicely structured. The motivation is clearly outlined, and the discussion of related work is extensive. I liked how the contributions are clearly linked to sections in §1, and the use of color to highlight important parts in the equations throughout the text. A major strength is the possibility of *supervised* contrastive learning without additional augmentations (and I think this could be highlighted more prominently as a strength).

A major shortcoming of the paper is a lack of full ImageNet results, and the small batch sizes considered in the paper. I should note, while the numerous analyses added to the paper are nice, a few well-executed experiments including the “gold standard” models for self supervised learning would have made the paper much stronger (and a clear accept).

I will discuss some additional weaknesses worth addressing / commenting on below.

### Weaknesses

I have the following concerns about the experiments:

* The batch size in the supervised contrastive learning paper was 6144 --- how do you justify the must smaller considered batch size in the paper? How would you rule out that a larger batch size could diminish the gains from your method?
* The supervised contrastive papers outlines a technique for training with batch size 256 (I think effectively 512 due to the augmentation) on a 8 GPU system 79.1% for full ImageNet. If compute was a limiting factor, could the authors comment why they didnt use this setup? There is also a range of methods (like MoCo and related methods) with momentum buffers or other methods that make it possible train CL models on ImageNet scale on 8 GPU setups.
* Any reason why Table 3 does not include measurements for full-scale images, i.e., ImageNet-100? For practitioners, this would be by far the most relevant result.
* Fair comparisons: Why does the result for SupervisedContrastive learning in Table 4 gets worse with a larger batch size? Might this be due to suboptimally tuned hyperparameters (especially the learning rate)?

Some weaknesses about the interpretation of results:

* The paper contains a lot of causal statements about a link between quality of MI estimation and the performance in downstream classification performance. Yet, the only evidence is Figure 2, which is a correlation analysis with a few point estimates. What common confounders could alternatively explain this result, and why did you think that there was no additional need to run more control experiment? What happens e.g. when different hyperparameter setups for Self/SupCon are compared; does the correlation between MI and Test Acc hold?
* Table 2: Since this is a new dataset (as far as I am aware of), could you minimally provide mean and standard deviations across three seeds? I find it hard to judge the variance and what constitutes progress on this task without seeing CIs; also, were hyperparameters selected fairly between the baseline and the tested CL models? To make this result more stronger, what about evaluating the existing full imagenet trained checkpoints of SupCon and including this as a comparison?

The method section 3 needs work (but please correct me if I am misinterpreting anything here):

* In Eq. 1, could you verify that $1[F(x_i) \neq F(x_p)]$ is equal to 1 for all cases, i.e. you included this to make the equation consistent with Eq 2? If this is the case, what was the motivation? Why did you not just use $\omega \neq \omega’$ in Eq 2? Could you please clarify if I am missing something? If I am correct that something is off, I would appreciate it if you could let me know what the revised equations would look like. The same concern applies to Eq. 2.
* “We dropped the dividing term of sums for brevity” -> Could you clarify and re-write this sentence? I do not get the reference.
* The notation is incomplete; it should be made clear that $P$ depends on $i$. Suggestions: Call it $P_i$ or $P(i)$. Also, $(i, p) \in I \times P$ does not work because of this, it should be $i \in I, p \in P_i$ (for example). The same applies to $J$, which contains all images except the anchor. Write this explicitly: $J \setminus
 \{i\}.

A few additional minor points, not sorted by priority:

* Use of barplots: Figure 2 and Figure 3 show barplots with truncated y axis, which is a bad practise. Much better and also more informative is an interaction plot (i.e., just plot a dot, along with an error bar). This is a major concern, because strong results are concluded from these plots.
* In Figure 2, the different points for NWJ, MINE and InfoNCE should not be connected --- the x axis is categorical, not continuous. Also, a much better of getting the point across would be a scatterplot between MI and Test Accuracy.
* Figure 6: Do not use the jet colormap. It is widely recognized that it distorts the visual perception of results. Please use viridis (as in Figure 5), or a similar variant with less issues. Especially for the dog images, the results conveyed by the jet map are misleading.
* Table 1, column “single view”: I would e.g. use checkmarks ✓ instead of the current O/X --- I would rather associate the “X” with “yes” then the “O”. So I suggest to disambiguate this.
* In the text, you wrote $P = i + B \mod 2B$ --- while this might be technically correct, I recommend to write $P = (i + B) \mod 2B$, especially readers with a programming background might be confused on a quick pass through the paper since operator precedence in e.g. Python would interpret the original statement as  $P = i + (B \mod 2B)$
* I find the distinction between “single view” and “multi view” a bit misleading / counter-intuitive. Could you comment whether you feel this is a standard term used in popular related work?
* It would be nice to actually include the image sizes in Table 3, for example after a linebreak in the first column (e.g. “CIFAR-100<linebreak>32x32$)
* Table 3:
   * It would be more interesting to show time / step, vs. time / epoch; otherwise you need to adapt for the fact that higher batch sizes mean “shorter epochs” due to less batches / epoch.
   * Why does the cost per training step go up? Is there an issue with data-loading in the implementation?

**Summary Of The Paper:**

The authors propose an improved algorithm for supervised contrastive learning, which they term “self-contrastive learning”. Instead of only contrasting representations at one point in the network, self-contrasting uses readouts at multiple stages in the network (remapped into an appropriate feature space by an additional readout network). This makes it possible to propose two variants: A “multi-batch” variant that first augments the images, and then contrasts views between samples of the same class, both augmented and non-augmented, and a “single-batch” variant that only contrasts between samples of the same class. Improvements are shown for CIFAR-10,100, TinyImageNet, as well as a smaller version of ImageNet with 100 classes, referenced as ImageNet-100.

**Summary Of The Review:**

The paper proposes an interesting addition to supervised contrastive learning, which removes the lack for data augmentations during training and yields improved results. The results look compelling to me, although I generally have the impression that they were executed under a lack of computational ressources (seen, e.g. by relatively small batch sizes although large batch sizes are known to improve performance, lack of multiple seeds for some experiments, etc.), and hence I am worried about the significance of them (how well baselines were tuned, etc.).

That being said, I am willing to improve the assigned score the paper. To consider an accept, the authors need to minimally show a proof of concept result on ImageNet (for example in a very controlled setup, reproducing the SubContrast result, adding their method and showing an improvement with no special additional tuning, to make the comparison as fair as possible.

As I see it, the method already works on full-scale images. What is the blocker for running a full scale experiment? Are you confident that the improvements hold on full scale ImageNet training? From the data in the paper right now, it is not clear that the improvement will hold in the regime simCLR-like loss functions are typically evaluated on (large batch sizes, ResNet50 models, full ImageNet).

Without full ImageNet results, the paper in my opinion does not meet the bar for ICLR, as the one crucial experiment that would contribute to adoption of the method is missing.

---

> ### Author Response · Authors · 2021-11-10
> **Response to Reviewer ZiPE (1-4)**
>
> **References**
> [1] Khosla, Prannay, et al. “Supervised contrastive learning.” arXiv preprint arXiv:2004.11362 (2020).
> [2] Kaiming He, Haoqi Fan, Yuxin Wu, Saining Xie, and Ross Girshick. Momentum contrast for unsupervised visual representation learning. arXiv preprint arXiv:1911.05722, 2019.
> [3] You, Yang, Igor Gitman, and Boris Ginsburg. “Large batch training of convolutional networks.” arXiv preprint arXiv:1708.03888 (2017).
> [4] Hjelm, R. Devon, et al. “Learning deep representations by mutual information estimation and maximization.” International Conference on Learning Representations. 2018.
> [5] Oord, Aaron van den, Yazhe Li, and Oriol Vinyals. “Representation learning with contrastive predictive coding.” arXiv preprint arXiv:1807.03748 (2018).
> [6] Tian, Yonglong, Dilip Krishnan, and Phillip Isola. “Contrastive multiview coding.” Computer Vision–ECCV 2020: 16th European Conference, Glasgow, UK, August 23–28, 2020, Proceedings, Part XI 16. Springer International Publishing, 2020.
> [7] Ermolov, Aleksandr, et al. “Whitening for self-supervised representation learning.” International Conference on Machine Learning. PMLR, 2021.
> [8] Jiang, Ziyu, et al. “Self-Damaging Contrastive Learning.” arXiv preprint arXiv:2106.02990 (2021).
> [9] Wu, Haiping, and Xiaolong Wang. “Contrastive Learning of Image Representations with Cross-Video Cycle-Consistency.” arXiv preprint arXiv:2105.06463 (2021).

---

> ### Author Response · Authors · 2021-11-10
> **Response to Reviewer ZiPE (1-3)**
>
> **Q7. You should correct Eq. 1 and Eq. 2 in Section 3.**
>
> **We guess the definition for positive set $P$ made you misleading.** We originally defined the positive set $P$ as the samples with the same class label, including the anchor itself. We wanted to use the consistent notation in Eq. 1 and Eq. 2, but this might have confused you.
>
> **Therefore, we modified the equations in the revised paper (refer to (R1) in the overall response) as described below.**
>
> **Eq. 1:** We modified the definition of a positive set $P$ and defined the negative set $N$ instead of $J$ for the sake of intuition. In detail, we defined the positive set $P_{i*}=$ {$p \in I  \setminus $ {$i$} $ | y_p=y_i$} that excludes the anchor index $i$. Besides, we removed $\mathbb{1}[F(x_i) \neq F(x_p)]$ because placing the indicator in log term is wrong, and we do not need this function.
>
> **Eq. 2:** First, we changed the term $\omega’, \omega’’$ to $\omega_1, \omega_2$, respectively. When $\omega \neq \omega_*$, positive set $P_{i*}$ should include the anchor sample $x_i$. Therefore, different from Eq. 1, we included the sample $i$ into $P_{i*}$, and added the detailed explanation for the positive set (i.e., $P_{i*} \leftarrow P_{i*} \setminus$ {$i$}) to the paragraph below Eq. 2.
>
> **Q8. What is the meaning of "dropped the dividing term of sums for brevity"?**
>
> **We meant that we removed the diving terms (e.g., the cardinality of the set of indices) in front of the sums in Eq. 1 and 2.** For example, in Eq. 1, in front of the sum $\sum_{i,p_1}$ there should be the dividing term $\frac{1}{|I||P_{i1}|}$ to rescale the loss by batch sizes. All other sums in denominators hold the same issue, but we omitted the dividing terms due to the lack of margin and brevity.
>
> **Q9. Your notation for the positive set $P$ is incomplete.**
>
> **We missed the rigorous definition of the positive set $P$ and, as you pointed out, set $P$ depends on $i$, so it should be $P_i$.** Moreover, we wanted to separate the positive index $p_1$ and $p_2$ because $\omega_1$ and $\omega_2$ can be different functions. Also, we modified set $J$ to $N_i$ to explicitly indicate the negative sample set. Therefore, it is defined as $N_i=$ {$n\in I | y_n \neq y_i$}, so it never includes the index $i$.
>
> **Q10. Correct the additional minor points.**
>
> **Thank you for your thorough and extensive feedback on our paper.** We will revise the paper as soon as possible, and as we update the papers, we will let you know about the modified parts. Especially, Table 3, Figure 6&7, and Table 1&6 are already updated. A few reminders are noted as below:
>
> - **The truncated y-axis in Figure 3:** We used the truncated y-axis because there is a difference of one decimal place for some results of the training accuracy (e.g., CIFAR-10 benchmark). However, it does not mean that our argument is not sufficient, as we already confirmed the meaningful confidence intervals in Table 1.
> - **Connected line in Figure 2:** Thank you for pointing out the awkward part in Figure 2. We originally removed the connected line, but it was hard to recognize the same estimator values. We hope you understand that lines do not mean continuous variables but rather show the clear trend of each estimator.
> - **Use viridis colormap:** Thanks to your comment, we changed to viridis colormap. Please refer to Figure 6&7 in the revised paper.
> - **Use checkmarks in Table 1&6:** We revised O/X as checkmarks in Table 1&6 to mitigate the ambiguity.
> - **Notation for positive set $P$:** We fixed $P=$ {$i+B \bmod 2B$} to $P=$ {$(i+B) \bmod 2B$} for better readability for programming-based readers.
> - **Use of the terms 'single-view' and 'multi-view':** Our paper is highly related to SupCon paper, and they used the term 'multiviewed batch' to indicate the set of augmented samples in a batch (see Section 3.2 in [1]). For an earlier paper, CMC [6] used the term 'multi-view' in a similar context. 'Single-view' is our novel term to make a clear comparison to the 'multi-view'. 'Single-view' implies not using an augmented sample but just using only a single image. Most contrastive learning frameworks rely on the multi-viewed setting; thus, 'single-view' is considered a novel term. In contrast, our main contribution is removing the need for 'multi-view', which is a novel approach.
> - **Comments for Table 3:** In Table 3, as we already have mentioned in Q3, we included the image sizes for each dataset and time / step measures.

---

> ### Author Response · Authors · 2021-11-10
> **Response to Reviewer ZiPE (1-2)**
>
> **Q5. The paper has few grounds for the correlation between MI and performance.**
>
> **With proposition 4.3., we gave two clues for the correlation between MI and performance: (1) Interaction information (i.e., $\mathcal{I}(F(x);c | T(x)) - \mathcal{I}(F(x);c)$) and (2) MI estimation between the intermediate and the last features (i.e., $\mathcal{I}(F(x);T(x))$).**
>
> **(1) Interaction information: We showed that minimizing SelfCon loss is highly correlated to maximizing interaction information, making the intermediate feature enhance the correlation between the last feature and the label.** We believe that this is theoretical evidence for the higher classification performance of SelfCon than SupCon. However, it is hard to estimate the interaction information. Besides, as we mentioned in Appendix A, we cannot strictly guarantee the bound of interaction information.
>
> **(2) MI estimation between the intermediate and the last features: Our intuition is that our SelfCon loss gives the intermediate feature more meaningful information about the label, improving the classification performance.** $T(x)$ is distilled from $F(x)$ with richer class-related information, and consequently, earlier layers are trained to produce better representation. We give empirical evidence with three estimators by finding the clear increasing trend of MI between the intermediate and last features. Many prior works in contrastive learning literature argued that contrastive learning increases the lower bound of MI between the features or inputs [4][5], but few of them empirically showed the ground for their claims. In this work, we would like to see if our SelfCon loss leads to the increase of MI between the intermediate and the last representations and thus attempt to measure MI with three estimators.
>
> **We also added a detailed explanation for the connection between SelfCon loss and performance improvement.** Because we used variational inference and introduced the probabilistic model as Eq. 8, representations of other classes should be farther to decrease the gap between SelfCon loss and MI. After all, SelfCon loss has improved performance because it aims to increase the above information while increasing the distinction between different class representations. Please refer to Appendix A.4 in the revised paper.
>
> **To further show the correlation between MI and test accuracy, we are experimenting with the interpolation between SupCon loss and SelfCon-M loss (SupCon loss is a special case of SelfCon-M loss).** The current formulation of Eq. 1 and Eq. 2 cannot make the exact interpolation between SupCon and SelfCon-M because SelfCon-M loss should have negative pairs from different levels of a network (i.e., backbone and sub-network), but SupCon loss cannot produce those. Therefore, we should modify the SelfCon-M loss to a supervised version of Eq. 42. In the supervised version of Eq. 42, we can break down the SupCon loss term and SelfCon-like loss term, with an interpolating parameter $\alpha$. If $\alpha=0$, it is equivalent to SupCon loss and if $\alpha=1$, it is almost equivalent to SelfCon-M loss. We will measure the correlation between the classification performance and MI via controlling the $\alpha$ hyperparameter as soon as possible.
>
> **Q6. Can you provide ImageNet-100 results with three random seeds?**
>
> **It is not sufficient to run three seeds with our computational resources, but we are very convinced that SelfCon-S outperforms other methods because of the significant increase (i.e., +1.5% for SelfCon-S vs. SupCon and +3.0% for SelfCon-S-Ensemble vs. SupCon).** Moreover, for a thorough comparison, we also verified SelfCon learning with the ResNet-50 backbone network. Referring to Table 2, there were similar increases from ResNet-18 to ResNet-50. We are certain that SelfCon methods outperformed SupCon methods in the same control setting since we used the same hyperparameter rules between every method (see Appendix B).
>
> **"Contrastive Multiview Coding" [6] firstly proposed ImageNet-100 benchmarks, and many recent papers have used [7][8][9].** This dataset is a subset of ImageNet, having 100 classes out of the original 1000 classes and the same full images for each class as ImageNet. Usually, ImageNet results have small CIs, and many works report the performance with only one seed (as [1] or [6] did), and even the performance improvement is marginal.

---

> > ### Author Response · Authors · 2021-11-16
> > **Response to Reviewer ZiPE (2-3)**
> >
> > **Q5 (updated response). The paper has few grounds for the correlation between MI and performance.**
> >
> > **To clearly show the correlation between the MI and test accuracy, we updated the correlation analysis of Figure 2.** As we have mentioned, we modified the Eq. 42 into a supervised version and tuned the hyperparameter $\alpha$ that controls the contribution of SelfCon loss. The updated results are found in Appendix K. We observed a clear increasing trend of both MI and test accuracy as the contribution of SelfCon gets larger.

---

> > > ### Comment · Reviewer_ZiPE · 2021-11-28
> > > **Re: The paper has few grounds for the correlation between MI and performance.**
> > >
> > > Thanks, I appreciate the effort to perform a controlled study.
> > >
> > > However, equation (44) is not interpolating between the two losses, this would be the case with an additional $1 - \alpha$ factor in front of the supervised loss. Could you comment if this is a potential issue when interpreting the results?
> > >
> > > > To clearly show the correlation between the mutual information and test accuracy, we experimented
> > > with the interpolation between SupCon loss and SelfCon-M loss (SupCon loss is a special case of
> > > SelfCon-M loss).
> > >
> > > Where do I see the accuracies for the experiments in Table 17? Otherwise, what is the conclusion from the experiment?

---

> > > > ### Author Response · Authors · 2021-11-28
> > > > **Response to Reviewer ZiPE (3-2)**
> > > >
> > > > Thanks again for further questions.
> > > >
> > > > To clearly state the Eq. 44, SelfCon-M loss in Eq. 2 can be decomposed as (SelfCon-M loss) = (SelfCon-M loss w/ anchor $F(x)$) + (SelfCon-M loss w/ anchor $G(x)$). Similarly, Eq. 44 states that **(SelfCon-M\* loss) = (SupCon loss) + $\alpha\cdot$  (SelfCon-M loss w/ anchor $G(x)$)**. If $\alpha$ is 1, SelfCon-M* becomes similar to SelfCon-M loss, since SupCon loss is computed only with anchor $F(x)$. **Therefore, Eq. 44 is approximately an interpolation between SupCon and SelfCon-M.** If there is $1-\alpha$ in front of SupCon loss, $\alpha=1$ reduces to only SelfCon-M w/ anchor $G(x)$, which is only the half of what we need.
> > > >
> > > > **By Appendix J, we wanted to show the correlation between the improved performance (or mutual information) and the main part of SelfCon loss, a contrastive task between the backbone and sub-network.** Figure 8 and Table 17 (is just detailed numbers of Figure 8) were the experimental results showing that the performance and MI increased as the influence of the contrastive task between $F(x)$ and $G(x)$ increased.
> > > >
> > > > Thank you for pointing out the detailed accuracy numbers of Figure 8 and we added to Table 17 as following numbers:
> > > >
> > > > | SupCon |  $\text{SelfCon-M*}_{0.025}$ |   $\text{SelfCon-M*}_{0.05}$ |  $\text{SelfCon-M*}_{0.075}$ |  $\text{SelfCon-M*}_{0.1}$  |  $\text{SelfCon-M*}_{0.15}$  | SelfCon-M  |
> > > > |:------:|:------:|:------:|:------:|:------:|:------:|:------:|
> > > > | 73.0 | 73.3 | 73.5 | 73.9 | 74.2 | 74.6 | 74.9 |

---

> ### Author Response · Authors · 2021-11-12
> **Response to Reviewer ZiPE (1-1)**
>
> We thank you for your thoughtful feedback. We are encouraged that you found our idea, the possibility of supervised contrastive learning without additional augmentations, insightful, and our ablation study impressive. The weaknesses of our paper are discussed as follows.
>
> **Q1. Why don't you use 6144 batch size as in SupCon paper [1]?**
>
> **Actually, SupCon [1] did not use the batch size of 6144, and they used 1024 for CIFAR-10 and CIFAR-100.** The batch size of 6144 in Section 4.5 of SupCon [1] was only for the ImageNet dataset. Refer to Table 2 of [1] where it is stated that "Note that the CIFAR-10 and CIFAR-100 results are from our PyTorch implementation and ImageNet from our TensorFlow implementation." In [their official Pytorch Github](https://github.com/HobbitLong/SupContrast), the authors of [1] mentioned that they used the batch size of 1024 with 8 GPUs.
>
> **For a fair comparison, we also experimented SelfCon with 8 GPUs and a batch size of 1024.** Note that all other hyperparameters (e.g., learning rate, temperature scale) are the same with [1] as well.
>
> **Q2. Why don't you experiment on full-scale ImageNet?**
>
> **As we mentioned in the overall response, full-scale ImageNet results are presented in Appendix G with ResNet-18 and batch size 1024.** Further experiments with a larger batch size (e.g., size of 2048) and longer epochs are undergoing. We also checked the paper [1] used 256 batch size on 8 GPU system using memory bank (K=8192) [2]. The authors of [1] indeed experimented with both the memory bank and momentum encoder technique. We have checked the experiment setting directly with the authors through e-mail. Since we did NOT want to depend on other techniques, we experimented on a batch size of 1024 without a memory bank for full-scale ImageNet.
>
> **Q3. How about including computational cost for ImageNet-100 in Table 3?**
>
> **We reported computational cost with ImageNet-100 images in the revised Table 3, which is measured on ResNet-18.** We appreciate that you found the missing details in Table 3. The originally given ResNet-50 results are listed in Appendix H without ImageNet-100 (ResNet-50 with batch size over 256 exceeded the GPU limit).
>
> **We also would like to highlight that we additionally reported time / step and included the image sizes, following your comments on minor points.** Obviously, time / step increases when the batch size gets larger since large batch size means large similarity matrix ($\mathcal{O}(B^2)$), thus requiring a higher computational cost. This makes time / step goes up as the batch size, but time / epoch goes down as larger batch size implies shorter steps per epoch.
>
> **Q4. Why do SupCon in Table 4 get worse with a larger batch size?**
>
> **The large batch size can stabilize the learning but can also reduce the regularization effect and result in a decreased test performance [3].** Please refer to the revised version of our paper and also (R3) of the overall response. SupCon uses multi-views that double the effective number of batch size. In the 64-batch experiment, doubling the effective batch size can make the learning stable; thus, SupCon outperformed SupCon-S, and SelfCon-M outperformed SelfCon-S. However, as the batch size becomes large, the stabilizing effect from multi-views decreases, and the necessity of regularization appears to be critical. Therefore, the optimal batch size of SupCon-S was higher than SupCon (SupCon-S 75.3% in 128-batch) since single-view itself helps regularization.
>
> **We believe that this is not the result of sub-optimally tuned hyperparameters (e.g., learning rate) because the hyperparameters exactly follow the setting in [1], and we have run their official code for reproducing SupCon experiments.** Also, we equally reduced the learning rate of SupCon and SelfCon in smaller batch sizes following the linear scaling rule. We are now running the sensitivity study of different learning rates and will inform you of the results as soon as possible.

---

> > ### Author Response · Authors · 2021-11-14
> > **Response to Reviewer ZiPE (2-2)**
> >
> > **Q4 (updated response). Why do SupCon in Table 4 get worse with a larger batch size?**
> >
> > You supposed that there could be a concern of using a sub-optimal learning rate on the large batch size. **Therefore, we ran the sensitivity study of different learning rates in a batch size of 1024 and summarized the results in Appendix I.** In conclusion, the performance comparison in Table 4 is consistent with hyperparameter tuning.

---

> > ### Author Response · Authors · 2021-11-16
> > **Response to Reviewer ZiPE (2-1)**
> >
> > **Q2 (updated response). Why don’t you experiment on full-scale ImageNet?**
> >
> > **In addition to the ImageNet results that were initially updated ($B=1024$ and 400 epochs), we did two more ablation studies to investigate SelfCon learning for the large-scale dataset.** (1) Pretraining with 800 epochs. (2) Pretraining with larger batch size ($B=2048$). In 800-epoch experiments, we observed the same trend as 400-epoch experiments, with higher overall classification accuracies. In 2048 batch size experiments, SelfCon-S was better than SupCon, even though SelfCon-S needs largely reduced time and memory consumption.

---

> > > ### Comment · Reviewer_ZiPE · 2021-11-28
> > > **Re: ImageNet**
> > >
> > > Thanks for these updated numbers.
> > >
> > > It appears to be the case that for larger scale training (larger batch sizes, larger models, full ImageNet), the performance improvements of the proposed method vanish (cf. Table 13, starts at +0.3, then +0.2, then +0.1).
> > >
> > > Also, at B=1024, SelfCon-S is outperformed by SupCon both after 400 and 800 epochs, but at B=2048, SelfCon-S outperforms SubCon. Could you please confirm that this is correct, and the 72.0 results in the last row is not for SelfCon-M?
> > >
> > > To what extend did you tune hyperparameters for this experiment? Table 15 is great, but shows how critical tuning of the hyperparameters is for a final result.

---

> > > > ### Author Response · Authors · 2021-11-28
> > > > **Response to Reviewer ZiPE (3-1)**
> > > >
> > > > Thanks for the further comments.
> > > >
> > > > **Q1. Performance improvements vanish for larger scale training.**
> > > >
> > > > * **Larger model: In Table 1, the performance gains in ResNet-18 experiments were consistently observed in ResNet-50 experiments.** For example, in the CIFAR-100 benchmark, we could see the increment from +2.4%p to +3.0%p, comparing SelfCon-S to SupCon.
> > > >
> > > > * **Larger batch: In Table 4, there was no decreasing trend for the gap between the multi-viewed method and the single-viewed counterpart.** Simply comparing 64 vs. 1024 batch size, we could see a larger gap in 1024 batch size.
> > > >
> > > > * **Full ImageNet: In Table 13, the number in the last row is for SelfCon-S, thus we cannot say that the performance gain of SelfCon-M vanished.** Although there is no result for SelfCon-M in 2048 batch size due to the lack of computational resources, we are also convinced that SelfCon-M has the same performance improvement (+0.3%p, +0.2%p) as 1024 batch size. (Also see the response to Q2 below.)
> > > >
> > > > **Q2. Is it correct that SelfCon-S outperforms SupCon at $B=2048$?**
> > > >
> > > > **In Table 4, we observed that multi-viewed methods outperform each counterpart for a small batch size (relative to the number of classes).** Therefore, in the case of ImageNet, we expected that SelfCon-M would show the best performance in a relatively small batch size (e.g., 1024 or 2048) and SelfCon-S would be the best when the batch size is increased.
> > > >
> > > > As we expected, in Table 13, SelfCon-M outperformed SupCon by +0.2%p or +0.3%p for 1024 batch size. Meanwhile, SelfCon-S was worse than SupCon in 1024 batch size but showed better performance in 2048 batch size. Perhaps, the performance gain will increase as the batch size becomes larger, and we are highly convinced that SelfCon-S will outperform even SelfCon-M in 3072 or 4096 batch size (as we observed in Table 4).
> > > >
> > > > **Q3. Did you tune hyperparameters for Table 13?**
> > > >
> > > > **We tuned the learning rate from 0.0625 to 0.25 in a short experiment of 70 epochs and 1024 batch size for SupCon and SelfCon-S.** We decided the optimal learning rate of 0.125, and note that it is different from the optimal learning rate of 0.0625 in the ImageNet-100 dataset. Besides, we tuned the sub-network structure among "fc", "small", and "same" (refer to Appendix C and the larger architecture, which is "same", showed the best).

---

> ### Author Response · Authors · 2021-11-26
> **Further Discussion**
>
> Dear Reviewer ZiPE,
>
> We sincerely appreciate your valuable comments for improving our work. We have put in efforts in the revision by adding experiments (e.g., full-scale ImageNet) to meet your needs about thoroughness and modifying the words and equations to clarify our motivation.
>
> We would like to send you a gentle reminder that the rolling discussion is now at the end of its period. We hope we have addressed your concerns, and we look forward to knowing if any of our response is still in question.
>
> Thanks,
> Authors

---

> ### Author Response · Authors · 2021-12-02
> **Looking forward to the further updates**
>
> *Dear Reviewer ZiPE,*
>
> We are pleased to have constructive discussions with you by your considerate feedback. We were encouraged that you gave an interest to our work and were motivated to revise our paper for completeness.
>
> Before the comment feature closes, we are willing to know that if there are some other concerns not resolved to make updates on the score.
>
> Best Regards,
> *Authors*

---

### Author Response · Authors · 2021-11-09
**Overall Response 1**

We thank all the reviewers for the detailed comments and constructive feedback. Before responding to the reviewers' comments, we would like to note some preliminary revisions in our paper. You can check the modifications that are marked in red-colored fonts. To respond to each reviewer's concerns and comments, we sincerely ask you to go through our comments below (we will upload each response as soon as possible).

**R1) Modifications in Eq. 1 and 2.**

We modified the notation and form of SupCon loss and SelfCon loss, thanks to the feedback from Reviewer ZiPE and Reviewer qb4T. Moreover, we removed the identity functions in Eq. 1 and 2 because the identity functions inside the log term should be outside the log. Similarly, we removed the identity functions in Eq. 40-42 in Appendix D.

**R2) ResNet-50 experiments for ImageNet-100 in Table 2.**

For a thorough comparison in a large-scale dataset, we also verified SelfCon learning with the ResNet-50 backbone network. We used the batch size of 256 for ResNet-50 experiments (see the setting in Appendix B).

**R3) Adding more experiments in Table 4.**

We added the results of $B$=128, $B$=512, and $B$=1024 cases in Table 4. Regarding the results, we revised the last paragraph of Section 5.3. Key findings are as follows. It is believed that a large batch size can reduce the regularization effect and result in decreasing the performance [1][2][3], and we observed a similar trend for the multi-viewed approach as the batch size increases. For example, in SupCon, multi-views double the effective batch size. Small batch sizes such as 64 can make the learning stable, but as the batch size becomes large, the stabilizing effect from multi-views decreases and the necessity of regularization appears to be critical. Hence, single-viewed approaches help regularization and have an optimal batch size larger than multi-viewed counterparts. For example, SelfCon-S, which includes single-view and sub-network, has a larger optimal batch size ($B$=256) since we found that single-view and sub-network both provide the regularization effect.

**R4) Emphasized 'supervised' framework.**

Overall, it might be confusing that we are dealing with a 'supervised' contrastive learning framework since prior works in contrastive learning are mainly discussed within the 'unsupervised' framework.

**R5) Fixed typo in Table 5.**

We corrected a typo of Tiny-ImageNet accuracy.

**R6) Modifications in Appendix A.2.**

We added the detailed explanation of negative samples $z_k$ in deriving SelfCon-S and SelfCon-M loss bound. Also, in Eq. 26-27, we revised the constant term by considering the multi-viewed batch of SelfCon-M.

**R7) Adding detailed experimental setting in Appendix D.1.**

We used different temperature values of 0.5 for the unsupervised SelfCon loss to be consistent with the setting of SimCLR [4].

**R8) Full-scale ImageNet experimental results in Appeindix G.**

We ran the experiments on full-scale ImageNet benchmarks and summarized the results in Appendix G.

**R9) Change the highlight color.**

We changed the highlight color from yellow to gray in the Tables. Note that we did not change any value in the Table.

**R10) Change the colormap in Figure 6, 7**

We changed the colormap from jet to viridis not to distort the visual perception of the result. Thanks to the reviewer ZiPE.

**References**
[1] You, Yang, Igor Gitman, and Boris Ginsburg. "Large batch training of convolutional networks." arXiv preprint arXiv:1708.03888 (2017).
[2] Luo, Ping, et al. "Towards understanding regularization in batch normalization." arXiv preprint arXiv:1809.00846 (2018).
[3] Wu, Jingfeng, et al. "On the noisy gradient descent that generalizes as sgd." International Conference on Machine Learning. PMLR, 2020.
[4] Chen, Ting, et al. "A simple framework for contrastive learning of visual representations." International conference on machine learning. PMLR, 2020.
[5] Khosla, Prannay, et al. "Supervised contrastive learning." arXiv preprint arXiv:2004.11362 (2020).

---

> ### Author Response · Authors · 2021-11-12
> **Overall Response 2**
>
> **R11) Modify Discussion 1.**
>
> To clearly explain the correlation between our theoretical analysis and the performance, we modified some sentences and changed the order of explanation for each term in Eq. 5.
>
> **R12) Modify Prop. 4.1.**
>
> We changed the upper bound of SupCon loss from $\mathcal{I}(x;z|c)$ to $\mathcal{I}(F(x);F(z)|c)$, to clarify the comparison between that of SelfCon loss. For this, part of the proof for Prop. 4.2 was moved to the proof for Prop. 4.1.
>
> **R13) Discussion for the connection between SelfCon loss and classification performance.**
>
> We added the detailed description to explain why SelfCon loss improves the classification performance in Appendix A.4. We hope that it resolves the questions of reviewer ZiPE, qb4T and hbNU.

---

> > ### Author Response · Authors · 2021-11-14
> > **Overall Response 4**
> >
> > **R15) Sensitivity study for learning rate in Appendix I.**
> >
> > In Table 4, we confirmed that supervised contrastive approaches degrade the performance in the large batch size. However, there could be a concern about the usage of sub-optimal learning rate. To mitigate the concern, **we further experimented with various learning rates for SupCon and SelfCon-S.** As referring to Appendix I, performance comparison in Table 4 is consistent with hyperparameter tuning.
> >
> > **R16) Ablation study for different augmentation policies in Appendix J.**
> >
> > We could think of scenarios where we have prior knowledge for augmentation. Also, there could be the opposite scenario where we do not know an optimal policy and just use simple augmentations. **Thus, we experimented with an Optimal(Strong) augmentation strategy and a Simple(Weak) augmentation strategy to compare the performances of SupCon and SelfCon-S under these scenarios.** The results are summarized in Table 16 in Appendix J.
> >
> > **R17) Additional experiment for correlation between MI and SelfCon loss in Appendix K.**
> >
> > **To clearly show the correlation between the MI and test accuracy, we additionally experimented with the correlation analysis.** As we have mentioned, we modified the Eq. 42 into a supervised version and tuned the hyperparameter $\alpha$ that controls the contribution of SelfCon loss. The updated results are found in Appendix K. We observed a clear increasing trend of both MI and test accuracy as the contribution of SelfCon gets larger.

---

> > ### Author Response · Authors · 2021-11-16
> > **Overall Response 3**
> >
> > **R14) Added more ImageNet experimental results in Appendix G.**
> >
> > **We added two ablation studies of ImageNet experiments, with respect to number of epochs and batch size for pretraining**. In 2048 batch size and 800 epochs, SelfCon-S was better than SupCon with largely reduced time and memory consumption.

---

### Decision · Program_Chairs · 2022-01-20

**Decision:**

Reject

**Comment:**

This paper modifies the loss of supervised contrastive (SupCon) learning by adding a self-contrastive loss. Utilizing a multi-exit network and contrasting the multiple outputs of this network, the proposed self-contrastive (SelfCon) learning removes the requirement of additional data augmentation samples for creating positive pairs. The proposed SelfCon loss is theoretically connected to the lower bound of a label conditional mutual information between the intermediate and last feature. The paper focuses its study on SupCon & SelfCon-M, which are multi-batch variates that first augment the images, and then contrast the views between both augmented and non-augmented samples of the same class, and on SupCon-S and SelfCon-S, which are single-batch variates that only contrast between the samples of the same class and do not require additional data augmentations. A wide variety of experiments have been done on CIFAR-10, CIFAR-100, TinyImageNet, ImageNet-100, and ImageNet, but mostly with relatively small networks.

The ratings for the paper were mixed [3,5,5,8 before rebuttal; 5,5,6,8 after rebuttal]. All four reviewers had provided detailed initial reviews, pointing out a long list of issues. The authors had incorporated these reviews to make a large number of improvements to their initial submission. After the author rebuttal period, while one reviewer raised the score from 5 to 6, two reviewers maintained their negative positions: Reviewer ZiPE is clearly concerned about the risk of accepting a method that may break as soon as a slightly larger model (ResNet50 instead 18) is used, the model is trained a bit longer, or the baselines are tuned, while Reviewer MBzi is unsatisfied with how the paper motivates its empirical construction from the perspective of mutual information maximization.

Given the disagreements between the reviewers, the AC has carefully read the paper to provide an additional review. Some concerning observations of the AC are summarized as follows:

1. Echoing the concern of Reviewer ZiPE, the performance gain of SelfCon-S over SupCon diminishes in ImageNet with ResNet-18, as shown in Table 13, making it become even more important for the authors to conduct experiments following more standard settings (e.g., ResNet-50 on ImageNet).

2. The main paper seems to suggest SelfCon-S outperforms SelfCon-M and SupCon outperforms SupCon-S, while Table 13 in the Appendix suggests the opposite.

3. Table 3 that compares SelfCon-S with SupCon appears very misleading, as SupCon consumes more memory and computation than SelfCon-S simply because it has used data augmentations. If SupCon-S is used, it would take less memory and computation than SelfCon-S.

4. SelfCon-S adds a subnetwork to the backbone to boost its performance, so technically, it has more parameters than the backbone. Comparing it with a baseline that only uses a backbone model does not seem to be that fair. This point has not been discussed in the paper.

5. Last but not least, echoing the concerns of Reviewer ZiPE and MBzi, the paper seems to try to validate the motivating of the added loss with mutual information maximization.  However, establishing the causal relationship between maximizing the mutual information of the intermediate and last layers and the classification performance needs much more than the correlation analysis provided in the paper.

Given the above-mentioned concerns, the AC does not consider the paper to be ready for publication at its current stage.